



# Aqueous SOA formation from the photo-oxidation of vanillin: Direct photosensitized reactions and nitrate-mediated reactions

Brix Raphael Go[1], Yan Lyu[1], Yan Ji[1], Dan Dan Huang[2], Xue Li[3], Theodora Nah[1], Chun Ho Lam[1], and Chak K. Chan[1]*

[1]School of Energy and Environment, City University of Hong Kong, Hong Kong, China
[2]Shanghai Academy of Environmental Sciences, Shanghai 200233, China
[3]Institute of Mass Spectrometry and Atmospheric Environment, Jinan University No. 601 Huangpu Avenue West, Guangzhou 510632, China

*Correspondence to*: Chak K. Chan (Chak.K.Chan@cityu.edu.hk)

**Abstract.** Vanillin (VL), a phenolic aromatic carbonyl abundant in biomass burning emissions, forms triplet excited states ($^3$VL*) under simulated sunlight leading to aqueous secondary organic aerosol (aqSOA) formation. This direct photosensitized oxidation of VL was compared with nitrate-mediated VL photo-oxidation under atmospherically relevant cloud and fog conditions, through examining the VL decay kinetics, product compositions, and light absorbance changes. The majority of the most abundant products from both VL photo-oxidation pathways were potential Brown carbon (BrC) chromophores. In addition, both pathways generated oligomers, functionalized monomers, and oxygenated ring-opening products, but nitrate promoted functionalization and nitration, which can be ascribed to its photolysis products (·OH, ·NO$_2$, and N(III), NO$_2^-$ or HONO). Moreover, a potential imidazole derivative observed from nitrate-mediated VL photo-oxidation suggested that ammonium may be involved in the reactions. The effects of secondary oxidants from $^3$VL*, pH, the presence of volatile organic compounds (VOCs) and inorganic anions, and reactants concentration and molar ratios on VL photo-oxidation were also explored. Our findings show that the secondary oxidants ($^1$O$_2$, O$_2^{·-}$/·HO$_2$, ·OH) from the reactions of $^3$VL* and O$_2$ play an essential role in VL photo-oxidation. Enhanced oligomer formation was noted at pH <4 and in the presence of VOCs and inorganic anions, probably due to additional generation of radicals (·HO$_2$ and CO$_3^{·-}$). Also, functionalization was dominant at low VL concentration, whereas oligomerization was favored at high VL concentration. Furthermore, guaiacol oxidation by photosensitized reactions of VL was observed to be more efficient relative to nitrate-mediated photo-oxidation. Lastly, potential VL photo-oxidation pathways under different reaction conditions were proposed. This study indicates that the direct photosensitized oxidation of VL, which nitrate photolysis products can further enhance, may be an important aqSOA source in areas influenced by biomass burning emissions.





## 1 Introduction

Aqueous reactions can be an important source of secondary organic aerosols (SOA) (Blando and Turpin, 2000; Volkamer et al., 2009; Lim et al., 2010; Ervens et al., 2011; Huang et al., 2011; Lee et al., 2011; Smith et al., 2014) such as highly-oxygenated and low-volatility organics (Hoffmann et al., 2018; Liu et al., 2019) which may affect aerosol optical properties due to contributions to Brown Carbon (BrC) (Gilardoni et al., 2016). BrC refers to organic aerosols that absorb radiation efficiently in the near-ultraviolet (UV) and visible regions (Laskin et al., 2015). The formation of aqueous SOA (aqSOA) via photochemical reactions involves oxidation, with hydroxyl radical ($^\bullet$OH) usually considered as the primary oxidant (Herrmann et al., 2010; Smith et al., 2014). The significance of photosensitized chemistry in atmospheric aerosols has recently been reviewed (George et al., 2015). For instance, triplet excited states of organic compounds ($^3$C*) from the irradiation of light-absorbing organics such as non-phenolic aromatic carbonyls (Canonica et al., 1995; Anastasio et al., 1996; Vione et al., 2006; Smith et al., 2014) have been reported to oxidize phenols at faster rates and with higher aqSOA yields compared to $^\bullet$OH (Sun et al., 2010; Smith et al., 2014; Yu et al., 2014; Smith et al., 2016). Aside from being an oxidant, $^3$C* can also be a precursor of singlet oxygen ($^1$O$_2$), superoxide (O$_2^{\bullet-}$) or hydroperoxyl ($^\bullet$HO$_2$) radical, and $^\bullet$OH (via HO$_2^\bullet$/O$_2^{\bullet-}$ formation) upon reactions with O$_2$ and substrates (e.g., phenols), respectively (Tinel et al., 2018). The $^3$C* concentration in typical fog water has been estimated to be >25 times than that of $^\bullet$OH, making $^3$C* the primary photo-oxidant for biomass burning phenolic compounds (Kaur and Anastasio, 2018; Kaur et al., 2019). Recent works on triplet-driven oxidation of phenols have mainly focused on changes of physicochemical properties (e.g., light absorption) and aqSOA yield (e.g., Smith et al., 2014, 2015, 2016), with few reports on reaction mechanisms and characterization of reaction products (e.g., Yu et al., 2014; Chen et al., 2020; Jiang et al., 2021).

Inorganic nitrate is a major component of aerosols and cloud/fog water. In cloud and fog water, the concentrations of inorganic nitrate can vary from 50 µM to >1000 µM, with higher levels typically noted under polluted conditions (Munger et al., 1983; Collett et al., 1998; Zhang and Anastasio, 2003; Li et al., 2011; Giulianelli et al., 2014; Bianco et al., 2020). Upon photolysis (Vione et al., 2006; Herrmann, 2007; Scharko et al., 2014), inorganic nitrate in cloud and fog water can contribute to BrC (Minero et al., 2007) and aqSOA formation (Huang et al., 2018; Klodt et al., 2019; Zhang et al., 2021) by generating $^\bullet$OH and nitrating agents (e.g., $^\bullet$NO$_2$). For example, the aqSOA yields from the photo-oxidation of phenolic carbonyls in nitrate are twice as high as that in sulfate solution (Huang et al., 2018). Nitration is a significant process in the formation of light-absorbing organics or BrC in the atmosphere (Jacobson, 1999; Kahnt et al., 2013; Mohr et al., 2013; Laskin et al., 2015; Teich et al., 2017; Li et al., 2020). Furthermore, nitrate photolysis has been proposed to be a potentially important process for SO$_2$ oxidation via the generation of $^\bullet$OH, $^\bullet$NO$_2$, and N(III) within particles (Gen et al., 2019a, 2019b), and it can also potentially change the morphology of atmospheric viscous particles (Liang et al., 2021). Accordingly, both $^3$C* and inorganic nitrate can contribute to aqSOA and BrC formation.

Biomass burning (BB) is a significant atmospheric source of both phenolic and non-phenolic aromatic carbonyls (Rogge et al., 1998; Nolte et al., 2001; Schauer et al., 2001; Bond et al., 2004). An example is vanillin (VL) (Henry's law


constant of $4.56 \times 10^5$ M atm$^{-1}$; Yaws, 1994), a model compound for methoxyphenols which are abundant in BB emissions
(Pang et al., 2019a), which has been shown to yield low-volatility products (Li et al., 2014) via aqueous ˙OH oxidation and
direct photodegradation. Photodegradation kinetics and aqSOA yields have been reported for direct VL photodegradation
(Smith et al., 2016), with oxygenated aliphatic-like compounds (high H:C, ≥1.5 and low O:C, ≤0.5 ratios) reported as the
most likely products (Loisel et al., 2021). Additionally, aqueous-phase reactions of phenols with reactive nitrogen species
have been proposed to be a significant source of nitrophenols and SOA (Grosjean, 1985; Kitanovski et al., 2014; Kroflič et
al., 2015; Pang et al., 2019a; Kroflič et al., 2021; Yang et al., 2021). For instance, nitrite-mediated VL photo-oxidation can
generate nitrophenols, and the reactions are influenced by nitrite/VL molar ratios, pH, and the presence of ˙OH scavengers
(Pang et al., 2019a).

As BB aerosols are typically internally mixed with other aerosol components (Zielinski et al., 2020), VL may

coexist with nitrate in BB aerosols. The aqueous-phase photo-oxidation of VL and nitrate may then reveal insights into the
atmospheric processing of BB aerosols. In addition, pollution from large BB events in central Amazonia has been reported to
interact with volatile organic compounds (VOCs) and soil dust (Rizzo et al., 2010). Moreover, the production, growth, and
chemical complexity of SOA can be influenced by the uptake and aerosol-phase reactions of VOCs (Pöschl, 2005; De Gouw
and Jimenez, 2009; Ziemann and Atkinson, 2012). Accordingly, studies incorporating other atmospherically relevant species
(e.g., VOCs and inorganic anions) in photo-oxidation experiments are warranted.

To evaluate the potential significance of VL and its reactions with nitrate in aqSOA formation in cloud/fog water,

we studied the direct photosensitized oxidation of VL and nitrate-mediated VL photo-oxidation under atmospherically
relevant conditions. In this work, reactions were characterized based on VL decay kinetics, light absorbance changes, and
products. The influences of secondary oxidants from VL triplets, solution pH, the presence of VOCs and inorganic anions,
and reactants concentration and molar ratios on these two photo-oxidation pathways were also assessed. The $^3C^*$ of non-
phenolic aromatic carbonyls (e.g., 3-4-dimethoxybenzaldehyde, DMB; a non-phenolic aromatic carbonyl) (Smith et al.,
2014; Yu et al., 2014; Jiang et al., 2021) and phenolic aromatic carbonyls (e.g., acetosyringone, vanillin) (Smith et al., 2016)
have been shown to oxidize phenols, but the reaction products from the latter are unknown. We then examined the photo-
oxidation of guaiacol, another non-carbonyl phenol, in the presence of VL and compared it with nitrate-mediated photo-
oxidation. Finally, we proposed photo-oxidation pathways of VL under different reaction conditions. This work presents a
comprehensive comparison of VL photo-oxidation by VL photosensitization and in the presence of inorganic nitrate.

## 2 Methods

### 2.1 Aqueous phase photo-oxidation experiments

Photo-oxidation experiments were performed in a 500-mL custom-built quartz photoreactor equipped with a magnetic
stirrer. The solutions were bubbled with synthetic air or nitrogen (N$_2$) (>99.995%) for 30 min before irradiation to achieve



air- and N$_2$-saturated conditions, respectively, and the bubbling was continued throughout the reactions (Du et al., 2011;
Chen et al., 2020). The aim of the air-saturated experiments was to enable the generation of secondary oxidants ($^1$O$_2$, O$_2^{\cdot-}$
/$^\cdot$HO$_2$, $^\cdot$OH) from $^3$VL* as O$_2$ is present. Conversely, the N$_2$-saturated experiments would inhibit the formation of these
secondary oxidants, leading to $^3$VL*-driven reactions. Solutions were irradiated through the quartz window of the reactor
using a xenon lamp (model 6258, Ozone free xenon lamp, 300 W, Newport) equipped with a longpass filter (20CGA-305 nm
cut-on filter, Newport) to eliminate light below 300 nm. Cooling fans positioned around the photoreactor and lamp housing
maintained reaction temperatures at 27±2 °C. The averaged initial photon flux in the reactor from 300 to 380 nm measured
using a chemical actinometer (2-nitrobenzaldehyde) was 2.6×10$^{15}$ photons cm$^{-2}$ s$^{-1}$ nm$^{-1}$ (Fig. S1). Although the
concentration of VL in cloud/fog water has been estimated to be <0.01 mM (Anastasio et al., 1996), a higher VL
concentration (0.1 mM) was used in this study to guarantee sufficient signals for product identification (Vione et al., 2019).
The chosen ammonium nitrate (AN) concentration (1 mM) was based on values observed in cloud and fog water (Munger et
al., 1983; Collett et al., 1998; Zhang and Anastasio, 2003; Li et al., 2011; Giulianelli et al., 2014; Bianco et al., 2020). We
also examined the role of VOCs (2-propanol, IPA) (1 mM) and inorganic anions (sodium bicarbonate, NaBC) (1 mM) in
these reactions. IPA can be classified as both a biogenic (from grass, Olofsson et al., 2003) and anthropogenic VOC (e.g.,
from solvents and industrial processes, Hippelein, 2004; Lewis et al., 2020), while bicarbonate is an inorganic anion
observed in fog water from both urban and rural locations (Collett et al., 1999; Straub et al., 2012; Straub, 2017). IPA and
NaBC are particularly interesting also because they can produce other radicals (e.g., $^\cdot$HO$_2$ and carbonate radical, CO$_3^{\cdot-}$) that
may react with nitrate photolysis products (Vione et al., 2009; Wang et al., 2021) and they can act as $^\cdot$OH scavengers
(Warneck and Wurzinger, 1988; Vione et al., 2009; Gen et al., 2019b; Pang et al., 2019a), although it must be noted that
these compounds were not added in excess for our experiments. Moreover, comparisons were made between the photo-
oxidation of guaiacol (0.1 mM), a non-carbonyl phenol, in the presence of VL (0.1 mM) or AN (1 mM). Samples (10 mL)
were collected hourly for a total of 6 h for offline optical and chemical analyses. Absorbance measurements, VL (and GUA)
decay kinetics (calibration curves for VL and GUA standard solutions; Fig. S2), small organic acids measurements, and
product characterization were conducted using UV-Vis spectrophotometry, ultra-high-performance liquid chromatography
with photodiode array detector (UHPLC-PDA), ion chromatography (IC), and UHPLC coupled with quadrupole time-of-
flight mass spectrometry (UHPLC-qToF-MS) equipped with an electrospray ionization (ESI) source and operated in the
positive ion mode (the negative ion mode signals were too low for our analyses), respectively. Each experiment was repeated
independently at least three times and measurements were done in triplicate. Details on the materials and analytical
procedures are provided in the Supporting Information (Text S1 to S6). The pseudo-first-order rate constant ($k$') for VL
decay was determined using the following equation (Huang et al., 2018):

$$ln\,([VL]_t/[VL]_0)\,=\,-\,k't \qquad\qquad\qquad\text{(Eq. 1)}$$





where $[VL]_t$ and $[VL]_0$ are the concentrations of VL at time $t$ and 0, respectively. Replacing VL with GUA in Eq. 1 enabled
the calculation of GUA decay.

**2.2 Calculation of normalized abundance of products**

The normalized abundance of a product, [P] (unitless), was calculated as follows:
$$[P] = \frac{A_{P,t}}{A_{VL,t}} \cdot \frac{[VL]_t}{[VL]_0} \qquad\qquad (Eq.\ 2)$$
where $A_{P,t}$ and $A_{VL,t}$ are the extracted ion chromatogram (EIC) signal peak areas of the product P and VL from UHPLC-
qToF-MS analyses at time $t$, respectively; $[VL]_t$ and $[VL]_0$ are the VL concentrations (μM) determined using UHPLC at time
$t$ and 0, respectively. Here, we relied on the more accurate measurements of [VL] using UHPLC for semi-quantification. It
should be noted that the ionization efficiency may greatly vary for different classes of compounds (Kebarle, 2000). Hence,
we assumed equal ionization efficiency of different compounds to calculate their normalized abundance, which is commonly
used to estimate O:C ratios of SOA (Bateman et al., 2012; Lin et al., 2012; De Haan et al., 2019). Typical fragmentation
behavior observed in MS/MS spectra for individual functional groups from Holčapek et al. (2010) are outlined in Table S1.

**3 Results and Discussion**

**3.1 Kinetics, mass spectrometric, and absorbance changes analyses during aqueous phase photo-oxidation of vanillin**

Table S2 summarizes the reaction conditions, initial VL (and GUA) decay rates, normalized abundance of products, and
average carbon oxidation state ($<OS_c>$ (of the 50 most abundant products). In general, the 50 most abundant products
contributed more than half of the total normalized abundance of products. For clarity purposes, the reactions involving
reactive species referred to in the following discussions are provided in Table 1.
As shown in Figure S3, VL underwent oxidation both directly and in the presence of nitrate upon simulated sunlight
illumination. VL absorbs light and is promoted to its excited singlet state ($^1VL^*$), then undergoes intersystem crossing (ISC)
to the excited triplet state, $^3VL^*$. In principle, $^3VL^*$ can oxidize ground-state VL (Type I photosensitized reactions) via H-
atom abstraction/electron transfer and form $O_2^{\bullet-}$ or $HO_2^{\bullet}$ in the presence of $O_2$ (Tinel et al., 2018), or react with $O_2$ (Type II
photosensitized reactions) to yield $^1O_2$ via energy transfer or $O_2^{\bullet-}$ via electron transfer (Lee et al., 1987; Foote et al., 1991).
The disproportionation of $HO_2^{\bullet}/O_2^{\bullet-}$ (Anastasio et al., 1996) and reaction of $HO_2^{\bullet}$ with $O_2^{\bullet-}$ (Du et al., 2011) form hydrogen
peroxide ($H_2O_2$), which is a photolytic source of $^{\bullet}OH$. Overall, air-saturated conditions, in which $O_2$ is present, enable the
generation of secondary oxidants from $^3VL^*$ ($^1O_2$, $O_2^{\bullet-}/^{\bullet}HO_2$, $^{\bullet}OH$). Moreover, $^{\bullet}OH$, $^{\bullet}NO_2$, and $NO_2^-/HNO_2$, i.e., N(III),
generated via nitrate photolysis (Reactions 1-3; Table 1) can also oxidize or nitrate VL. In this work, the direct
photosensitized oxidation of VL (by $^3VL^*$ or secondary oxidants from $^3VL^*$ and $O_2$) and nitrate-mediated VL photo-
oxidation are referred to as VL* and VL+AN, respectively.





### 3.1.1 Effect of secondary oxidants from VL triplets

As mentioned earlier, secondary oxidants ($^1O_2$, $O_2^{\bullet-}/^{\bullet}HO_2$, $^{\bullet}OH$) can be generated from $^3VL^*$ when $O_2$ is present (e.g., under air-saturated conditions), while $^3VL^*$ is the only oxidant expected under $N_2$-saturated conditions. To examine the contributions of $^3VL^*$-derived secondary oxidants and $^3VL^*$ only on VL photo-oxidation, experiments under both air- and $N_2$-saturated conditions (Fig. S3a) were carried out at pH 4, which is representative of moderately acidic aerosol and cloud pH values (Pye et al., 2020). No significant VL loss was observed for dark experiments. The low decay rate for VL* under $N_2$-saturated conditions suggests a minimal role for $^3VL^*$ in VL photo-oxidation. Contrastingly, the VL* decay rate under air-saturated conditions was 4 times higher, revealing the importance of $^3VL^*$-derived secondary oxidants for photosensitized oxidation of VL. Aside from $^{\bullet}OH$, $O_2^{\bullet-}/^{\bullet}HO_2$ and $^1O_2$ can also promote VL photo-oxidation (Kaur and Anastasio, 2018; Chen et al., 2020). $^1O_2$ is also an efficient oxidant for unsaturated organic compounds and has a lifetime that is much longer than $^3C^*$ (Chen et al., 2020). Similar to VL*, the decay rate for VL+AN under air-saturated conditions was faster (6.6 times) than $N_2$-saturated conditions, which can be due to several reactions facilitated by nitrate photolysis products and the enhancement of N(III)-mediated photo-oxidation in the presence of $O_2$ as reported in early works (Vione et al., 2005; Kim et al., 2014; Pang et al., 2019a). An example is enhanced VL nitration likely from increased $^{\bullet}NO_2$ formation such as from the reaction of $^{\bullet}OH$ and $O_2^{\bullet-}$ with $NO_2^-$ (Reactions 4 and 5, respectively; Table 1) or the autoxidation of $^{\bullet}NO$ from $NO_2^-$ photolysis (Reactions 6-9; Table 1) in aqueous solutions (Pang et al., 2019a). Reactions involving $^{\bullet}HO_2/O_2^{\bullet-}$ which may originate from the photolysis of nitrate alone, likely from the production and subsequent photolysis of peroxynitrous acid (HOONO) (Reaction 10; Table 1) (Jung et al., 2017; Wang et al., 2021), or the reactions of $^3VL^*$ in the presence of $O_2$, may have contributed as well. For instance, Wang et al. (2021) recently demonstrated that nitrate photolysis generates $^{\bullet}HO_2/O_2^{\bullet-}{}_{(aq)}$ and $HONO_{(g)}$ in the presence of dissolved aliphatic organic matter (e.g., nonanoic acid, ethanol), with the enhanced $HONO_{(g)}$ production caused by secondary photochemistry between $^{\bullet}HO_2/O_2^{\bullet-}{}_{(aq)}$ and photoproduced $NO_{x(aq)}$ (Reactions 11 and 12; Table 1), in agreement with Scharko et al. (2014). The significance of this increased HONO production is enhanced $^{\bullet}OH$ formation (Reaction 13; Table 1). In addition, $^{\bullet}HO_2$ can react with $^{\bullet}NO$ (Reaction 10; Table 1) from $NO_2^-$ photolysis (Reaction 6; Table 1) to form HOONO, and eventually $^{\bullet}NO_2$ and $^{\bullet}OH$ (Reaction 14; Table 1) (Pang et al., 2019a). Nevertheless, the comparable decay rates for VL* and VL+AN imply that VL* chemistry still dominates even at 1:10 molar ratio of VL/nitrate, probably due to the much higher molar absorptivity of VL compared to that of nitrate (Fig. S1) and the high VL concentration (0.1 mM) used in this study. Although we have no quantification of the oxidants in our reaction systems as it is outside the scope of this study, these observations clearly substantiate that secondary oxidants from $^3VL^*$, which are formed when $O_2$ is present, are required for efficient photosensitized oxidation of VL and nitrate-mediated VL photo-oxidation.

The products from VL* under $N_2$-saturated conditions were mainly oligomers (e.g., $C_{16}H_{14}O_4$) (Fig. 1a), consistent with triplet-mediated oxidation forming higher molecular weight products, probably with less fragmentation relative to oxidation by $^{\bullet}OH$ (Chen et al., 2020). A threefold increase in the normalized abundance of products was noted upon addition



of nitrate (VL+AN under $N_2$-saturated conditions; Fig. 1b), and in addition to oligomers, functionalized monomers (e.g.,
$C_8H_6O_5$) and nitrogen-containing compounds (e.g., $C_8H_9NO_3$; No. 2, Table S3) were also observed, in agreement with ·OH-
initiated oxidation yielding more functionalized/oxygenated products compared to triplet-mediated oxidation (Chen et al.,
2020). Compared to $N_2$-saturated conditions, the normalized abundance of products such as oligomers and functionalized
monomers (e.g., demethylated VL; Fig. S4) were significantly higher under air-saturated conditions (Figs. 1c-d), likely due
to the secondary oxidants from $^3VL^*$ and $O_2$ and their interactions with nitrate photolysis products. The nitrogen-containing
compounds (e.g., $C_{16}H_{10}N_2O_9$; No. 3, Table S3) were also more relatively abundant under air-saturated conditions. For both
VL* and VL+AN under air-saturated conditions, the most abundant product was $C_{10}H_{10}O_5$ (No. 4, Table S3), a substituted
VL. Irradiation of VL by 254-nm has also been reported to lead to VL dimerization and functionalization via ring-retaining
pathways, as well as small oxygenates but only when ·OH from $H_2O_2$ were involved (Li et al., 2014). In this work, small
organic acids were observed from both VL* and VL+AN under air-saturated conditions (Fig. S5) due to simulated sunlight
that could access the 308-nm VL band (Smith et al., 2016). Interestingly, we observed a potential imidazole derivative
($C_5H_5N_3O_2$; Fig. 1d) from VL+AN under air-saturated conditions, which may have formed from reactions induced by
ammonium. This compound was not observed in a parallel experiment in which AN was replaced with sodium nitrate (SN)
(Fig. S6a; see Sect. 3.3 for discussion). The molecular transformation of VL upon photo-oxidation was examined using the
van Krevelen diagrams (Fig. S7). For all experiments (A1-19; Table S2) in this study, the O:C and H:C ratios of the products
were mainly similar to those observed from the aging of other phenolics (Yu et al., 2014) and BB aerosols (Qi et al., 2019).
Oligomers with O:C ratios ≤0.6 were dominant in VL* under $N_2$-saturated conditions. For VL+AN under $N_2$-saturated
conditions, smaller molecules ($n_c$ ≤8) with higher O:C ratios (up to 0.8) were also observed. More products with higher O:C
ratios (≥0.6) were noted under air-saturated conditions for both VL* and VL+AN. The H:C ratios were mostly around 1.0,
indicating that the products for experiments A5 to A8 (Table S2) were mainly aromatic species. Compounds with H:C ≤1.0
and O:C ≤0.5 are common for aromatic species, while compounds with H:C ≥1.5 and O:C ≤0.5 are typical for more aliphatic
species (Mazzoleni et al., 2012; Kourtchev et al., 2014; Jiang et al., 2021). Moreover, majority of the products for
experiments A5 to A8 have double bond equivalent (DBE) values >7, which corresponds to oxidized aromatic compounds
(Xie et al., 2020). In contrast, Loisel et al. (2021) reported mainly oxygenated aliphatic-like compounds (H:C, ≥1.5 and O:C,
≤0.5 ratios) from the direct irradiation of VL (0.1 mM), which may be due to their use of ESI in the negative ion mode,
which has higher sensitivity for detecting compounds such as carboxylic acids (Holčapek et al., 2010; Liigand et al., 2017)
and solid-phase extraction (SPE) procedure causing the loss of some oligomers (LeClair et al., 2012; Zhao et al., 2013;
Bianco et al., 2018). Among experiments A5 to A8 (Table S2), VL+AN under air-saturated conditions (A7) had the highest
normalized abundance of products and <$OS_c$>, most probably due to the combined influence of the secondary oxidants from
$^3VL^*$ and $O_2$, and nitrate photolysis products. In our calculations, the increase in <$OS_c$> (except for VOCs and inorganic
anions experiments; A9 to A12; Table S2) was lower than those in ·OH- or triplet-mediated oxidation of phenolics (e.g.,
phenol, guaiacol) measured using an aerosol mass spectrometer (Sun et al., 2010; Yu et al., 2014), likely because we
excluded contributions from ring-opening products which may have higher $OS_c$ values as these products are not detectable in



the positive ion mode. Thus, the $<OS_c>$ in this study likely were lower estimates. In brief, the secondary oxidants from $^3VL^*$
and $O_2$ increased the abundance of products and promoted the formation of more oxidized aqSOA. These trends were
reinforced in the presence of nitrate, indicating synergistic effects between secondary oxidants from VL triplets and nitrate
photolysis products.

Illumination of phenolic aromatic carbonyls with high molar absorptivities ($\varepsilon_{\lambda max}$) (~8 to 22 × $10^3$ $M^{-1}$ $cm^{-1}$) leads to
an overall loss of light absorption but increased absorbance at longer wavelengths (>350 nm), where the carbonyls did not
initially absorb light (Smith et al., 2016). Fig. 2a illustrates the changes in total absorbance from 350 to 550 nm of VL* and
VL+AN under air- and $N_2$-saturated conditions. The absorption spectra of VL* under air- and $N_2$- saturated conditions (pH
4) at different time intervals are shown in Fig. S8. For both VL* and VL+AN, evident absorbance enhancement was
observed under air-saturated conditions, while the absorbance changes under $N_2$-saturated conditions were minimal,
consistent with the VL decay trends. This absorbance enhancement can be explained by the formation of oligomers with
large, conjugated π-electron systems (Chang and Thompson, 2010) and hydroxylated products (Li et al., 2014; Zhao et al.,
2015), in agreement with the observed reaction products. In this work, phenoxy radicals can be generated from several
processes such as the oxidation (Vione et al., 2019) of ground-state VL by $^3VL^*$ via H-atom abstraction (Huang et al., 2018)
and photoinduced O-H bond-breaking (Berto et al., 2016). Moreover, $^3VL^*$ can initiate H-atom abstraction from the -CHO
group of VL, generating ketyl radicals via Norrish-type reactions (Vione et al., 2019). Also, similar reactions can be initiated
by ˙OH (Gelencsér et al., 2003; Hoffer et al., 2004; Chang and Thompson, 2010; Sun et al., 2010), which in this study can be
generated from the reaction between $^3VL^*$ and $O_2$, as well as nitrate photolysis. Oligomers can then form via the coupling of
phenoxy radicals or phenoxy and ketyl radicals (Sun et al., 2010; Berto et al., 2016; Vione et al., 2019). Absorbance increase
at >350 nm has also been reported for photosensitized oxidation of phenol and 4-phenoxyphenol (De Laurentiis et al., 2013a,
2013b) and direct photolysis of tyrosine and 4-phenoxyphenol (Bianco et al., 2014) in which dimers have been identified as
initial substrates. The continuous absorbance enhancement throughout 6 h of irradiation correlated with the observation of
oligomers and nitrated compounds after irradiation. However, the increasing concentration of small organic acids (Fig. S5)
throughout the experiments suggests that fragmentation, which results in the decomposition of initially formed oligomers
and formation of smaller oxygenated products (Huang et al., 2018), is important at longer irradiation times. Overall, these
trends establish that secondary oxidants from $^3VL^*$and $O_2$ are necessary for the efficient formation of light-absorbing
compounds from both VL* and VL+AN.
**3.1.2 Effect of pH**
The reactivity of $^3C^*$ (Smith et al., 2014, 2015, 2016), aromatic photonitration by nitrate (Machado and Boule, 1995;
Dzengel et al., 1999; Vione et al., 2005; Minero et al., 2007), and N(III)-mediated VL photo-oxidation (Pang et al., 2019a)
have been demonstrated to be pH-dependent. In this study, the effect of pH on VL photo-oxidation was investigated within
the range of 2.5 to 5, corresponding to typical cloud (2-7) pH values (Pye et al., 2020). The decay rates for both VL* and
VL+AN increased as pH decreased (VL* and VL+AN at pH 2.5: 1.5 and 1.3 times faster than at pH 4, respectively) (Fig.





S3b). For VL*, this pH trend indicates that $^3$VL* are more reactive in their protonated form, which is opposite to that
reported for 0.005 mM VL (Smith et al., 2016), likely due to the concentration dependence of the relative reactivities of
protonated and neutral forms of $^3$VL*. It has been reported that the quantum yield for direct VL photodegradation is higher at
pH 5 than at pH 2 for 0.005 mM VL, but they are not statistically different for 0.03 mM VL (Smith et al., 2016). Also,
increases in hydrogen ion concentration can enhance the formation of HO$_2$$^\bullet$ and H$_2$O$_2$ and in turn, $^\bullet$OH formation (Du et al.,
2011). In addition to these pH influences on VL*, the dependence of N(III) (NO$_2$$^-$ + HONO) speciation on solution acidity
(Pang et al., 2019a) also contributed to the observed pH effects for VL+AN. At pH 3.3, half of N(III) exists as HONO
(Fischer and Warneck, 1996; Pang et al., 2019a), which has a higher quantum yield for $^\bullet$OH formation than that of NO$_2$$^-$ in
the near-UV region (Arakaki et al., 1999; Kim et al., 2014). The increased $^\bullet$OH formation rates as pH decreases can lead to
faster VL decay (Pang et al., 2019a). Also, NO$_2$$^-$/HONO can generate $^\bullet$NO$_2$ via oxidation by $^\bullet$OH (Reactions 4 and 15; Table
1) (Pang et al., 2019a). As pH decreases, the higher reactivity of $^3$VL* and HONO being the dominant N(III) species can
lead to faster VL photo-oxidation.

As pH decreased, the normalized abundance of products, particularly oligomers and functionalized monomers, was

higher for both VL* and VL+AN, further indicating that $^3$VL* are more reactive in their protonated form. The most
abundant products observed were a substituted VL (C$_{10}$H$_{10}$O$_5$) and VL dimer (C$_{16}$H$_{14}$O$_6$; No. 5, Table S3) at pH 4 and pH
<4, respectively (Figs. 1c-h). Furthermore, a tetramer was observed only in VL* at pH 2.5. For VL+AN, the normalized
abundance of nitrogen-containing compounds also increased at lower pH (Table S2), likely due to increased $^\bullet$OH and $^\bullet$NO$_2$
formation. The potential imidazole derivative (C$_5$H$_5$N$_3$O$_2$) was observed only at pH 4 possibly due to the pH dependence of
ammonium speciation (p$K_a$ = 9.25). Imidazole formation requires the nucleophilic attack of ammonia on the carbonyl group
(Yu et al., 2011), and at pH 4, the concentration of dissolved ammonia in VL+AN was about 10 or 30 times higher than that
at pH 3 or pH 2.5, respectively. At different pH, the O:C and H:C ratios in VL* and VL+AN had no significant differences
(Figs. S7c-d and S9), but molecules with higher O:C ratios (>0.6) were more abundant at pH <4. Accordingly, the <OS$_c$> at
pH <4 for both VL* and VL+AN were higher than that at pH 4, consistent with higher <OS$_c$> observed at pH 5 compared to
pH 7 for the $^\bullet$OH-mediated photo-oxidation of syringol (Sun et al., 2010). Essentially, the higher reactivity of $^3$VL* and
predominance of HONO over nitrite at lower pH result in increased formation of products mainly composed of oligomers
and functionalized monomers.

The higher absorbance enhancement for both VL* and VL+AN (Fig. 2b) as pH increased may be attributed to

redshifts and increased visible light absorption of reaction products (Pang et al., 2019a). When a phenolic molecule
deprotonates at higher pH, an ortho- or para- electron-withdrawing group, such as a nitro or aldehyde group, can attract a
portion of the negative charge towards its oxygen atoms through induced and conjugated effects, leading to the extension of
chromophore from the electron-donating group (e.g., -O$^-$) to the electron-withdrawing group via the aromatic ring (Carey,
2000; Williams and Fleming, 2008; Pang et al., 2019a). Hence, the delocalization of the negative charge in phenolates leads
to significant redshifts (Mohr et al., 2013).





### 3.1.3 Effect of VOCs and inorganic anions

Aerosols are a complex mix of organic and inorganic compounds (Kanakidou et al., 2005). We explored the photo-oxidation behavior of VL, with and without nitrate, in the presence of VOCs (2-propanol, IPA) and inorganic anions (sodium bicarbonate, NaBC). For both VL* and VL+AN, there was no significant change in VL decay (Figs. S3c-d), and comparable absorbance enhancements (Figs. 2c-d) were observed upon the addition of IPA and NaBC. However, the characterization of reaction products revealed the distinct effects of these compounds on the photo-oxidation of VL. Both IPA and NaBC increased the normalized abundance of products from VL* (by a factor of 2.4 and 1.4, respectively) and VL+AN (by a factor of ~4) (Table S2). The major product observed in VL*+IPA (Fig. S10a) was a dimer ($C_{16}H_{14}O_6$). Also, higher oligomers up to tetramers (e.g., $C_{31}H_{22}O_{12}$) not observed in VL* were noted. A possible explanation may be the additional generation of $^\bullet HO_2$ from the reaction of IPA with $^\bullet OH$ (Warneck and Wurzinger, 1988) (Reactions 16 and 17; Table 1), which can originate from $^3VL^*$ or nitrate photolysis, inducing reactions such as oxidation and nitration. As discussed earlier, $^\bullet HO_2$ can form $H_2O_2$, a photolytic source of $^\bullet OH$ (Anastasio et al., 1996; Du et al., 2011). In the presence of IPA, the increase in normalized abundance of products (VL+AN+IPA: 3.8 times vs. VL*+IPA: 2.4 times; Table S2) and $<OS_c>$ (VL+AN+IPA: -0.13 to 0.08 vs. VL*+IPA: -0.16 to -0.10; Table S2) being more evident for VL+AN compared to VL* also supports the potential importance of reactions involving $^\bullet HO_2$ and nitrate photolysis products such as the secondary photochemistry between $^\bullet HO_2/O_2^{\bullet-}{}_{(aq)}$ and photoproduced $NO_{x(aq)}$ enhancing $HONO_{(g)}$ production from nitrate photolysis in the presence of dissolved aliphatic organic matter (Wang et al., 2021) as discussed in Sect. 3.1.1. This chemistry may have operated in VL+AN+IPA considering that $^\bullet HO_2/O_2^{\bullet-}$ may originate from multiple sources in this experiment: nitrate photolysis (Reaction 10; Table 1) (Jung et al., 2017; Wang et al., 2021), the reactions of $^3VL^*$ in the presence of $O_2$ (see Sect. 3.1), or reaction of IPA with $^\bullet OH$ (Warneck and Wurzinger, 1988) (Reactions 16 and 17; Table 1). In other words, the role of nitrate in VL photo-oxidation is enhanced in the presence of IPA, likely due to additional $^\bullet HO_2/O_2^{\bullet-}$ formation. In VL+AN+IPA, nitrate photolysis likely converted $C_{16}H_{14}O_6$ (from VL*+IPA) to $C_{15}H_{12}O_8$ (Figs. S10a-b) via demethylation and then multiple hydroxylations. Nitrate photolysis generates $^\bullet OH$, and demethylation has been reported to be enhanced at high $^\bullet OH$ exposure (Gold et al., 1983). Moreover, alcohols can affect the structure of water, causing a localized patterning or organization that changes the solvation environment, which can account for reactivity enhancement in the presence of alcohol-containing solvents (Berke et al., 2019). Berke et al. (2019) has demonstrated that IPA and other alcohols (e.g., ethanol) can promote the production of light-absorbing compounds, i.e., imidazoles, from the reactions between glyoxal and ammonium sulfate. This phenomenon has been attributed to the formation of micro-heterogeneities of hydrated alcohol molecules in a complex solution environment composed of solvated sulfate ions and a mixture of reactants and products upon the addition of alcohols. As proposed by an earlier study (Onori and Santucci, 1996), if the water in the SOA-mimicking solutions exists in two forms, bulk and hydrating, the micro-heterogeneities may interact with water/nitrate matrix to sequester the reactants and products, concentrating them within a smaller effective solvent volume and consequently resulting in increased normalized abundance of products (Berke et al., 2019).





For NaBC which does not produce $^•HO_2$ upon reactions with $^•OH$ under air-saturated conditions (Gen et al., 2019b),
the increased normalized abundance of products may be due to other reactions promoted by the carbonate radical ($CO_3^{•-}$),
which can be generated from the reactions of bicarbonate/carbonate with $^•OH$ (Reactions 18 and 19; Table 1) (Neta et al.,
1988; Wojnárovits et al., 2020) or $^3VL^*$ (Reactions 20 and 21; Table 1) (Canonica et al., 2005). $CO_3^{•-}$ is a selective oxidant
that reacts with organic molecules at a lower rate than $^•OH$ and readily reacts with electron-rich parts of phenols, aromatic
amines, and sulfur-containing compounds (e.g., glutathione) through both electron transfer and H-abstraction (Huang and
Mabury, 2000; Wojnárovits et al., 2020). Similar to IPA, the enhancement of normalized abundance of products
(VL+AN+NaBC: 4.3 times vs. VL*+NaBC: 1.4 times; Table S2) and $<OS_c>$ (VL+AN+NaBC: -0.13 to 0.08 vs.
VL*+NaBC: -0.16 to -0.11; Table S2) was more obvious for VL+AN+NaBC than VL*+NaBC, further underlining the
contributions of nitrate photolysis products. For example, it has been reported that carbonate and bicarbonate can
substantially increase the photodegradation of electron-rich compounds (e.g., catechol) by nitrate (Vione et al., 2009).
Bicarbonate can enhance the photolysis of nitrate via a solvent-cage effect, reacting with photolysis-derived $^•OH$ before it
escapes the surrounding cage of the water molecules. This prevents the recombination of $^•OH$ and $^•NO_2$ inside the solvent
cage that otherwise would yield back $NO_3^- + H^+$, which reduces the quantum yield of $^•OH$ photoproduction (Bouillon and
Miller, 2005). This scavenging of in-cage $^•OH$ by bicarbonate would then hinder recombination, resulting in a higher
generation rate of $CO_3^{•-} + ^•OH$ with bicarbonate compared to $^•OH$ alone without bicarbonate. However, in our experiments,
NaBC did not cause any substantial change in the decay of VL for both VL* and VL+AN, although it promoted higher
normalized abundance of products. The major product in VL*+NaBC was a functionalized monomer ($C_7H_4O_4$; No. 6, Table
S3; Fig. S10c). Unlike VL*+IPA, no tetramers were observed in VL*+NaBC. Similar to VL+AN+IPA, the addition of
NaBC to VL+AN resulted in trimers and a high-abundance dimer ($C_{15}H_{12}O_8$; No. 7, Table S3) (Figs. S10b and S10d).
Overall, VL+AN+IPA had more oligomers while VL+AN+NaBC had more functionalized monomers (e.g., $C_8H_6O_4$; No. 8,
Table S3). These findings suggest that aside from low pH (<4), the formation of oligomers from VL photo-oxidation can
also be promoted by presence of VOCs and inorganic anions likely via the generation of radicals such as $^•HO_2$ and $CO_3^{•-}$
which can also interact with nitrate photolysis products.
The addition of IPA or NaBC to VL* resulted in products with higher O:C and H:C ratios (Figs. S11a and S11c).
Although the products were more abundant in VL*+IPA than with NaBC, the distribution of their products in van Krevelen
diagrams was rather similar. The increased in $<OS_c>$ in the presence of IPA or NaBC was more significant for VL+AN than
VL*, likely due to the interactions of nitrate photolysis products with $^•HO_2$ and $CO_3^{•-}$. For VL+AN, IPA and NaBC also
increased the O:C and H:C ratios (Figs. S11b and S11d), and most products had $OS_c$ >0, similar to less volatile and semi-
volatile oxygenated organic aerosols (LV-OOA and SV-OOA) (Kroll et al., 2011).





### 3.1.4 Distribution of potential BrC compounds

Figure S12 plots the DBE values vs. number of carbons ($n_c$) (Lin et al., 2018) for the 50 most abundant products from pH 4 experiments under air-saturated conditions, along with reference to DBE values corresponding to fullerene-like hydrocarbons (Lobodin et al., 2012), cata-condensed polycyclic aromatic hydrocarbons (PAHs) (Siegmann and Sattler, 2000), and linear conjugated polyenes with a general formula $C_xH_{x+2}$. As light absorption by BrC requires uninterrupted conjugation across a significant part of the molecular structure, compounds with DBE/$n_C$ ratios (shaded area in Fig. S12) greater than that of linear conjugated polyenes are potential BrC compounds (Lin et al., 2018). Based on this criterion and the observed absorbance enhancement at >350 nm (Fig. 2), the majority of the 50 most abundant products from pH 4 experiments under air-saturated conditions were potential BrC chromophores composed of monomers and oligomers up to tetramers. However, as ESI-detected compounds in BB organic aerosols has been reported to be mainly molecules with $n_c$ <25 (Lin et al., 2018), there may be higher oligomers that were not detected in our reaction systems.

### 3.2 Effect of reactants concentration and molar ratios on the aqueous photo-oxidation of vanillin

To examine the influence of VL and nitrate concentration and their molar ratios on VL photo-oxidation, we also characterized the reaction products from lower [VL] (0.01 mM VL*; A14; Table S2), lower concentrations and an equal molar ratio of VL/nitrate (0.01 mM VL + 0.01 mM AN; A15; Table S2), and lower [VL] and 1:100 molar ratio of VL/nitrate (0.01 mM VL + 1 mM AN; A16; Table S2) at pH 4. The normalized abundance of products from low [VL] experiments (A14-A16; Table S2) were up to 1.4 times higher than that of high [VL] experiments (A5 and A7; Table S2). Nevertheless, the major products for both low and high [VL] experiments were functionalized monomers (Figs. 1c-d and S13a-c) such as $C_8H_6O_4$ and $C_{10}H_{10}O_5$. For both VL* and VL+AN, the contribution of <200 m/z to the normalized abundance of products was higher at low [VL] than at high [VL], while the opposite was observed for >300 m/z (Fig. S13d). This indicates that functionalization was favored at low [VL], as supported by the higher <$OS_c$>, while oligomerization was the dominant pathway at high [VL], consistent with more oligomers or polymeric products reported from high phenols concentration (e.g., 0.1 to 3 mM) (Li et al., 2014; Slikboer et al., 2015; Ye et al., 2019). A possible explanation is that at 1:1 VL/nitrate, VL efficiently competes with $NO_2^-$ for •OH (from nitrate or nitrite photolysis, Reaction 4; Table 1) and indirectly reduces •$NO_2$. Similarly, hydroxylation has been suggested to be an important pathway for 1:1 VL/nitrite than in 1:10 VL/nitrite (Pang et al., 2019a). This may also be the reason why 1:1 VL/nitrate (A15; Table S2) had higher <$OS_c$> than 1:100 (A16; Table S2) VL/nitrate but had fewer N-containing compounds compared to the latter. Moreover, the contribution of <200 m/z to the normalized abundance of products was higher for 1:1 than 1:100 VL/nitrate molar ratio, further suggesting the formation of more oxidized products.





### 3.3 Participation of ammonium in the aqueous photo-oxidation of vanillin


Imidazole and imidazole derivatives have been reported to be the major products of glyoxal and ammonium sulfate reactions
at pH 4 (Galloway et al., 2009; Yu et al., 2011; Sedehi et al., 2013; Gen et al., 2018; Mabato et al., 2019). Here, we
compared VL+AN and VL+SN at pH 4 in terms of reaction products and oxidative characteristics to confirm the
participation of ammonium in the aqueous photo-oxidation of VL. In both experiments, the normalized abundance of the
products was comparable (A7 and A13; Table S2), with $C_{10}H_{10}O_5$ as the most abundant product (Figs. 1d and S6a), but in
VL+SN, there was a significant amount of a VL dimer ($C_{15}H_{12}O_8$; No. 9, Table S3). Moreover, the nitrogen-containing
compounds were distinct. Aside from the potential imidazole derivative ($C_5H_5N_3O_2$; No. 10, Table S3), $C_8H_9NO_3$ was also
observed from VL+AN but only under $N_2$-saturated conditions (Fig. 1b), probably due to further oxidation by secondary
oxidants from $^3VL^*$. The product analysis suggests the participation of ammonium in the aqueous-phase reactions.
Ammonium salts are an important constituent of atmospheric aerosols particles (Jimenez et al., 2009), and reactions between
dicarbonyls (e.g., glyoxal) and ammonia or primary amines have been demonstrated to form BrC (De Haan et al., 2009, 2011;
Nozière et al., 2009; Shapiro et al., 2009; Lee et al., 2013; Powelson et al., 2014; Gen et al., 2018; Mabato et al., 2019).
Relative to VL+AN, the products from VL+SN had higher O:C ratios (e.g., $C_7H_4N_2O_7$; No. 11, Table S3), $OS_c$, and $\langle OS_c \rangle$
values (Table S2).

### 3.4 Oxidation of guaiacol by photosensitized reactions of vanillin and photolysis of nitrate


The oxidation of phenols by $^3C^*$ has been mainly studied using non-phenolic aromatic carbonyls (Anastasio et al., 1996;
Smith et al., 2014, 2015; Yu et al., 2014; Chen et al., 2020) and aromatic ketones (Canonica et al., 2000) as triplet precursors.
Recently, $^3VL^*$ have also been shown to oxidize syringol (Smith et al., 2016), a non-carbonyl phenol, although the reaction
products remain unknown. In this section, we discussed the photo-oxidation of guaiacol (GUA), a lignocellulosic BB
pollutant (Kroflič et al., 2015) that is also a non-carbonyl phenol, in the presence of VL (GUA+VL) or nitrate (GUA+AN).
The dark experiments did not show any substantial loss of VL or GUA (Fig. S3e). Due to its poor light absorption in the
solar range, GUA is not an effective photosensitizer (Smith et al., 2014; Yu et al., 2014). Accordingly, the direct GUA
photodegradation resulted in minimal decay, which plateaued after ~3 hours. However, in the presence of VL or nitrate, the
GUA decay was faster by 2.2 (GUA+VL) and 1.2 (GUA+AN) times, respectively, than for direct GUA photodegradation.
This enhanced GUA decay rate may be due to the following main reactions: oxidation of GUA by $^3VL^*$ (or the secondary
oxidants it generates upon reaction with $O_2$), oxidation by $^{\bullet}OH$ produced from nitrate photolysis, or nitration by $^{\bullet}NO_2$ from
nitrate photolysis. As mentioned earlier, the $^3VL^*$ chemistry appears to be more important than that of nitrate photolysis
even at 1:10 molar ratio of VL/nitrate on account of the much higher molar absorptivity of VL compared to that of nitrate
(Fig. S1) and the high VL concentration (0.1 mM) used in this study. The decay of VL in GUA+VL (A18; Table S2) was 3
times slower than that of VL* (A5; Table S2), which may be due to competition between ground-state VL and GUA for
reactions with $^3VL^*$ (or the secondary oxidants it generates upon reaction with $O_2$) or increased conversion of $^3VL^*$ back to





the ground state through the oxidation of GUA (Anastasio et al., 1996; Smith et al., 2014). The corresponding absorbance
changes for the GUA experiments (Fig. 1e) were consistent with the observed decay trends. The minimal absorbance
changes for the direct GUA photodegradation also plateaued after ~3 hours. Moreover, the difference between GUA photo-
oxidation in the presence of VL or nitrate was more evident, with the former showing much higher absorbance enhancement.
Similarly, Yang et al. (2021) also observed greater light absorption during nitrate-mediated photo-oxidation relative to direct
GUA photodegradation.
For the direct GUA photodegradation, GUA+VL, and GUA+AN, the normalized abundance of products was
calculated only for GUA+VL (2.2; Table S2), as the GUA signal from the UHPLC-qToF-MS in the positive ion mode was
weak, which may introduce large uncertainties during normalization. Nonetheless, the number of products detected from
these experiments (178, 266, and 844 for the direct GUA photodegradation, GUA+AN, and GUA+VL, respectively)
corroborates the kinetics and absorbance results. The major products (Fig. 3a) from the direct photodegradation of GUA
were $C_{14}H_{14}O_4$ (No. 19, Table S3), a typical GUA dimer, and a trimer ($C_{21}H_{20}O_6$; No. 20, Table S3) which likely originated
from photoinduced O-H bond-breaking (Berto et al., 2016). In general, higher absolute signal areas was noted for oligomers
(e.g., $C_{14}H_{14}O_4$, $C_{21}H_{20}O_6$) and hydroxylated products (e.g., $C_7H_8O_4$) in both GUA+VL and GUA+AN, similar to those
observed from GUA oxidation by triplets of 3,4-dimethoxybenzaldehyde (DMB; a non-phenolic aromatic carbonyl) or $^•$OH
(from $H_2O_2$ photolysis) (Yu et al., 2014). In contrast to the GUA aqSOA reported by Yu et al. (2014), the photo-oxidation of
GUA in this study yielded nitrated compounds (e.g., $C_9H_{14}N_2O_6$, $C_{11}H_{14}N_2O_9$) from GUA+AN and VL dimers (e.g.,
$C_{16}H_{12}O_6$) from GUA+VL. However, based on a recent work on the aqueous photo-oxidation of guaiacyl acetone (another
aromatic phenolic carbonyl) by DMB triplets, the hydroxylation and dimerization of DMB can also contribute to aqSOA
(Jiang et al., 2021). The contributions from DMB-participated reactions were only minor due to the low initial DMB
concentration (0.005 mM). Relative to GUA+AN, higher signals for dimers such as $C_{14}H_{14}O_4$ and $C_{16}H_{12}O_6$ were noted in
GUA+VL, possibly due to both GUA and ground-state VL being available as oxidizable substrates for $^3VL^*$ and the
secondary oxidants it can generate. Also, a potential GUA tetramer ($C_{28}H_{24}O_8$, No. 21, Table S3) was observed only in
GUA+VL, consistent with higher oligomer formation from the triplets-mediated photo-oxidation of phenolics relative to
$^•$OH-assisted photo-oxidation (Yu et al., 2014). In general, the products from the direct GUA photodegradation, GUA+VL,
and GUA+AN had similar $OS_c$ values (-0.5 to 0.5) (Figs. 3b-d), falling into the criterion of BBOA and SV-OOA (Kroll et
al., 2011). In this work, efficient GUA photo-oxidation was observed in the presence of VL and AN, forming aqSOA
composed of oligomers, hydroxylated products, and nitrated compounds (for GUA+AN). The higher product signals from
GUA+VL compared to GUA+AN is likely due to the availability of both GUA and ground-state VL as aqSOA precursors.

**3.5 Photo-oxidation pathways of vanillin via direct photosensitization and in the presence of nitrate**

The most probable pathways of direct photosensitized and nitrate-mediated photo-oxidation of VL were proposed (Fig. 4). In
Scheme 1 (pH 4 and pH <4 under air-saturated conditions), $^3VL^*$ and $^•$OH (from $^3VL^*$ or nitrate photolysis) can initiate H-





atom abstraction to generate phenoxy or ketyl radicals (Huang et al., 2018; Vione et al., 2019). At pH 4, ring-opening
products (Fig. S5) from fragmentation in both VL* and VL+AN may have reacted with VL or dissolved ammonia to
generate $C_{10}H_{10}O_5$ (Pang et al., 2019b) and a potential imidazole derivative ($C_5H_5N_3O_2$), respectively. Moreover, nitrate
photolysis products promoted functionalization and nitration (e.g., $C_{16}H_{10}N_2O_9$). At pH <4, the reactivity of $^3$VL* increased
as suggested by the abundance of oligomers (e.g., $C_{16}H_{14}O_6$) and increased normalized abundance of N-containing
compounds.

In Scheme 2 (pH 4, IPA or NaBC, under air-saturated conditions), additional radicals generated ($^{\bullet}HO_2$ and $CO_3^{\bullet-}$)
likely promoted more reactions. An abundant dimer ($C_{16}H_{14}O_6$) and higher oligomers (e.g., tetramers, $C_{31}H_{22}O_{12}$) were
identified in VL*+IPA, possibly due to $^{\bullet}HO_2$-initiated reactions, while a functionalized monomer ($C_7H_4O_4$) was abundant in
VL*+NaBC. In general, nitrate enhanced both oligomerization and functionalization in VL+IPA or VL+NaBC. In
VL+AN+IPA, $C_{15}H_{12}O_8$ likely originated from $C_{16}H_{14}O_6$ via demethylation and multiple hydroxylations. In VL+AN+NaBC,
$C_8H_6O_4$ was possibly generated via H-atom abstraction from $-OCH_3$ by $^{\bullet}OH$, and further addition with $O_2$ is energy
barrierless (Priya and Lakshmipathi, 2017; Sun et al., 2019), generating a hydroperoxide ($-OCH_2OOH$) that readily
decompose to form $-OCH_2O^{\bullet}$ and $^{\bullet}OH$ (Yaremenko et al., 2016). $-OCH2O^{\bullet}$ is finally transformed into $-OCHO$ with the
elimination of $HO_2$ in the presence of $O_2$ (Sun et al., 2019). Moreover, the abundance of $C_{15}H_{12}O_8$ was higher in
VL+AN+NaBC than in VL*+NaBC.
**4 Conclusions and atmospheric implications**
This study shows that the photo-oxidation of VL via its direct photosensitized reactions and in the presence of nitrate can
generate aqSOA composed of oligomers, functionalized monomers, oxygenated ring-opening products, and nitrated
compounds (for nitrate-mediated reactions). The characterization of products presented in this work complements earlier
studies (e.g., Smith et al. 2014, 2015, 2016) that mainly discussed the kinetics and aqSOA yield of triplet-driven oxidation of
phenols. Although nitrate did not substantially affect the VL decay rates, likely due to much higher molar absorptivity of VL
than nitrate and high VL concentration used in this work, the presence of nitrate promoted functionalization and nitration,
indicating the significance of nitrate photolysis in this aqSOA formation pathway. While nitration can be an important
process for producing light-absorbing organics or BrC (Jacobson, 1999; Kahnt et al., 2013; Mohr et al., 2013; Laskin et al.,
2015; Teich et al., 2017; Li et al., 2020), its effect on triplet-generating aromatics has not yet been examined in detail. On a
related note, a recent work (Ma et al., 2021) mimicking phenol oxidation by DMB (a non-phenolic aromatic carbonyl)
triplets in more concentrated conditions in aerosol liquid water (ALW) showed that significantly higher AN concentration
(0.5 M) increased the photodegradation rate constant for guaiacyl acetone (an aromatic phenolic carbonyl with high Henry's
law constant, $1.2 \times 10^6$ M atm$^{-1}$; McFall et al., 2020) by >20 times which was ascribed to $^{\bullet}OH$ formation from nitrate
photolysis (Brezonik and Fulkerson-Brekken, 1998; Chu and Anastasio, 2003). The same study also estimated that reactions
of phenols with high Henry's law constants ($10^6$ to $10^9$ M atm$^{-1}$) can be important for SOA formation in ALW, with





mechanisms mainly governed by $^3$C* and $^1$O$_2$ (Ma et al., 2021). Likewise, Zhou et al. (2019) reported that the direct
photodegradation of acetosyringone was faster by about 6 times in the presence of 2 M NaClO$_4$. However, the opposite was
noted for the photodegradation of VL in sodium sulfate or sodium nitrate, which would occur slower (2 times slower in 0.5
M sodium sulfate and ~10 times slower in 0.124 M sodium nitrate) in ALW relative to dilute aqueous phase in clouds. These
suggest that the nature of inorganic ions may have an essential role in the photodegradation of organic compounds in the
aqueous phase (Loisel et al., 2021).

Furthermore, a potential imidazole derivative observed from the VL+AN (A7; Table S2) experiment suggests that
ammonium may participate in aqSOA formation from the photo-oxidation of phenolic aromatic carbonyls. Also, the
oligomers from these reaction systems may be rather recalcitrant to fragmentation based on their high abundance, even at the
longest irradiation time used in this study. Nonetheless, the increasing concentration of small organic acids over time implies
that fragmentation becomes important at extended irradiation times. Aromatic carbonyls and nitrophenols have been reported
to be the most important classes of BrC in cloud water heavily affected by biomass burning in the North China Plain
(Desyaterik et al., 2013). Correspondingly, the most abundant products from our reaction systems (pH 4, air-saturated
solutions) are mainly potential BrC chromophores. These suggest that aqSOA generated in cloud/fog water from the
oxidation of biomass burning aerosols via direct photosensitized reactions and nitrate photolysis products can impact aerosol
optical properties and radiative forcing, particularly for areas where biomass burning is intensive.

Our results indicate that the photo-oxidation of VL is influenced by secondary oxidants from VL triplets, pH, the
presence of VOCs and inorganic anions, and reactants concentration and molar ratios. Compared to N$_2$-saturated conditions,
more efficient VL photo-oxidation was observed under air-saturated conditions (O$_2$ is present), which can be attributed to the
generation of secondary oxidants (e.g., $^1$O$_2$, O$_2$$^{•-}$/$^•$HO$_2$, $^•$OH) from $^3$VL*. Further enhancement of VL photo-oxidation under
air-saturated conditions in the presence of nitrate indicates synergistic effects between secondary oxidants from VL triplets
and nitrate photolysis products. Additionally, the formation of oligomers from VL photo-oxidation was observed to be
promoted at low pH (<4) or in the presence of IPA/NaBC, which likely generated additional radicals such as $^•$HO$_2$ and CO$_3$$^{•-}$.
As aerosols comprise more complex mixtures of organic and inorganic compounds, it is worthwhile to explore the impacts
of other potential aerosol constituents on aqSOA formation and photo-oxidation studies. This can also be beneficial in
understanding the interplay among different reaction mechanisms during photo-oxidation. Low VL concentration favored
functionalization, while oligomerization prevailed at high VL concentration, consistent with past works (Li et al., 2014;
Slikboer et al., 2015; Ye et al., 2019). Hydroxylation was observed to be important for equal molar ratios of VL and nitrate,
likely due to VL competing with nitrite for $^•$OH. The oxidation of guaiacol, a non-carbonyl phenol, by photosensitized
reactions of vanillin was also shown to be more efficient than that by nitrate photolysis products.

In this study, we investigated reactions of VL and nitrate at concentrations in cloud/fog water. The concentrations of
VL and nitrate can be significantly higher in aqueous aerosol particles. As a major component of aerosols, the concentration
of nitrate can be as high as sulfate (Huang et al., 2014). More studies should then explore the direct photosensitized
oxidation and nitrate-mediated photo-oxidation of other biomass burning-derived phenolic aromatic carbonyls, particularly



those with high molar absorption coefficients and can generate $^3C^*$. The influences of reaction conditions should also be investigated to better understand the oxidation pathways. Considering that biomass burning emissions are expected to increase continuously, further studies on these aqSOA formation pathways are strongly suggested.

*Data availability.*

The data used in this publication are available to the community and can be accessed by request to the corresponding author.

*Author contributions.*

BRG designed and conducted the experiments; YL provided assistance in measurements and helped to analyze experimental data; YJ provided assistance in measurements; BRG, YL, and CKC wrote the paper. All co-authors contributed to the discussion of the manuscript.

*Competing interests.*

The authors declare that they have no conflict of interest.

*Acknowledgments.*

This work was financially supported by the National Natural Science Foundation of China (41875142 and 42075100). D.D.H. acknowledges support from the National Natural Science Foundation of China (21806108). X.L. acknowledges support from the Local Innovative and Research Teams Project of Guangdong Pearl River Talents Program (2019BT02Z546). T.N. acknowledges support from the Hong Kong Research Grants Council (21304919) and City University of Hong Kong (9610409). C.H.L. acknowledges support from the City University of Hong Kong (9610458 and 7005576).

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





**Table 1**. List of reactions involving reactive species relevant to this study.

| No. | Reactions | References |
|---|---|---|
| 1 | $NO_3^- + h\nu \rightarrow {}^{\bullet}NO_2 + O^-$; $\phi = 0.01$ | Vione et al., 2006; Scharko et al., 2014 |
| 2 | $O^- + H_3O^+ \leftrightarrow {}^{\bullet}OH + H_2O$ | |
| 3 | $NO_3^- + h\nu \rightarrow NO_2^- + O(^3P)$; $\phi = 0.001$ | |
| 4 | $NO_2^- + {}^{\bullet}OH \rightarrow {}^{\bullet}NO_2 + OH^-$ ($k = 1.0 \times 10^{10}\,M^{-1}\,s^{-1}$) | Mack and Bolton, 1999; Pang et al., 2019a |
| 5 | $O_2^{\bullet-} + NO_2^- + 2H^+ \rightarrow {}^{\bullet}NO_2 + H_2O_2$ | Vione et al., 2001; Pang et al., 2019a |
| 6 | $NO_2^- + h\nu \rightarrow {}^{\bullet}NO + O^-$; $\phi_{OH,300} = 6.7(\pm0.9)\%$ | Fischer and Warneck, 1996; Mack and Bolton, 1999; Pang et al., 2019a |
| 7 | ${}^{\bullet}NO + O_2 \leftrightarrow {}^{\bullet}ONOO$ | Goldstein and Czapski, 1995a; Pang et al., 2019a |
| 8 | ${}^{\bullet}ONOO + {}^{\bullet}NO \rightarrow ONOONO$ | |
| 9 | $ONOONO \rightarrow 2{}^{\bullet}NO_2$ | |
| 10 | $NO_3^- + h\nu \rightarrow {}^{\bullet}NO_2 + OH$ (reactions 1 & 2) $\rightarrow HOONO \underset{hv}{\overset{hv}{\rightleftharpoons}} {}^{\bullet}NO + {}^{\bullet}HO_2$ (p$K_a = 6.8$) | Goldstein et al., 2005; Vione et al., 2005; Sturzbecher-Höhne et al., 2009; Abida et al., 2011; Wang et al., 2021 |
| 11 | ${}^{\bullet}HO_2 \rightleftharpoons H^+ + O_2^{\bullet-} \xrightarrow{NO_2} OONO_2^- \xrightarrow{H_2O} O_2 + NO_2^- \xleftarrow{+H^+} HONO$ (p$K_a = 4.8$) (p$K_a = 3.2$) | Lammel et al., 1990; Goldstein et al., 1998; Wang et al., 2021 |
| 12 | $O_2^{\bullet-} + NO \rightleftharpoons OONO^-$ $\xrightarrow{HOONO} NO_2^- + HOONO_2 \rightleftharpoons OONO_2^- + H^+$ (p$K_a = 5.9$) $\rightarrow NO_3^-$ | Goldstein and Czapski, 1995b; Wang et al., 2021 |
| 13 | $HNO_2 + h\nu \rightarrow {}^{\bullet}NO + OH$; $\phi_{OH,300} = 36.2(\pm4.7)\%$ | Fischer and Warneck, 1996; Kim et al., 2014; Pang et al., 2019a |
| 14 | $HOONO \rightarrow {}^{\bullet}NO_2 + {}^{\bullet}OH$ ($k = 0.35 \pm 0.03\,s^{-1}$) | Goldstein et al., 2005; Pang et al., 2019a |
| 15 | $HNO_2 + {}^{\bullet}OH \rightarrow {}^{\bullet}NO_2 + H_2O$ ($k = 2.6 \times 10^9\,M^{-1}\,s^{-1}$) | Kim et al., 2014; Pang et al., 2019a |
| 16 | $(CH_3)_2CHOH + {}^{\bullet}OH \rightarrow (CH_3)_2COH^{\bullet} + H_2O$ | Warneck and Wurzinger, 1988; Pang et al., 2019a |
| 17 | $(CH_3)_2COH^{\bullet} + O_2 \rightarrow (CH_3)_2CO + {}^{\bullet}HO_2$ | |
| 18 | ${}^{\bullet}OH + HCO_3^- \rightarrow CO_3^{\bullet-} + H_2O$ ($k = 8.5 \times 10^6\,M^{-1}\,s^{-1}$) | Wojnárovits, 2020 |
| 19 | ${}^{\bullet}OH + CO_3^{2-} \rightarrow CO_3^{\bullet-} + OH^-$ ($k = 3.9 \times 10^8\,M^{-1}\,s^{-1}$) | |
| 20 | ${}^3C^* + HCO_3^- \rightarrow CO_3^{\bullet-} + H^+ + C^{\bullet-}$ ($k = 10^6\text{-}10^7\,M^{-1}\,s^{-1}$; $^3C^*$: triplet aromatic ketones) | Canonica et al., 2005 |
| 21 | ${}^3C^* + CO_3^{2-} \rightarrow CO_3^{\bullet-} + C^{\bullet-}$ ($k = 10^6\text{-}10^7\,M^{-1}\,s^{-1}$; $^3C^*$: triplet aromatic ketones) | |





**Figure 1.** Reconstructed mass spectra of assigned peaks from (a) VL* pH 4 (N₂-saturated; A6), (b) VL+AN pH 4 (N₂-saturated; A8), (c) VL* pH 4 (air-saturated; A5), (d) VL+AN pH 4 (air-saturated; A7), (e) VL* pH 3 (air-saturated; A3), (f)



VL+AN pH 3 (air-saturated; A4), (g) VL* pH 2.5 (air-saturated; A1), and (h) VL+AN pH 2.5 (air-saturated; A2) after 6 h
of simulated sunlight irradiation. The normalized abundance of products was calculated from the ratio of the peak area of the
product to that of VL (Eq. 2). The 50 most abundant products contributed more than half of the total normalized abundance
of products, and they were identified as monomers (blue), dimers (green), trimers (red), and tetramers (orange). Grey peaks
denote peaks with low abundance or unassigned formula. Examples of high-intensity peaks were labeled with the
corresponding neutral formulas. Note the different scales on the y-axes.



**Figure 2.** (a-d) Increase in light absorption under different experimental conditions for direct photosensitized oxidation of VL (VL*) and nitrate-mediated VL photo-oxidation (VL+AN): (a) Effect of secondary oxidants from VL triplets on VL* and VL+AN at pH 4 under $N_2$- (A6, A8) and air-saturated (A5, A7) conditions. (b) Effect of pH on VL* and VL+AN at pH 2.5 (A1, A2), 3 (A3, A4), and 4 (A5, A7) under air-saturated conditions. (c) Effect of VOCs and inorganic anions: IPA (A9) and NaBC (A10) on VL* at pH 4 under air-saturated conditions. (d) Effect of VOCs and inorganic anions: IPA (A11) and NaBC (A12) on VL+AN at pH 4 under air-saturated conditions. (e) Increase in light absorption during direct GUA photodegradation (A17) and photo-oxidation of GUA in the presence of VL (GUA+VL; A18) or nitrate (GUA+AN; A19) at pH 4 under air-saturated conditions after 6 h of simulated sunlight irradiation. Error bars represent 1 standard deviation.

1152

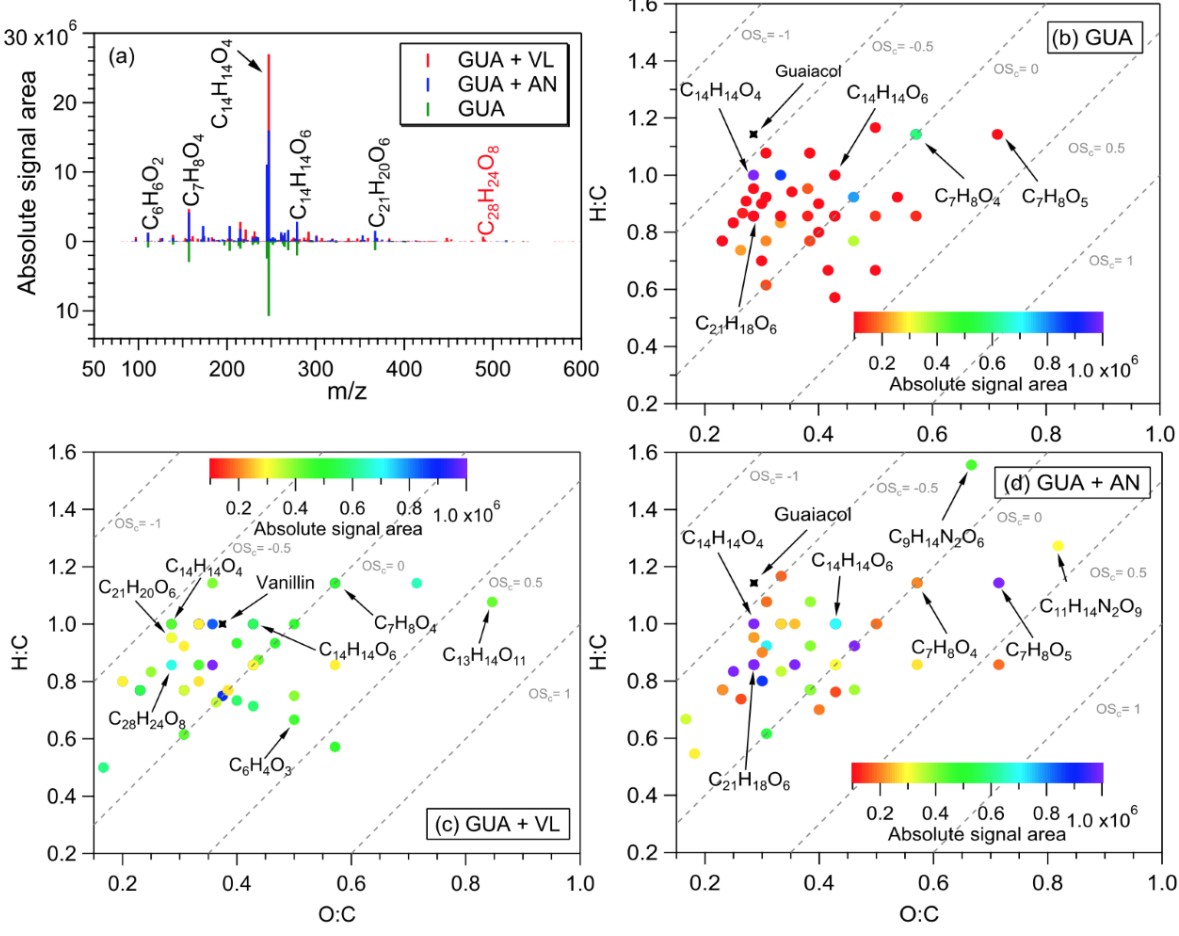

1153

**Figure 3.** (a) Reconstructed mass spectra of assigned peaks from the direct GUA photodegradation (A17) and photo-oxidation of GUA in the presence of VL (GUA+VL; A18) or nitrate (GUA+AN; A19) at pH 4 under air-saturated conditions after 6 h of simulated sunlight irradiation. The y-axis is the absolute signal area of the products. Examples of high-intensity peaks were labeled with the corresponding neutral formulas. (b-d) van Krevelen diagrams of the 50 most abundant products from the (b) direct photodegradation of GUA (A17), (c) GUA+VL (A18), and (d) GUA+AN (A19) at pH 4 under air-saturated conditions after 6 h of simulated sunlight irradiation. The color bar denotes the absolute signal area. The grey dashed lines indicate the carbon oxidation state values (e.g., $OS_c$ =-1, 0, and 1).






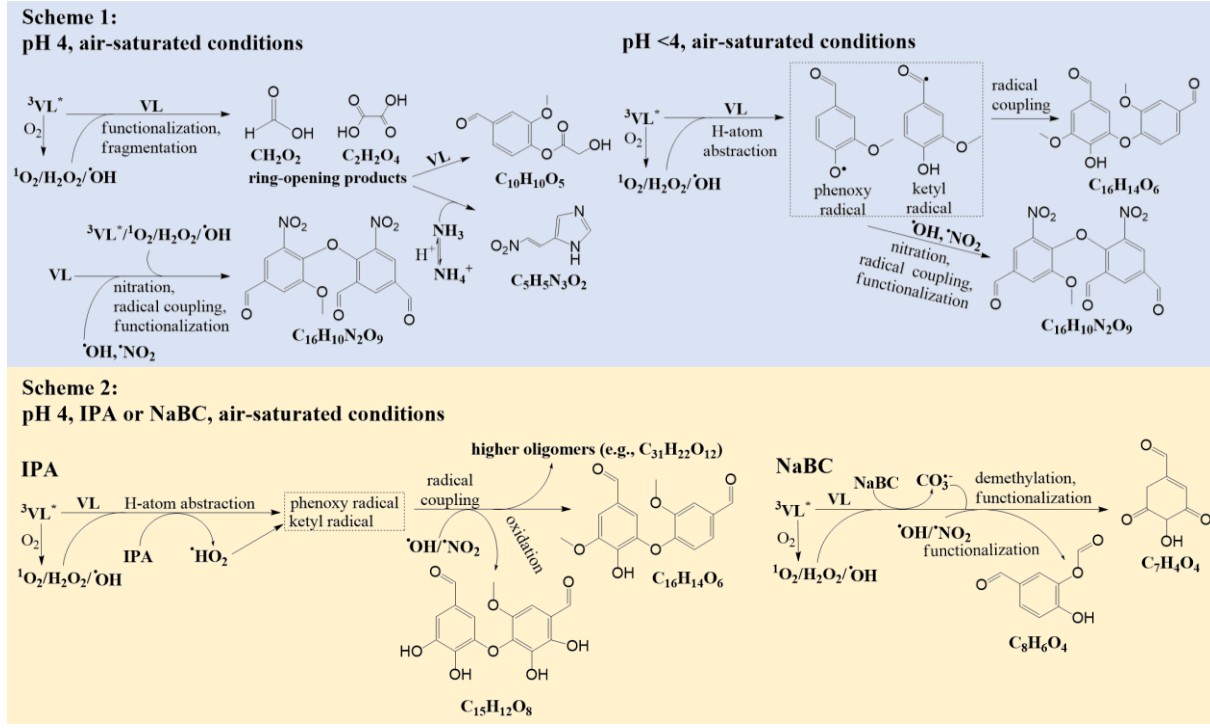

**Figure 4.** Potential photo-oxidation pathways of VL via direct photosensitized reactions and in the presence of nitrate to
illustrate the effects of secondary oxidants from VL triplets, pH, and the presence of VOCs (IPA) and inorganic anions
(NaBC). Product structures were proposed based on the molecular formulas, DBE values, and MS/MS fragmentation
patterns. The molecular formulas presented were the most abundant products or products with a significant increase in
normalized abundance.