# Peer review of "Aqueous SOA formation from the direct photosensitized oxidation of vanillin in the absence and presence of ammonium nitrate"

_Atmospheric Chemistry and Physics, 2021_

## Author Comment (AC1)

Author Response for "Aqueous SOA formation from the photo-oxidation of vanillin: Direct photosensitized reactions and nitrate-mediated reactions" by Mabato et al.

We thank the reviewer for the thorough review and many constructive comments that helped improve the manuscript. Our point-by-point responses are below (changes to the original manuscript text and supporting information are in red, moved content in double-line strikethrough, and removed content in strikethrough). Please note that the line numbers in the responses refer to our revised manuscript with tracked changes. Also, please note that because we restructured the manuscript, the numbering of some figures and tables in the revised manuscript is different from those in the original manuscript.

**Reviewer 1**

**Overview**

The authors examined the aqueous photodegradation of vanillin (VL), a carbonyl-containing phenol emitted from biomass burning, and accompanying formation of aqueous SOA (aqSOA). They then measured the composition of the aqSOA using high-resolution mass spectrometry and UV/Vis absorption. They also determined the impact of purging solutions with N2 (to remove dissolved oxygen) as well as the addition of ammonium nitrate (a photochemical source of hydroxyl radical, OH) and/or one of two hydroxyl radical scavengers (isopropyl alcohol or bicarbonate). They attempt to explain their results qualitatively based on a few dozen reactions, but there is little experimental attempt to text the mechanisms.

Vanillin has been studied in several past works, but this paper adds new information on the composition of the resulting aqSOA. The purging with N2 is novel, but the interpretation of the results is not clear, and I disagree that these experiments show that secondary oxidants dominate VL loss. Unfortunately, the nitrate concentration added was too low to impact kinetics (because VL direct photodegradation is so fast), but it's interesting that it impacted the products formed. Finally, the authors seem compelled to try to mechanistically explain most of their results, but their explanations are very speculative and should be significantly cut. There are a several other major and minor issues, as described below.

Response: In this study, we aimed to investigate the photo-oxidation of VL at atmospherically relevant cloud and fog conditions. As mentioned in lines 112–116, 'Although the concentration of VL in cloud/fog water has been estimated to be < 0.01 mM (Anastasio et al., 1996), a higher VL concentration (0.1 mM) was used in this study to guarantee sufficient signals for product identification (Vione et al., 2019). The chosen ammonium nitrate (AN) concentration (1 mM) was based on values observed in cloud and fog water (Munger et al., 1983; Collett et al., 1998; Zhang and Anastasio, 2003; Li et al., 2011; Giulianelli et al., 2014; Bianco et al., 2020).' Our study is not intended to identify the concentrations of nitrate that would affect the kinetics. This sentence has been added to the text as follows:

Line 116: It should be noted that this study is not intended to identify the concentrations of nitrate that would affect the kinetics.

**Major Comments**

1. The normalized abundance of products (line 131) is used throughout the paper as a key metric, but it's unclear if this is a robust endpoint, in part because its uncertainty is never discussed. (a) Based on the major products that have been identified (both via MS and IC), what is the likely range of ionization efficiencies (IEs) of the products and how much uncertainty does this introduce in the product abundance measure? (b) There is additional uncertainty in the quantification of VL, which is described on line 134 as semi-quantitative. (c) Altogether, what is the relative uncertainty in P from day to day and experiment to experiment? (d) This is an issue because there are several times when the normalized product abundance results are inconsistent with other, seemingly more quantitative metrics. For example, in section 3.1.3., the presence of OH scavengers had no significant effect upon VL decay or aqSOA light absorbance, but there were differences in the normalized abundance of products. Given the uncertainty in IE and other aspects of the product measure, I would be wary of attributing much significance to the normalized abundance of products as an endpoint when it's inconsistent with the more quantitative measures.

Response: The normalized abundance of products in this study is a semi-quantitative analysis intended to provide an overview of how the signal intensities (as normalized in Eq. 2) changed under different experimental conditions, but not to quantify the absolute concentration of products. Even if relative abundance (product peaks are normalized to the highest peak), which has been widely utilized (e.g., Lee et al., 2014, https://pubs.acs.org/doi/10.1021/es502515r; Romonosky et al., 2017, https://pubs.acs.org/doi/10.1021/acs.jpca.6b10900; Fleming et al., 2018, https://doi.org/10.5194/acp-18-2461-2018; Klodt et al., 2019, https://pubs.acs.org/doi/pdf/10.1021/acsearthspacechem.9b00222) in the literature is used instead of normalized abundance in our analysis, the major products detected as well as the conclusions of this study will remain the same. Regardless, we agree with the reviewer that we should emphasize the potential uncertainties in the normalized abundance of products. Detailed responses to the relevant sub-questions are as follows:

(a) Ionization efficiencies can indeed vary between different compounds. Unfortunately, the availability of measured relative ionization efficiencies (RIE) for different compounds is limited. We are not in a position to provide this information. The reviewer is correct that ESI ionization is not ideal for product quantification. Nevertheless, Nguyen et al. (2013) (https://doi.org/10.1039/C2AY25682G) found a positive correlation between ESI signal and "adjusted mass" (= molecular mass × H: C). Based on that study, the uncertainty would be a factor of 2 – 4 if only the "adjusted mass" is considered, and further complications of matrix effect and polarity are disregarded. However, what we compared is not the absolute concentrations (or contributions) of the products observed. The comparison was based on how the signal intensities (as normalized in Eq. 2) changed under different experimental conditions. We compared the

responses of the same products (or at least the same class of products) as conditions varied, and their ionization efficiency might not be very different within the same class, according to the "adjusted mass" concept by Nguyen et al. (2013). We have revised Sect. 2.2 to highlight the inherent uncertainties for this metric due to ionization efficiencies which can vary for different compounds as follows:

Lines 146–167: Comparisons of peak abundance in mass spectrometry have been used in many recent studies (e.g., Lee et al., 2014; Romonosky et al., 2017; Wang et al., 2017; Fleming et al., 2018; Song et al., 2018; Klodt et al., 2019; Ning et al., 2019) to show the relative importance of different types of compounds (Wang et al., 2021). However, ionization efficiency may greatly vary for different classes of compounds (Kebarle, 2000; Schmidt et al., 2006; Leito et al., 2008; Perry et al., 2008; Kruve et al., 2014) and so uncertainties may arise from comparisons of peak areas among compounds. In this work, we assumed equal ionization efficiency of different compounds, which is commonly used to estimate O:C ratios of SOA (Bateman et al., 2012; Lin et al., 2012; De Haan et al., 2019), to calculate their normalized abundance. The normalized abundance of a product, [P] (unitless), was calculated as follows:

$$[P] = \frac{A_{P,t}}{A_{VL,t}} \cdot \frac{[VL]_t}{[VL]_0} \qquad \text{(Eq. 2)}$$

where $A_{P,t}$ and $A_{VL,t}$ are the extracted ion chromatogram (EIC)  peak areas of the product P and VL from UHPLC-qToF-MS analyses at time $t$, respectively; $[VL]_t$ and $[VL]_0$ are the VL concentrations (μM) determined using UHPLC at time $t$ and 0, respectively. Here, we relied on the direct quantification  of [VL] using UHPLC (see Fig. S2 for VL calibration curve)  It should be noted that the normalized abundance of products in this study is a semi-quantitative analysis intended to provide an overview of how the signal intensities changed under different experimental conditions but not to quantify the absolute concentration of products. Moreover, the major products detected in this study are probably those with high concentration or high ionization efficiency in the positive ESI mode. The use of relative abundance (product peaks are normalized to the highest peak) (e.g., Lee et al., 2014; Romonosky et al., 2017; Fleming et al., 2018; Klodt et al., 2019) would yield the same major products reported. Typical fragmentation behavior observed in MS/MS spectra for individual functional groups from Holčapek et al. (2010) are outlined in Table S1.

(b) The semi-quantification here refers to the normalized abundance of products, not the VL concentration. [VL] was directly quantified using UHPLC (see Fig. S2 for VL calibration curve). The reported [VL] are the average of results from triplicate experiments and the uncertainties from which and those from the MS signal intensities were propagated (now added in Table 2, formerly S2). This has been clarified in the text as follows:

Line 157: Here, we relied on the direct quantification  of [VL] using UHPLC (see Fig. S2 for VL calibration curve) .

(c) Given the same instrumental settings, the variations caused by the instrumental fluctuations would be smaller than the effects caused by the difference in ionization efficiency among different species. If there is any, it would be taken into account by the normalization in Eq. 2, which is why the normalized signal intensities were used instead of absolute signal intensities. For reference, relative uncertainties for MS signal peak areas of VL at the same concentration measured from different experiments range from 0.14 to 0.25. Moreover, the propagated uncertainties from the MS signal intensities and [VL] are now shown in Table 2 (formerly S2).

(d) For $^{\bullet}$OH scavengers experiments, the insignificant changes for VL decay and absorbance enhancement might not be reflected in the products observed using UHPLC-qToF-MS in positive ESI mode. It is possible that the products observed might not have contributed significantly to all products formed and may not be the primary contributors to the absorbance enhancement. The absorbance enhancement may not necessarily correlate directly with the products detected. However, as mentioned in our response to major comment #4, we decided to omit the section for $^{\bullet}$OH scavengers based on the likely minor contribution of $^{\bullet}$OH to VL photo-oxidation in this study.

2. Throughout the manuscript, the low decay rate of VL* under N2 is taken to mean that the triplet state of VL isn't involved in VL decay and that secondary, O2-dependent, oxidants are responsible for VL decay. However, the N2-purging control experiment result is ambiguous, since secondary steps in VL decay via triplets might require oxygen to proceed. For example, a major fate of the ketyl radical formed by the 3VL* + VL reaction is to add oxygen. In the absence of oxygen, the ketyl radical will still form, but it's forward path (O2 addition) is blocked, possibly leading to eventual return to the reactants (and little apparent VL decay). So N2 purging is likely to not only remove secondary oxidants, but also to interfere with subsequent steps in the 3VL* - VL reactions. Thus the oft-stated conclusion that secondary oxidants from 3VL* are responsible for VL decay is not correct (e.g., on line 184). Without knowing the impact of O2 on the reaction intermediates in the triplet reaction, it is impossible to know what the N2-purging result means.

Another strike against the "secondary oxidants" theory is that the proposed secondary oxidants are unlikely to be important for VL decay. For example, the 1O2* + VL reaction is slow under the pH conditions here (where there is negligible phenolate). In air-saturated solutions, the 1O2* and 3VL* concentrations should be roughly equal (see the McNeil and Canonica review in ESPI), but at pH 4 (and below) the rate constants for phenols with 3C* are much faster than the 1O2* values. The bottom line is the 1O2* is unlikely to be important. Similarly, HO2/O2- was proposed as an oxidant for phenols, but these are very weak oxidants that react slowly with phenols. Finally, OH is apparently unimportant as well, based the OH scavengers having no significant impact on VL decay; however, it is possible that most of the IPA or bicarbonate was

purged from the sample prior to illumination (as discussed below). Regardless, photolysis of H2O2 (formed from the 3VL* + VL reaction) will be slow, giving little OH.

Response: Thank you for this thorough and important analysis. We apologize for the confusion related to the role of VL triplets. The reviewer is correct that VL triplets are indeed important for VL decay and that the secondary oxidants generated in the presence of $O_2$ likely have only minor roles in the photo-oxidation of VL in this study. We also agree that without a detailed investigation of the effect of $O_2$ on the reactive intermediates, it is difficult to interpret the mechanism of the $N_2$ experiments. In principle, initial oxidation by triplets can proceed without $O_2$, forming phenoxy (which is in resonance with a carbon-centered cyclohexadienyl radical that has a longer lifetime) and ketyl radicals (Neumann et al., 1986a, 1986b; Anastasio et al., 1996). The coupling of phenoxy radicals or phenoxy and cyclohexadienyl radicals can form oligomers, as observed for both $N_2$- and air-saturated experiments. However, the little decay of VL under $N_2$-saturated condition indicates that these radicals probably predominantly decayed via back-hydrogen transfer to regenerate VL (Lathioor et al., 1999). A possible explanation for this is the involvement of $O_2$ in the secondary steps of VL decay, likely concerning the fate of the ketyl radical, as the reviewer pointed out. We have amended the discussions to include these possibilities and to emphasize the importance of VL triplets as follows:

Line 187:  VL photo-oxidation under $N_2$ and air-saturated conditions

Lines 188–220:  The photo-oxidation of VL  under both $N_2$- and air--saturated conditions (Fig. S3a) were carried out at pH 4, which is representative of moderately acidic aerosol and cloud pH values (Pye et al., 2020). No significant VL loss was observed for dark experiments. The oxidation of ground-state VL by $^3VL^*$ via H-atom abstraction or electron transfer can form phenoxy (which is in resonance with a carbon-centered cyclohexadienyl radical that has a longer lifetime) and ketyl radicals (Neumann et al., 1986a, 1986b; Anastasio et al., 1996). The coupling of phenoxy radicals or phenoxy and cyclohexadienyl radicals can form oligomers as observed for both $N_2$- and air-saturated experiments (see discussions later). However, the little decay of VL under $N_2$-saturated condition indicates that these radicals probably predominantly decayed via back-hydrogen transfer to regenerate VL (Lathioor et al., 1999). A possible explanation for this is the involvement of $O_2$ in the secondary steps of VL decay. For instance, a major fate of the ketyl radical is reaction with $O_2$ (Anastasio et al., 1996). In the absence of $O_2$, radical formation occurs, but the forward reaction of ketyl radical and $O_2$ is blocked, leading to the regeneration of VL as suggested by the minimal VL decay. Aside from potential inhibition of secondary oxidants generation (Chen et al., 2020), $N_2$ purging may have also hindered the secondary steps for VL decay.

 Contrastingly, the VL\* decay rate constant under air-saturated conditions was 4 times higher. As mentioned earlier, secondary oxidants ($^1O_2$, $O_2^{\bullet-}$/$^{\bullet}HO_2$, $^{\bullet}OH$) can be generated from $^3$VL\* when $O_2$ is present (e.g., under air-saturated conditions). However, the photo-oxidation of VL in this study is likely mainly governed by $^3$VL\* and that these secondary oxidants have only minor participation.  $^1O_2$ is also a potential oxidant for phenols (Herrmann et al., 2010; Minella et al., 2011; Smith et al., 2014), but $^1O_2$ reacts much faster (by ~60 times) with phenolate ions compared to neutral phenols (Tratnyek and Hoigne, 1991; Canonica et al., 1995; McNally et al., 2005). Under the pH values (pH 2.5 to 4) considered in this study, the amount of phenolate ion is negligible, so the reaction between VL and $^1O_2$ should be slow. Interestingly, however, $^1O_2$ has been shown to be important in the photo-oxidation of 4-ethylguaiacol ($pK_a$ = 10.3) by $^3$C\* of 3,4-dimethoxybenzaldehyde (solution with pH of ~3) (Chen et al., 2020). Furthermore, while the irradiation of other phenolic compounds can produce $H_2O_2$, a precursor for $^{\bullet}OH$ (Anastasio et al., 1996), the amount of $H_2O_2$ is small. Based on this, only trace amounts of $H_2O_2$ were likely generated from VL\* (Li et al., 2014) under-air saturated conditions, suggesting that contribution from $^{\bullet}OH$ was minor. Overall, these suggest that VL photo-oxidation in this study is driven by $^3$VL\*. Further study on the impact of $O_2$ on the reactive intermediates involved is required to understand the exact mechanisms occurring under air-saturated conditions. Nonetheless, the VL\* decay trends clearly indicate that $O_2$ is important for efficient VL photo-oxidation .

Revisions made elsewhere in the text:

Line 23: The effects of oxygen ($O_2$)

Line 25: Our findings show that  $O_2$ plays an essential role in VL photo-oxidation.

Line 91: The influences of $O_2$

Line 184: In this work, the direct photosensitized oxidation of VL (VL only experiments)

Lines 237–240: Although we have no quantification of the oxidants in our reaction systems as it is outside the scope of this study, these observations clearly substantiate that  photosensitized oxidation of VL and nitrate-mediated VL photo-oxidation are more efficient in the presence of $O_2$.

Lines 247–254: Oligomers,  functionalized monomers (e.g., demethylated VL; Fig. S4), and nitrogencontaining compounds (e.g., $C_{16}H_{10}N_2O_9$; No. 3, Table S2) (for VL+AN) had higher normalized abundance  under air-saturated conditions  (Figs. 1c-d), likely due to efficient  $^3VL^*$-initiated oxidation and enhanced VL nitration in the presence of  $O_2$  For both VL* and VL+AN under air-saturated conditions, the most abundant product was $C_{10}H_{10}O_5$ (No. 4, Table S2), a substituted VL.

Lines 287–289: Among experiments A5 to A8 , VL+AN under air-saturated conditions (A7) had the highest normalized abundance of products and $<OS_c>$,  probably due to the combined influence of  $^3VL^*$ and enhanced VL nitration in the presence of $O_2$.

Lines 295–300: In brief, the presence of  $O_2$ increased the normalized abundance of products and promoted the formation of more oxidized aqSOA.  Compared to $N_2$-saturated condition, the higher normalized abundance of nitrogen-containing products under air-saturated condition for VL+AN (at pH 4) suggests a potential enhancement of VL nitration in the presence of $O_2$.

Lines 324–326: Overall, these trends establish that  $O_2$  is necessary for the efficient formation of light-absorbing compounds from both VL* and VL+AN.

Lines 493–495: Aside from the potential imidazole derivative ($C_5H_5N_3O_2$; No. 5, Table S2), $C_8H_9NO_3$ (No. 2, Table S2) was also observed from VL+AN but only under $N_2$-saturated conditions (Fig. 1b), probably due to further oxidation by  $^3VL^*$.

Line 512: This enhanced GUA decay rate constant may be due to the  oxidation of GUA by $^3VL^*$

Lines 520: which may be due to competition between ground-state VL and GUA for reactions with $^3VL^*$

Line 543: possibly due to both GUA and ground-state VL being available as oxidizable substrates for $^3VL^*$ .

Line 601: Our results indicate that the photo-oxidation of VL is influenced by $O_2$ , pH

Lines 602–612:  Under $N_2$-saturated conditions, the absence of $O_2$ likely hindered the secondary steps in VL decay (e.g., reaction of ketyl radical and $O_2$), regenerating VL as suggested by the minimal VL decay

saturated conditions (O₂ is present), which can be attributed to the generation of secondary oxidants (e.g., ¹O₂, O₂•⁻/•HO₂, •OH) from ³VL*. Further enhancement of VL photo-oxidation under air-saturated conditions in the presence of nitrate indicates synergistic effects between secondary oxidants from VL triplets and nitrate photolysis products. In contrast, $^3$VL*-initiated reactions proceeded rapidly under air-saturated conditions (O₂ is present) as indicated by higher VL decay rate constant and increased normalized abundance of products. For pH 4 experiments, the presence of both O₂ and nitrate resulted in the highest normalized abundance of products (including N-containing compounds) and <OS_c>, which may be due to O₂ promoting VL nitration. Nevertheless, further work is necessary to assess the effect of O₂ on the reactive intermediates involved in $^3$VL*-driven photo-oxidation and elucidate the mechanisms of VL photo-oxidation under air-saturated conditions.

3. Mechanism discussion. The authors seem compelled to try to explain all of their observations using one or more reactions, but since there is no quantitative examination of these mechanisms, they are all very speculative and mostly not useful. Worse, in some (many?) cases, the proposed mechanisms are inconsistent with some of the data. Fundamentally, without building a kinetic model of the mechanism and testing it against the observations, it is difficult to know whether the proposed reactions are important. The authors put too much emphasis on trying to mechanistically explain their observations and these explanations end up being mostly conjectures that are not grounded in data. These mechanistic speculations should be greatly reduced, especially if they are inconsistent with the kinetic or light absorption data and/or if they rely primarily upon the "normalized abundance of products" metric, which seems highly uncertain.

For example, on line 226, what trends were reinforced in the presence of nitrate? Looking at Table S2, ammonium nitrate has no effect on the kinetics, does not change the normalized product abundance at pH 2.5 or 3 (but does increase it at pH 4), and has no impact on OS(C). Later, in Fig. 2, we see that the presence of nitrate only negligibly increased the long-wavelength absorbance of the products. Overall, the bulk of the observations suggest that nitrate has a minor impact on VL decay, consistent with the fast direct photodegradation of VL.

Response: We concur with the reviewer that the current manuscript contains several speculative reaction mechanisms. As building a kinetic model of the mechanisms is beyond the scope of this study, we proposed major pathways for aqSOA formation instead (Fig. 2, formerly 4). As suggested in major comment #7, Fig. 2 has been shown for the first time when potential aqSOA formation pathways were discussed, then referred to throughout the text.

The trends in line 297 (formerly 226) pertain to nitrate enhancing the increased normalized abundance of products and formation of more oxidized aqSOA from VL photo-oxidation in the presence of O₂ (VL* and VL+AN under air-saturated conditions) at pH 4, suggesting a potential enhancement of VL nitration in the presence of O₂. This has been revised as follows:

Lines 295–300: In brief, the presence of secondary oxidants from ³VL* and O₂ increased the normalized abundance of products and promoted the formation of more oxidized aqSOA. These

 Compared to $N_2$-saturated condition, the higher normalized abundance of nitrogen-containing products under air-saturated condition for VL+AN (at pH 4) suggests a potential enhancement of VL nitration in the presence of $O_2$.

4. I am concerned that the authors purged IPA and bicarbonate from solution during each experiment since solutions were bubbled continuously. Do they have any way to know if these OH scavengers were removed before or during illumination? Similarly, guaiacol shouldn't undergo direct photochemical loss under illumination above 300 nm, so the apparent decay measured in the dark could be evaporation during purging. If the purging was slow enough, each bubble would achieve Henry's law equilibrium with the solution, which would allow you to estimate the rates of IPA and bicarbonate (lost as CO2) from the rate constant for GUA loss and the ratio of Henry's law constants for GUA and OH scavenger. For GUA, which can be measured by HPLC, the authors should report the fraction of the initial concentration (0.1 mM) that was lost after the 30 min of purging in the dark and the fraction then lost in the dark control for the illumination experiment. Then for IPA and bicarbonate, some estimate of their fraction lost during purging would be helpful. At the very least, this issue needs to be raised and addressed.

Response: The reviewer has correctly pointed out that the contribution of $^\bullet$OH to VL photo-oxidation in this study is likely minor, which is also suggested by other published literature (Anastasio et al., 1996; Li et al., 2014). We, therefore, decided to omit this section (and related sentences) and instead focus on the other findings of the paper.

5. Section 3.1.2. (a) Are VL (and GUA) decay rate constants normalized for photon fluxes? (b) Given the variability in kinetic decays, are the relative small differences in decay rate constants between pH 2.5 and 4 statistically different? (c) Do the authors have a good measure of the variability of the kinetics, e.g., the standard deviation of j(VL) based on triplicate experiments? Given that the decays are not first order, it is more difficult to discern differences in rate constants, so I would be cautious.

Response: (a) We thank the reviewer for pointing this out. The decay rate constants were initially not normalized for photon fluxes, although the reported values are the average of results from triplicate experiments. The values have been updated to the photon flux-normalized decay rate constants, and the following information were added to the text:

Line 142: The decay rate constants were normalized to the photon flux measured for each experiment through dividing $k'$ by the measured 2-nitrobenzaldehyde (2NB) decay rate constant, $j$(2NB) (see Text S6 for more details).

(b) Yes, the differences in the decay rate constants between pH 2.5 and 4 (VL*: decay rate constant at pH 2.5 is 1.6 times higher than at pH 4; VL+AN: decay rate constant at pH 2.5 is 1.4 times higher than at pH 4) are small but statistically significant ($p < 0.05$). In addition, the variability for the decay rate constant measurements among triplicate experiments for all

conditions in this study is low (the standard deviation for each condition is now added in Table 2, formerly S2). We have added this in the text as follows:

Line 331: The decay rate̶s̶ constants for both VL* and VL+AN increased as pH decreased (VL* and VL+AN at pH 2.5: 1.6̶5̶ and 1.4̶3̶ times faster than at pH 4, respectively) (Fig. S3b). These differences in decay rate constants are small but statistically significant ($p < 0.05$).

Other relevant changes in the text are as follows:

Line 510: The enhancement of GUA decay rate constant in the presence of VL is statistically significant ($p < 0.05$), while that in the presence of AN is not ($p > 0.05$).

(c) Yes, the reported decay rate constants are the average of results from triplicate experiments, and the standard deviation for each condition is now added in Table 2, formerly S2. The footnote of Table 2 has been revised as follows:

Table 2: [b]The data fitting was performed in the initial linear region. Each value is the average of results from triplicate experiments. Errors represent one standard deviation.

6. Lines 283-289. I would be surprised if deprotonation of phenols is responsible for the higher absorbance of the aqSOA at pH 4 compared to pH 3 and 2.5. For one, the pKa values of methoxy-substituted phenols are near 10, so there's no appreciable phenolate at pH 4. Nitro-substituted phenols can have much lower pKas, but absorbance of the aqSOA formed in the presence of nitrate is nearly the same as in the absence of nitrate, so it seems nitrophenols are a minor part of the light absorption. Another possibility is that different products are made at pH 2.5 compared to pH 4. Measuring the pH dependence of the aqSOA formed at pH 2.5 and 4 would allow you to determine whether the pH dependence is rooted in acid-base chemistry of the products or of the reactions.

Response: Thank you for the suggestion. As suggested, to understand the pH effect further, we measured the pH dependence of the aqSOA formed from VL* at pH 4 and 2.5. The reviewer is correct that deprotonation of phenols does not sufficiently explain the higher absorbance enhancement observed at pH 4 compared to pH 2.5. Based on the comparable pH dependence of the aqSOA formed from VL* at pH 4 and 2.5 (see figure below), the pH dependence observed is likely due to the acid-base chemistry of the reactions, probably involving $^3$VL* or the excimer of VL (Smith et al., 2016). Smith et al. (2016) reported that the direct photodegradation rate constants for 0.005 mM VL at pH ≤ 3 are nearly two times lower than at pH ≥ 5. The opposite trend observed in this study for 0.1 mM VL (VL* decay rate constant at pH 2.5 is 1.6 times higher than at pH 4) may be due to the reactivities of the protonated and neutral forms of the $^3$VL* being dependent on the VL concentration (Smith et al., 2016). Also, it has been reported that the quantum yield for direct VL photodegradation is higher at pH 5 than at pH 2 for 0.005 mM VL, but they are not statistically different for 0.03 mM VL (Smith et al., 2016). Changes on the text are as follows:

[Figure]

**Figure S10**. UV-Vis absorption spectra of VL*-derived aqSOA formed at (a) pH 4 and (b) pH 2.5 over a range of pH conditions from 1.5 to 10.5.

Lines 375–386: Higher absorbance enhancement for both VL* and VL+AN (Fig. 3b) was observed as pH increased . To determine whether the pH dependence is due to the acid-base chemistry of the products or of the reactions, we measured the pH dependence of the aqSOA formed from VL* at pH 4 and 2.5 over a range of pH conditions from 1.5 to 10.5 (Fig. S10). For both cases, the intensity of absorption at longer wavelengths significantly increased as the pH of the solutions was raised. Moreover, the comparable pH dependence of the two solutions suggest that the observed pH dependence may be attributed to the acid-base chemistry of the reactions, which may involve $^3$VL* or the excimer of VL (Smith et al., 2016), as discussed earlier. ~~When a phenolic molecule deprotonates at higher pH, an ortho- or para- electron-withdrawing group, such as a nitro or aldehyde group, can attract a portion of the negative charge towards its oxygen atoms through induced and conjugated effects, leading to the extension of chromophore from the electron-donating group (e.g., –O⁻) to the electron-withdrawing group via the aromatic ring (Carey, 2000; Williams and Fleming, 2008; Pang et al., 2019a). Hence, the delocalization of the negative charge in phenolates leads to significant redshifts (Mohr et al., 2013).~~

For reference, changes in the revised text that are relevant to lines 375–386 are shown below:

Lines 331–352: The decay rate constants for both VL* and VL+AN increased as pH decreased (VL* and VL+AN at pH 2.5: 1.6 and 1.4 times faster than at pH 4, respectively) (Fig. S3b). These differences in decay rate constants are small but statistically significant ($p < 0.05$). The p$K_a$ for the VL triplet has been reported to be 4.0 (Smith et al., 2016). As there are a greater fraction of VL triplets that are protonated at pH 2.5 (0.96) than at pH 4 (0.5), it is possible that the pH dependence of the decay rate constants observed in this study is due to $^3$VL* being more reactive in its protonated form. Smith et al. (2016) also observed a pH dependence for the direct photodegradation of VL (0.005 mM) (rate constants at pH ≤ 3 are ~two times lower than at pH ≥ 5) which they attributed to the sensitivity of the excimer of VL (i.e., the charge-transfer complex

formed between an excited state VL molecule and a separate ground state VL molecule; Birks, 1973, Turro et al., 2010) to acid-base chemistry. The opposite trend observed in this study for 0.1 mM VL may be due to the reactivities of the protonated and neutral forms of the $^3$VL* being dependent on the VL concentration (Smith et al., 2016).  It has been reported that the quantum yield for direct VL photodegradation is higher at pH 5 than at pH 2 for 0.005 mM VL, but they are not statistically different for 0.03 mM VL (Smith et al., 2016). ~~Also, increases in hydrogen ion concentration can enhance the formation of HO$_2$· and H$_2$O$_2$ and in turn, ·OH formation (Du et al., 2011). In addition to these pH influences on VL*, the dependence of N(III) (NO$_2$⁻ + HONO) speciation on solution acidity (Pang et al., 2019a) also contributed to the observed pH effects for VL+AN. At pH 3.3, half of N(III) exists as HONO (Fischer and Warneck, 1996; Pang et al., 2019a), which has a higher quantum yield for ·OH formation than that of NO$_2$⁻ in the near-UV region (Arakaki et al., 1999; Kim et al., 2014). The increased ·OH formation rates as pH decreases can lead to faster VL decay (Pang et al., 2019a). Also, NO$_2$⁻/HONO can generate ·NO$_2$ via oxidation by ·OH (Reactions 4 and 15; Table 1) (Pang et al., 2019a).HONO being the dominant N(III) species can lead to faster VL photo-oxidation~~ may have led to faster VL photo-oxidation.

7. Section 3.5. This section repeats what has been stated before. I would delete this section, show Figure 4 the first time discussing possible mechanisms, then refer to the Figure throughout the discussion of mechanisms (which is hopefully much shorter in the revised version).

Response: We agree with the reviewer. Section 3.5 has been deleted, and Fig. 4, now 2, has been shown for the first time when potential aqSOA formation pathways were discussed, then referred to throughout the text as follows:

Line 259: This compound was not observed in a parallel experiment in which AN was replaced with sodium nitrate (SN) (Fig. S6a; see Sect. 3.3 for discussion). The potential aqSOA formation  pathways via  VL photo-oxidation in this study are summarized in Fig. 2  At pH 4, ring-opening products (Fig. S5) from fragmentation in both VL* and VL+AN may have reacted with VL or dissolved ammonia to generate C$_{10}$H$_{10}$O$_5$ (No. 4, Table S2) (Pang et al., 2019b) or a potential imidazole derivative (C$_5$H$_5$N$_3$O$_2$; No. 5, Table S2), respectively. Moreover, nitrate photolysis products promoted functionalization and nitration (e.g., C$_{16}$H$_{10}$N$_2$O$_9$; No. 3, Table S2).

Line 364: At pH < 4, $^3$VL* likely have higher reactivity as suggested by the increased normalized abundance of oligomers (e.g., $C_{16}H_{14}O_6$; No. 6, Table S2 and $C_{31}H_{24}O_{11}$) and N-containing compounds (e.g., $C_{16}H_{10}N_2O_9$; No. 3, Table S2 and $C_{13}H_{14}N_2O_{10}$) (Fig. 2).

**Minor Comments**

1. Line 25. This notion of "efficiency" (i.e., which reaction path is faster) depends on the concentrations of the two oxidant precursors, VL and nitrate. Thus it's not a universally true statement.

Response: Thank you for this suggestion. The apparent quantum efficiency of GUA photodegradation ($\phi_{GUA}$) in the presence of either VL or nitrate during simulated sunlight illumination can be defined as (Anastasio et al., 1996; Smith et al., 2014, 2016):

$$\Phi_{GUA} = \frac{\text{mol GUA destroyed}}{\text{mol photons absorbed}} \qquad \text{(Eq. S9)}$$

$\Phi_{GUA}$ was calculated using the measured rate of GUA decay and rate of light absorption by either VL or nitrate through the following equation:

$$\Phi_{GUA} = \frac{\text{rate of GUA decay}}{\text{rate of light absorption by VL or nitrate}} = \frac{k\prime_{GUA} \times [GUA]}{\sum[\left(1 - 10^{-\varepsilon_\lambda [C] l}\right) \times I'_\lambda]} \qquad \text{(Eq. S10)}$$

where $k'_{GUA}$ is the pseudo-first-order rate constant for GUA decay, [GUA] is the concentration of GUA (M), $\varepsilon_\lambda$ is the base-10 molar absorptivity ($M^{-1}$ $cm^{-1}$) of VL or nitrate at wavelength $\lambda$, [C] is the concentration of VL or nitrate (M), $l$ is the pathlength of the illumination cell (cm), and $I'_\lambda$ is the volume-averaged photon flux (mol-photons $L^{-1}$ $s^{-1}$ $nm^{-1}$) determined from 2NB actinometry:

$$j(2NB) = 2.303 \times \Phi_{2NB} \times l \times \Sigma_{300\ nm}^{350\ nm}(\varepsilon_{2NB,\lambda} \times I'_\lambda \times \Delta\lambda) \qquad \text{(Eq. S11)}$$

The $\phi_{GUA}$ in the presence of nitrate ($1.3 \times 10^{-2} \pm 2.9 \times 10^{-3}$) is ~14 times larger than that in the presence of VL ($9.0 \times 10^{-4} \pm 4.0 \times 10^{-4}$), suggesting that nitrate-mediated photo-oxidation of GUA is more efficient than that by photosensitized reactions of VL. We have revised this in the text as follows and added the information shown above in the supporting information: **Text S7**. Estimation of the apparent quantum efficiency of guaiacol photodegradation.

Line 29: Furthermore, comparisons of the apparent quantum efficiency of guaiacol photodegradation indicate that in this study, guaiacol oxidation by photosensitized reactions of VL  is less efficient relative to nitrate-mediated photo-oxidation. Other relevant revisions in the text are as follows:

Line 514: As mentioned earlier,  $^3$VL* chemistry appears to be more important than that of nitrate photolysis even at 1:10 molar ratio of VL/nitrate on account of the much higher molar absorptivity of VL compared to that of nitrate (Fig. S1) and the high VL concentration (0.1 mM) used in this study. However, the apparent quantum efficiency of GUA photodegradation ($\phi_{GUA}$) in the presence of nitrate ($1.3 \times 10^{-2} \pm 2.9 \times 10^{-3}$) is ~14 times larger than that in the presence of VL ($9.0 \times 10^{-4} \pm 4.0 \times 10^{-4}$), suggesting that nitrate-mediated photo-oxidation of GUA is more efficient than that by photosensitized reactions of VL (see Text S7 for the more details).

Line 619: The oxidation of guaiacol, a non-carbonyl phenol, by photosensitized reactions of vanillin was also shown to be  less efficient than that by nitrate photolysis products based on its lower apparent quantum efficiency.

2. L. 42. "respectively" doesn't serve a purpose in this sentence.

Response: Agree, we have deleted 'respectively' from line 50 (formerly 42).

3. Section 2.1. What was the initial volume of solution illuminated? Were solutions stirred? What was the flow rate of gas (N2 or air) through the solution before and during the experiment?

Response: The initial volume of the illuminated solution is 500 mL. The solutions were continuously mixed throughout the experiments. A constant flow rate of 0.5 dm$^3$/min was used before and during the experiments. These have been added to the text as follows:

Line 102: Photo-oxidation experiments were performed in a  custom-built quartz photoreactor. The solutions (initial volume of 500 mL) were continuously mixed throughout the experiments using  a magnetic stirrer. The solutions were bubbled with synthetic air or nitrogen (N$_2$) (> 99.995%) (0.5 dm$^3$/min) for 30 min before irradiation to achieve air- and N$_2$-saturated conditions, respectively, and the bubbling was continued throughout the reactions (Du et al., 2011; Chen et al., 2020).

4. L. 100. Was there a difference in the temperature between the illuminated and dark solutions?

Response: For all experiments, the range in temperature fluctuations was 27 ± 2 °C.

5. L. 106. If the authors are going to abbreviate 2-propanol as IPA, it would be better to call it isopropyl alcohol to help readers remember the name of the abbreviation. NaBC is a poor choice for an abbreviation for sodium bicarbonate since BC stands for black carbon typically. Better to simply use its chemical formula, NaHCO3 or HCO3-, depending on the context.

Response: Sentences related to •OH scavengers have been deleted from the original manuscript.

6. L. 111. 2-propanol and bicarbonate were added in some experiments, but the description of why is odd. Their primary role will be OH scavengers, so it's strange to call them a VOC and

inorganic anion, respectively.  2-propanol is not a common atmospheric gas, so it's a poor choice of model VOC.  Similarly, calling bicarbonate an "inorganic anion" is a poor choice of words, since sulfate and nitrate are the classic inorganic ions.  Better to refer to 2-propanol and bicarbonate as "OH scavengers" since that is their main role.

Response: Sentences related to $^{\bullet}$OH scavengers have been deleted from the original manuscript.

7. L. 113. What does it mean that the OH scavengers were not added "in excess"? Since they're reacting with OH (which will have a very low concentration) they are technically in excess. Better to avoid this discussion, as it's not fruitful.  If you want to dive more into the OH scavengers, you could calculate the fraction of OH each intercepts in their respective solutions or the amount that they suppress the OH concentration.  (But, again, this depends on if the species were purged from solution.)

Response: Sentences related to $^{\bullet}$OH scavengers have been deleted from the original manuscript.

8. L. 151. The disproportionation of $HO_2/O_2^-$ is the same as the reaction of $HO_2$ with $O_2^-$, so this sentence repeats itself.

Response: The reviewer is right. We have corrected this sentence as follows:

Line 181: The disproportionation of $HO_2^{\bullet}/O_2^{\bullet-}$ (Anastasio et al., 1996)  form hydrogen peroxide ($H_2O_2$), which is a photolytic source of $^{\bullet}$OH.

9. L. 163. It's unclear what the authors mean by "…a minimal role for $^3VL^*$ in VL photo-oxidation". Do they mean that $^3VL^*$ + VL is an unimportant reaction (but see above about this) or that the direct photodegradation of VL doesn't proceed through the triplet state?

Response: In this study, the photodegradation of VL is mainly governed by VL triplets, as explained in our response to major comment #2 (Please see this response for the revised text).

10. L. 167. It is not true that $^1O_2$ has a much longer lifetime than $^3C^*$; rather, the lifetimes are approximately the same. In cloud and fog drops, the lifetime of $^1O_2$ is controlled by water deactivation and is approximately 5 us (see Bilski et al., 1997). The lifetime of $^3C^*$ is controlled by reaction with dissolved O2 and is approximately 1/((2E9 M-1 s-1)*(250uM)) ~ 2 us. Also, rather than the oxidant lifetime, it is the product of the oxidant concentration times its second-order rate constant that determines the relative importance of a given oxidant.

Response: We thank the reviewer for the correction. This sentence has been modified as part of the revision for major comment #2 (Please see this response for the revised text).

11. Page 7. This whole page is one paragraph. It should be trimmed to reduce speculative discussions of mechanisms, then broken into smaller pieces, focused on certain themes/points.

Response: We agree with the reviewer. Section 3.1.1 has been revised to reduce speculative discussions of mechanisms (Revisions for lines 188–220 are shown in response to major comment #2). Changes in the text are as follows:

Lines 220–240: The decay rate constant for VL+AN under air-saturated conditions was also higher  (6.6 times) than $N_2$-saturated conditions which may  be due to  reactions facilitated by nitrate photolysis products  that may have been enhanced in the presence of $O_2$  (Vione et al., 2005; Kim et al., 2014; Pang et al., 2019a). As shown later, more nitrogen-containing species were observed under air-saturated conditions. An example is enhanced VL nitration likely from increased $^•NO_2$ formation such as from the reaction of $^•OH$ and $O_2^{•-}$ with $NO_2^-$ (Reactions 4 and 5, respectively; Table 1) or the autoxidation of $^•NO$ from $NO_2^-$ photolysis (Reactions 6–9; Table 1) in aqueous solutions (Pang et al., 2019a). ~~Reactions involving $^•HO_2/O_2^{•-}$ which may originate from the photolysis of nitrate alone, likely from the production and subsequent photolysis of peroxynitrous acid (HOONO) (Reaction 10; Table 1) (Jung et al., 2017; Wang et al., 2021), or the reactions of $^3VL*$ in the presence of $O_2$, may have contributed as well. For instance, Wang et al. (2021) recently demonstrated that nitrate photolysis generates $^•HO_2/O_2^{•-}{}_{(aq)}$ and $HONO_{(g)}$ in the presence of dissolved aliphatic organic matter (e.g., nonanoic acid, ethanol), with the enhanced $HONO_{(g)}$ production caused by secondary photochemistry between $^•HO_2/O_2^{•-}{}_{(aq)}$ and photoproduced $NO_{x(aq)}$ (Reactions 11 and 12; Table 1), in agreement with Scharko et al. (2014). The significance of this increased HONO production is enhanced $^•OH$ formation (Reaction 13; Table 1). In addition, $^•HO_2$ can react with $^•NO$ (Reaction 10; Table 1) from $NO_2^-$ photolysis (Reaction 6; Table 1) to form HOONO, and eventually $^•NO_2$ and $^•OH$ (Reaction 14; Table 1) (Pang et al., 2019a).s~~ constants for VL* and VL+AN imply that $^3VL*$ chemistry still dominates even at 1:10 molar ratio of VL/nitrate, probably due to the much higher molar absorptivity of VL compared to that of nitrate (Fig. S1) and the high VL concentration (0.1 mM) used in this study. Although we have no quantification of the oxidants in our reaction systems as it is outside the scope of this study, these observations clearly substantiate that photosensitized oxidation of VL and nitrate-mediated VL photo-oxidation are more efficient in the presence of $O_2$.

Lines 241–300: The products from VL* under $N_2$-saturated conditions were mainly oligomers (e.g., $C_{16}H_{14}O_4$) (Fig. 1a), consistent with triplet-mediated oxidation forming higher molecular weight products, probably with less fragmentation relative to oxidation by $^•OH$ (Yu et al., 2014; Chen et al., 2020). A threefold increase in the normalized abundance of products was noted upon addition of nitrate (VL+AN under $N_2$-saturated conditions; Fig. 1b), and in addition to oligomers, functionalized monomers (e.g., $C_8H_6O_5$) and nitrogen-containing compounds (e.g., $C_8H_9NO_3$; No. 2, Table S2) were also observed, in agreement with $^•OH$-initiated oxidation yielding more functionalized/oxygenated products compared to triplet-mediated oxidation (Yu et al., 2014; Chen et al., 2020). Oligomers,  functionalized monomers (e.g., demethylated VL; Fig. S4), and nitrogen-containing compounds (e.g., $C_{16}H_{10}N_2O_9$; No. 3,Table S2) (for VL+AN) had higher normalized abundance  under air-saturated conditions

[revised manuscript text omitted]

12. L. 196. The text here and elsewhere discusses the abundance of specific products (not just the normalized product abundance). The abundance of each product should be added to Table S3, along with some estimate of the relative uncertainty of these values.

Response: The abundance mentioned in line 249 (formerly 196) pertains to the normalized abundance of all nitrogen-containing products for VL* at pH 4 under air-saturated conditions, not for specific products.

13. L. 224. How much lower are OS(C) values here compared to previous work on aqSOA? Compare these values.

Response: Our measured <$OS_c$> range from -0.28 to +0.12, while other studies on phenolic aqSOA formation reported <$OS_c$> ranging from -0.14 to +0.47 (Sun et al., 2010) and 0.04 to 0.74 (Yu et al., 2014). This information has been added to the text as follows:

Line 286: Our measured <$OS_c$> range from -0.28 to +0.12, while other studies on phenolic aqSOA formation reported <$OS_c$> ranging from -0.14 to +0.47 (Sun et al., 2010) and 0.04 to 0.74 (Yu et al., 2014). This is likely because we excluded contributions from ring-opening products, which may have higher $OS_c$ values as these products are not detectable in the positive ion mode.

14. L. 240. Is there any evidence that $^3VL^* + O_2$ directly makes OH? This would seem energetically unfavorable and also to be minor compared to energy transfer to make $^1O_2$.

Response: We do not have direct evidence of •OH formation from $^3VL^*$ + $O_2$, although trace amounts of $H_2O_2$ were likely formed during VL photodegradation (Li et al., 2014) similar to the case of other phenolic compounds (Anastasio et al., 1996). This statement has been added to line 241.

Line 316: Trace amounts of $H_2O_2$ were likely formed during VL photodegradation (Li et al., 2014), similar to the case of other phenolic compounds (Anastasio et al., 1996).

15. L. 242. In the presence of $O_2$, the ketyl radical is probably too short lived (it reacts with O2 to make an alpha-hydroxy peroxyl radical) to combine appreciably with a phenoxyl radical. But the phenoxyl radical is in resonance with a carbon-centered cyclohexadienyl radical that is longer lived; these two species can couple (Yu et al., ACP, 2014).

Response: We thank the reviewer for the additional information. We have now revised line 311 to include this as follows:

Line 317: Oligomers can then form via the coupling of phenoxy radicals or phenoxy and cyclohexadienyl  radicals (Sun et al., 2010; Yu et al., 2014; Vione et al., 2019).

16. L. 261. The rate constant for $H_2O_2$ formation is fastest near the pKa of $HO_2$, i.e., pH 4.8, so one wouldn't expect greater $H_2O_2$ formation at pH 2.5 compared to pH 4. But this also depends on the pH dependence of the $HO_2/O_2^-$ sources and sinks.

Response: Thank you for pointing this out. This statement has been deleted.

17. L. 264. This discussion of the pH dependence of N(III) photolysis doesn't seem applicable since the addition of nitrate makes a negligible contribution to VL decay. Just because N(III) photolysis is pH dependent doesn't mean it matters here.

Response: Thank you for pointing out this error. The reviewer is correct that the pH dependence of N(III) is not relevant for the discussion of pH effects on VL decay rate constants. Some of these statements have been transferred to the next paragraph (from line 360) to provide a potential explanation for the increased normalized abundance of nitrogen-containing compounds at lower pH. Revisions for lines 333–354 are shown in our response to major comment #6. Other changes in the text are as follows:

Line 359–364: For VL+AN, the normalized abundance of nitrogen-containing compounds was also higher  at lower pH (Table S2), likely due to increased •OH and •$NO_2$ formation, which may be caused by the dependence of N(III) ($NO_2^-$ + HONO) speciation on solution acidity (Pang et al., 2019a). At pH 3.3, half of N(III) exists as HONO (Fischer and Warneck, 1996; Pang et al., 2019a), which has a higher quantum yield for •OH formation than that of $NO_2^-$ in the near-UV region

(Arakaki et al., 1999; Kim et al., 2014). Also, $NO_2^-$/HONO can generate $^\bullet NO_2$ via oxidation by $^\bullet OH$ (Reactions 4 and 10; Table 1) (Pang et al., 2019a).

Line 372: Essentially, the higher reactivity of $^3VL^*$ and predominance of HONO over nitrite at lower pH may have resulted in increased formation of products mainly composed of oligomers and functionalized monomers.

18. L. 299. Why would the presence of HO2 lead to more dimer formation? HO2 (and O2-) are too weak to oxidize phenols at any significant rate.

Response: Sentences related to $^\bullet OH$ scavengers have been deleted from the original manuscript.

19. Lines 298-301: This argument is circular: IPA cannot make more OH by scavenging OH and turning it into HOOH, which then photolyzes to make OH. Think of the associated stoichiometry. IPA will suppress [OH] because it is an OH sink, thus rendering OH an insignificant oxidant for VL. The observation that IPA has a negligible impact on VL decay (Fig. S3c) indicates that OH is not important as an oxidant for VL (with or without IPA) or that the IPA was mostly purged from the system.

Response: Sentences related to $^\bullet OH$ scavengers have been deleted from the original manuscript.

20. L. 310. IPA makes no difference in the VL kinetics, whether nitrate is present or not. So please don't make sweeping statements such as "...the role of nitrate in VL photo-oxidation is enhanced in the presence of IPA...". And don't suggest that OH is an important intermediate in the formation of a product in the presence of IPA (e.g., line 311), since IPA will greatly suppress the OH concentration.

Response: Sentences related to $^\bullet OH$ scavengers have been deleted from the original manuscript.

21. L. 313-322. It is hard to believe that 1 mM of IPA can significantly disrupt the structure of 55 M water. In any case, there is no increase in light absorption by the aqSOA formed in the presence of IPA (Fig. 2), so the Berke mechanism seems unimportant. Most of this should be deleted.

Response: Sentences related to $^\bullet OH$ scavengers have been deleted from the original manuscript.

22. L. 327. It is difficult to imagine that carbonate radical is a significant oxidant in these experiments: carbonate rate constants are relatively slow (compared to triplets or OH) and VL photodegradation is very fast. If the authors want to propose carbonate radical as an important sink, they need to do some calculations of its steady-state concentration and estimate the corresponding rate of VL loss. Again, the qualitative normalized abundance of products is driving these uncertain statements, while the quantitative photodecay rates and light absorption are showing there is no significant effect of bicarbonate. Lead with the latter observations, as they are more robust.

Response: Sentences related to •OH scavengers have been deleted from the original manuscript.

23. L. 336. 1 mM IPA or bicarbonate is not high enough to reduce the cage effect from nitrate photolysis. In any case, IPA or bicarbonate are OH sinks, so they will suppress, not enhance, the OH concentration.

Response: Sentences related to •OH scavengers have been deleted from the original manuscript.

24. L. 376. If this proposed mechanism was true, then VL decay would be significantly faster in the presence of nitrate, but this is not the case. It's not clear what the authors are trying to explain here - is it the increase in oligomerization at higher [VL]? The explanation for this is probably that the concentrations of phenoxyl radicals (and the related, carbon-centered cyclohexadienyl radicals) increase with [VL], making radical-radical recombination to form oligomers a more significant fate.

Response: Thank you for pointing this out. We apologize for the confusion. There should have been an explanation here for the increased oligomerization observed at higher [VL]. We agree with the reviewer that a possible reason for this is the increased concentration of radicals with [VL] (added to the revised text). The succeeding lines are for comparing the potential pathways at 1:1 VL/nitrate and 1:100 VL/nitrate only at low [VL]. The text has been revised to clarify these as follows:

Lines 466–485: For both VL* and VL+AN, the contribution of < 200 m/z to the normalized abundance of products was higher at low [VL] than at high [VL], while the opposite was observed for > 300 m/z (Fig. S12d). This indicates that functionalization was favored at low [VL], as supported by the higher <$OS_c$>, while oligomerization was the dominant pathway at high [VL], consistent with more oligomers or polymeric products reported from high phenols concentration (e.g., 0.1 to 3 mM) (Li et al., 2014; Slikboer et al., 2015; Ye et al., 2019). This is probably due to an increased concentration of phenoxy radicals (in resonance with a carbon-centered cyclohexadienyl radical) at high [VL], promoting radical-radical polymerization (Sun et al., 2010; Li et al., 2014).  At low [VL], the contribution of < 200 m/z to the normalized abundance of products was higher for 1:1 than 1:100 VL/nitrate molar ratio,  suggesting the prevalence of functionalization for the former  In addition, 1:1 VL/nitrate (A11; Table S2) had higher <$OS_c$> than 1:100 VL/nitrate (A12; Table S2), indicating the formation of more oxidized products, but had fewer N-containing compounds compared to the latter. A possible explanation is that at 1:1 VL/nitrate, VL may  competes with $NO_2^-$ for •OH (from nitrate or nitrite photolysis, Reaction 4; Table 1) and indirectly reduce •$NO_2$. Similarly, hydroxylation has been suggested to be a more important pathway for 1:1 VL/nitrite than in 1:10 VL/nitrite (Pang et al., 2019a).  Fragmentation, which leads to the decomposition of previously formed oligomers and generation

of small, oxygenated products such as organic acids (Huang et al., 2018), may also occur for the low [VL] experiments. However, its importance would likely be observed at longer irradiation times, similar to the high [VL] experiments.

25. L. 405. GUA should not undergo any direct photochemistry, so its decay in the absence of VL or AN suggests either that there is an oxidant-making contaminant in the system (that is consumed within a few hours) or that GUA is evaporating during illumination. But there is no GUA loss in the dark: is this because the temperature was cooler in the dark?

Response: Vione et al. (2019) also observed the direct photodegradation of GUA (0.1 mM) upon irradiation using Xe lamp. Moreover, Sun et al. (2010) reported an intermediate rate of direct photoreaction for GUA (0.1 mM), yielding aqSOA including GUA dimers (similar to what we observed). There was no significant loss of GUA both after 30 min of purging in the dark (-0.36%) and after 360 min of dark control experiments (-1.7%). Also, the temperature fluctuations (27 ± 2 °C) were minimal.

26. L. 498. This sentence mentions "Further enhancement of VL photo-oxidation...in the presence of nitrate...", but VL photo-oxidation (i.e., photodegradation) was not enhanced in the presence of nitrate.

Response: This statement refers to the presence of both $O_2$ and nitrate resulting in the highest normalized abundance of products (including N-containing compounds) and $<OS_c>$ among experiments A5–A8. We have revised this as follows:

Line 606–611:  In contrast, $^3VL^*$-initiated reactions proceeded rapidly under air-saturated conditions ($O_2$ is present) as indicated by higher VL decay rate constant and increased normalized abundance of products. For pH 4 experiments, the presence of both $O_2$ and nitrate resulted in the highest normalized abundance of products (including N-containing compounds) and $<OS_c>$, which may be due to the presence of $O_2$ promoting VL nitration.

27. L. 1004. The author order is incorrect on the Tinel et al. ref.

Response: Thank you for the correction. We have now amended the author order for this reference as follows and revised the corresponding in-text citations:

George, C., Brüggemann, M., Hayeck, N., Tinel, L., and Donaldson, D. J.: Interfacial photochemistry: physical chemistry of gas-liquid interfaces, in: Developments in Physical & Theoretical Chemistry, edited by: Faust, J. A. and House, J. E., Elsevier, 435–457, https://doi.org/10.1016/B978-0-12-813641-6.00014-5, 2018.

28. Table 1. The quantum yield for Rxn 3 is not 0.001.  This is a misperception based on the O(3P) result of Warneck and Wurzinger (J Phys Chem, 1988); their paper shows a value of ~0.01 for more direct (nitrite) measurements.  Benedict et al. (Env Sci Technol., 2017) confirmed this higher

value. This error doesn't affect the current work, but it would be a shame to propagate the misperception.

Response: We thank the reviewer for the careful read. We now have corrected this value in Table 1:

$NO_3^- + h\nu \rightarrow NO_2^- + O(^3P)$; $\varphi$ = 0.1

29. Figure 4. (a) The resolution of the figure is poor, so it's fuzzy and hard to read. (b) Scheme 1 suggests that oligomers are only formed at pH < 4, which isn't true, as past work has shown oligomer formation in similar phenol systems at pH 5. (c) Ketyl radicals formed by $^3C^*$ + phenol typically are shown as phenoxyl OH group (a result of the triplet abstracting a hydrogen) and no double bond between the C and O. As stated earlier, their lifetimes are short in the presence of O2, so they're unlikely to do the coupling as shown here.

Response:

(a) This may be a formatting issue. We ensured that this is avoided in the revised version.

(b) The reviewer is right that oligomer formation in similar phenol systems was also observed at pH 5. However, the molecular formulas initially presented in Fig. 4 were the most abundant products or products with a significant increase in normalized abundance. To avoid this confusion, Fig. 4, now 2, has been revised to show the major products for each condition, with a marker for the most abundant products.

(c) Thank you for catching this error. Ketyl radical has been deleted in Fig. 2.

**Supplemental Material Notes**

1. General note – it would have been helpful to have line numbers in the supplement.

Response: Line numbers have now been added to the supporting information as well.

2. Text S3. Were calibration curves only made once? Were they actually used in quantifying VL and GUA? (I don't see the need since absolute values are not needed in the kinetic plots.)

Response: No, the calibration curves were prepared weekly to account for potential changes in the detector response of UHPLC. These calibration curves were used to quantify VL and GUA as the calculated VL concentration was used for estimating the normalized abundance of products.

3. Text S6. (a) It's unclear what is meant by "Then, the average relative intensity absorbed by 2NB solution as a function of wavelength was calculated." Can you show this with an equation? (b) How much did the photon flux vary between experiments? Was this determined? If not, this variation is a source of variability in the kinetic measurements.

Response: (a) We apologize for the confusion. This statement pertains to a scaling factor (SF) that was used to determine the absolute photon flux in the reactor, $I'_\lambda$. Similar to Smith et al. (2014, 2016), we measured the spectral shape of the photon output of our illumination system (i.e., the relative flux at each wavelength) using a high-sensitivity spectrophotometer (Brolight Technology Co. Ltd, Hangzhou, China). Using an SF, this measured relative photon output, $I^{relative}_\lambda$, is related to $I'_\lambda$ as follows:

$$I'_\lambda = I^{relative}_\lambda \times SF \qquad\qquad \text{(Eq. S6)}$$

Substitution of Eq. S6 into Eq. S5 and rearrangement yields:

$$j(2NB) = 2.303 \times (10^3 \text{ cm}^3 \text{ L}^{-1} \times 1 \text{ mol/N}_A \text{ mlc}) \times \sum(I'_\lambda \times \Delta\lambda \times \varepsilon_{2NB,\lambda} \times \Phi_{2NB}) \qquad \text{(Eq. S5)}$$

Where $j(2NB)$ is the 2NB decay rate constant, $N_A$ is Avogadro's number, $I'_\lambda$ is the actinic flux (photons cm$^{-2}$ s$^{-1}$ nm$^{-1}$), $\Delta\lambda$ is the wavelength interval between actinic flux data points (nm), and $\varepsilon_{2NB,\lambda}$ and $\Phi_{2NB,\lambda}$ are the base-10 molar absorptivity (M$^{-1}$ cm$^{-1}$) and quantum yield (molecule photon$^{-1}$) for 2NB, respectively. Values of $\varepsilon_{2NB,\lambda}$ (in water) at each wavelength under 298 K and a wavelength-independent $\Phi_{2NB}$ value of 0.41 were adapted from Galbavy et al. (2010).

$$SF = \frac{j(2NB)}{2.303 \times (10^3 \text{ cm}^3 \text{ L}^{-1} \times 1 \text{ mol/N}_A \text{ mlc}) \times \sum(I^{relative}_\lambda \times \Delta\lambda \times \varepsilon_{2NB,\lambda} \times \Phi_{2NB})} \qquad \text{(Eq. S7)}$$

and substitution of (Eq. S6) into (Eq. S7) yields:

$$I'_\lambda = I^{relative}_\lambda \frac{j(2NB)}{2.303 \times (10^3 \text{ cm}^3 \text{ L}^{-1} \times 1 \text{ mol/N}_A \text{ mlc}) \times \sum(I^{relative}_\lambda \times \Delta\lambda \times \varepsilon_{2NB,\lambda} \times \Phi_{2NB})} \qquad \text{(Eq. S8)}$$

Finally, $I'_\lambda$ was estimated through Eq. S8. The estimated photon flux in the aqueous reactor is shown in Figure S1.

We have added the above information to Text S6.

(b) The $j(2NB)$ in this study varied from 0.0021 to 0.0026 s$^{-1}$. The decay rate constants have now been normalized to the photon flux, and the updated values are shown in Table 2, formerly S2.

4. Table S2. (a) VL (and GUA) decays are rate constants, not decay rates. (b) For reference, it would be helpful to give the OS(C) of VL. (c) What is pH of expt. A19?

Response: (a) Thank you for catching this error. This has been corrected in Table S2, now 2, as well as elsewhere in the text:

**Table 2.** Reaction conditions, initial VL (and GUA) decay rate constants

Line 142: the calculation of GUA decay rate constant.

Line 172: initial VL (and GUA) decay rate constants,

Line 204: Contrastingly, the VL* decay rate constant under air-saturated conditions

Line 220: The decay rate constant for VL+AN

Line 235: Nevertheless, the comparable decay rate constants

Line 331: The decay rate constants

Line 509: GUA decay rate constant was  higher by 2.2 (GUA+VL)

Line 512: This enhanced GUA decay rate constant

Line 574: Although nitrate did not substantially affect the VL decay rate constants,

(b) Agree, we have now added the $OS_c$ of VL (-0.25) to Table 2.

(c) The pH for exp 19, now 15, is 4, same with other experiments involving GUA. This is already listed in the second column of Table 2.

5. Figure S1. The vanillin spectrum has a problem around 305 nm - a large discontinuity that is probably caused by lamp switch. Either reacquire the spectrum or replace with a published value.

Response: Thank you for catching this. Fig. S1 has been revised to correct this.

6. Figure S3. Were the decays ever determined multiple times for the same condition? It would be helpful to show these results and derive a relative uncertainty for decay rate constants.

Response: Yes, the decays reported in Fig. S3 are the average of results from triplicate experiments, and the error bars for each data point are already shown. The photon flux-normalized decay rate constants have also been updated in Table S2, now 2, along with the standard deviation for each condition (Please see our response to major comment #5). The following sentence has been added in the methods section to clarify this:

Line 132: Each experiment was repeated independently at least three times and measurements were done in triplicate. The reported decay rate constants and absorbance enhancement are the average of results from triplicate experiments, and the corresponding errors represent one standard deviation.

We also added this note to figure captions when applicable: most error bars are smaller than the markers.

7. Figure S6. How can we tell that the imidazole formed in the AN experiment was not formed in the SN experiment?  It would be helpful to put a marker on the two plots of Fig. S6 to show where the imidazole showed up in the AN experiment.

Response: Thank you for pointing this out. Fig. S6 has been revised to show a marker for the potential imidazole compound.

**Recommendation**

I recommend that the manuscript be majorly revised and then reconsidered.

[revised manuscript text omitted]

---

## Author Comment (AC2)

Author Response for "Aqueous SOA formation from the photo-oxidation of vanillin: Direct photosensitized reactions and nitrate-mediated reactions" by Mabato et al.

We thank the reviewer for the thorough review and many constructive comments that helped improve the manuscript. Our point-by-point responses are below (changes to the original manuscript text and supporting information are in red, moved content in double-line strikethrough, and removed content in strikethrough). Please note that the line numbers in the responses refer to our revised manuscript with tracked changes. Also, please note that because we restructured the manuscript, the numbering of some figures and tables in the revised manuscript is different from those in the original manuscript.

**Reviewer 2**

The manuscript describes very well-designed studies of vanillin photooxidation in bulk liquid solutions where pH, concentrations, reactant ratios, dissolved gases (N2 or O2), ions (nitrate, bicarbonate) and other species (isopropanol) were varied in many combinations. The work is technically sound, with the loss of reactants, the identification and quantification of products, and the absorbance changes in solution all monitored hourly. The authors exhaustively discuss the differences between each experimental variation, pulling out as much detail as possible. This paper will be of interest to those interested in biomass burning aerosol and brown carbon formation, and is publishable after major revision to address the following points.

1. In places the discussion veers off into speculation, or suggests theories that aren't adequately explained enough to be convincing to the reader, as noted below. Generally the discussion is convincing and well-connected to the literature, but the discussion section reads like it has a thousand detailed conclusions, leaving the reader often feeling "lost in the weeds" and blunting the impact of the work. In general, the focus of the paper could be improved by moving Table 1 to the SI, removing a lot of speculative discussion, and bringing Tables S2 and maybe S3 from the SI to the main paper. These tables are more vital to the discussion at many points, in my opinion.

Response: Thank you for the suggestion. Speculative discussions have been removed from the revised text. First, Section 3.1.1 has been amended to emphasize the importance of VL triplets:

Line 187:  VL photo-oxidation under $N_2$ and air-saturated conditions

Lines 188–240:  The photo-oxidation of VL  under both $N_2$- and air--saturated conditions (Fig. S3a) were carried out at pH 4, which is representative of moderately acidic aerosol and cloud pH values (Pye et al., 2020). No significant VL loss was observed for dark experiments. The oxidation of ground-state VL by $^3VL^*$ via H-atom

abstraction or electron transfer can form phenoxy (which is in resonance with a carbon-centered cyclohexadienyl radical that has a longer lifetime) and ketyl radicals (Neumann et al., 1986a, 1986b; Anastasio et al., 1996). The coupling of phenoxy radicals or phenoxy and cyclohexadienyl radicals can form oligomers as observed for both $N_2$- and air-saturated experiments (see discussions later). However, the little decay of VL under $N_2$-saturated condition indicates that these radicals probably predominantly decayed via back-hydrogen transfer to regenerate VL (Lathioor et al., 1999). A possible explanation for this is the involvement of $O_2$ in the secondary steps of VL decay. For instance, a major fate of the ketyl radical is reaction with $O_2$ (Anastasio et al., 1996). In the absence of $O_2$, radical formation occurs, but the forward reaction of ketyl radical and $O_2$ is blocked, leading to the regeneration of VL as suggested by the minimal VL decay. Aside from potential inhibition of secondary oxidants generation (Chen et al., 2020), $N_2$ purging may have also hindered the secondary steps for VL decay.

 Contrastingly, the VL* decay rate constant under air-saturated conditions was 4 times higher. As mentioned earlier, secondary oxidants ($^1O_2$, $O_2^{\bullet-}$/$^\bullet HO_2$, $^\bullet OH$) can be generated from $^3$VL* when $O_2$ is present (e.g., under air-saturated conditions). However, the photo-oxidation of VL in this study is likely mainly governed by $^3$VL* and that these secondary oxidants have only minor participation.  $^1O_2$ is also a potential oxidant for phenols (Herrmann et al., 2010; Minella et al., 2011; Smith et al., 2014), but $^1O_2$ reacts much faster (by ~60 times) with phenolate ions compared to neutral phenols (Tratnyek and Hoigne, 1991; Canonica et al., 1995; McNally et al., 2005). Under the pH values (pH 2.5 to 4) considered in this study, the amount of phenolate ion is negligible, so the reaction between VL and $^1O_2$ should be slow. Interestingly, however, $^1O_2$ has been shown to be important in the photo-oxidation of 4-ethylguaiacol ($pK_a$ = 10.3) by $^3$C* of 3,4-dimethoxybenzaldehyde (solution with pH of ~3) (Chen et al., 2020). Furthermore, while the irradiation of other phenolic compounds can produce $H_2O_2$, a precursor for $^\bullet OH$ (Anastasio et al., 1996), the amount of $H_2O_2$ is small. Based on this, only trace amounts of $H_2O_2$ were likely generated from VL* (Li et al., 2014) under-air saturated conditions, suggesting that contribution from $^\bullet OH$ was minor. Overall, these suggest that VL photo-oxidation in this study is driven by $^3$VL*. Further study on the impact of $O_2$ on the reactive intermediates involved is required to understand the exact mechanisms occurring under air-saturated conditions. Nonetheless, the VL* decay trends clearly indicate that $O_2$ is important for efficient VL photo-oxidation The decay rate constant for VL+AN under air-saturated conditions was also higher  (6.6 times) than $N_2$-saturated conditions which may  be due to  reactions facilitated by nitrate photolysis products  that may have been enhanced in the presence of $O_2$  (Vione et al., 2005; Kim et al., 2014; Pang et al., 2019a). As shown later, more nitrogen-containing species were observed under air-saturated conditions. An example is enhanced VL nitration likely from increased $^\bullet NO_2$ formation such as from the reaction of $^\bullet OH$ and $O_2^{\bullet-}$ with $NO_2^-$ (Reactions 4 and 5, respectively; Table 1) or the autoxidation of $^\bullet NO$ from $NO_2^-$ photolysis (Reactions 6–9; Table 1) in aqueous solutions (Pang et al., 2019a).

~~Reactions involving $^\bullet HO_2/O_2^{\bullet-}$ which may originate from the photolysis of nitrate alone, likely from the production and subsequent photolysis of peroxynitrous acid (HOONO) (Reaction 10; Table 1) (Jung et al., 2017; Wang et al., 2021), or the reactions of $^3VL^*$ in the presence of $O_2$, may have contributed as well. For instance, Wang et al. (2021) recently demonstrated that nitrate photolysis generates $^\bullet HO_2/O_2^{\bullet-}$ (aq) and $HONO_{(g)}$ in the presence of dissolved aliphatic organic matter (e.g., nonanoic acid, ethanol), with the enhanced $HONO_{(g)}$ production caused by secondary photochemistry between $^\bullet HO_2/O_2^{\bullet-}$ (aq) and photoproduced $NO_{x(aq)}$ (Reactions 11 and 12; Table 1), in agreement with Scharko et al. (2014). The significance of this increased HONO production is enhanced $^\bullet OH$ formation (Reaction 13; Table 1). In addition, $^\bullet HO_2$ can react with $^\bullet NO$ (Reaction 10; Table 1) from $NO_2^-$ photolysis (Reaction 6; Table 1) to form HOONO, and eventually $^\bullet NO_2$ and $^\bullet OH$ (Reaction 14; Table 1) (Pang et al., 2019a).rates~~ constants for VL* and VL+AN imply that $^3VL^*$ chemistry still dominates even at 1:10 molar ratio of VL/nitrate, probably due to the much higher molar absorptivity of VL compared to that of nitrate (Fig. S1) and the high VL concentration (0.1 mM) used in this study. Although we have no quantification of the oxidants in our reaction systems as it is outside the scope of this study, these observations clearly substantiate that  photosensitized oxidation of VL and nitrate-mediated VL photo-oxidation are more efficient in the presence of $O_2$.

Moreover, Section 3.1.3 (Effect of VOCs and inorganic anions) and related sentences have also been deleted based on the likely minor contribution of $^\bullet OH$ to VL photo-oxidation in this study as pointed out by Reviewer 1 and suggested by other published literature (Anastasio et al., 1996; Li et al., 2014). Also, Section 3.5 has been deleted and Fig. 4, now 2, has been shown for the first time when potential aqSOA formation pathways were discussed, then referred to throughout the text. Table 1, which was reduced to half, was maintained in the revised version, while Table S2 (now 2) was moved to the main text, as suggested by the reviewer.

2. I do not trust using results for IPA to make generalizations about the effect of all VOCs on vanillin photooxidation. The authors repeat this questionable generalization several times throughout the manuscript, including twice in the abstract. Especially because the authors' explanation for the effect of IPA on their results relies on alcohol / water microstructure arguments, generalization to all VOCs seems unwarranted. Plus, IPA would be present only at very low concentrations in aqueous aerosol or cloud droplets due to its high volatility. It would be more appropriate if the authors remove (or heavily qualify) all statements about VOCs.

Response: We concur with the reviewer that the initial generalization for the effect of all VOCs on VL photo-oxidation based on IPA results is unsubstantiated. Moreover, the contribution of $^\bullet OH$ to VL photo-oxidation in this study is likely minimal, as pointed out by Reviewer 1 and suggested by other published literature (Anastasio et al., 1996; Li et al., 2014). Section 3.1.3 (Effect of VOCs and inorganic anions) and related sentences have been deleted altogether. Nonetheless, we maintain that it would be worthwhile to explore the effects of other potential aerosol constituents on aqSOA formation and photo-oxidation studies (lines 627–631).

3. At several points, the authors discuss rather small differences between experiments (factors of 1.2 to 1.5) as significant, but the uncertainties in the parameter values being compared are never quantified. This raises doubts in readers' minds about which differences are actually statistically significant. Some discussion of uncertainties and random error is needed.

Response: Thank you for pointing this out. We have added relevant statements to discuss these uncertainties as follows:

Line 132: Each experiment was repeated independently at least three times and measurements were done in triplicate. The reported decay rate constants and absorbance enhancement are the average of results from triplicate experiments, and the corresponding errors represent one standard deviation.

The footnote of Table S2, now 2, has been revised as follows:

Table 2: [b]The data fitting was performed in the initial linear region. Each value is the average of results from triplicate experiments. Errors represent one standard deviation.

The revised text also now indicates whether a difference of less than a factor of 2 is statistically significant or not:

Line 331: The decay rates constants for both VL* and VL+AN increased as pH decreased (VL* and VL+AN at pH 2.5: 1.65 and 1.43 times faster than at pH 4, respectively) (Fig. S3b). These differences in decay rate constants are small but statistically significant ($p < 0.05$).

Line 510: The enhancement of GUA decay rate constant in the presence of VL is statistically significant ($p < 0.05$), while that in the presence of AN is not ($p > 0.05$).

4. The argument that $^3$VL* is more reactive in its protonated form as an explanation for the observed pH effects does not make sense to me. The pKa of VL is 7.4, which means that more than 99.9% of it is protonated in all experiments, negating the possibility of any detectable acceleration at low pH by this mechanism. Furthermore, the authors describe reasonable alternative explanations for their observed pH effects, such as the more efficient photolysis of HONO vs NO2- producing more OH radicals at low pH. However, the questionable claim that 3VL* is more reactive in its protonated form is repeated several times throughout the manuscript (for example, lines 267, 270, 280, 449 and 500). This claim needs to be convincingly justified or removed from the manuscript.

Response: The p$K_a$ (7.4) mentioned by the reviewer is for ground state VL, while the p$K_a$ for the VL triplet has been reported to be 4.0 (Smith et al., 2016). As there are a greater fraction of VL triplets that are protonated at pH 2.5 (0.96) than at pH 4 (0.5), it is possible that the pH dependence observed in this study is due to $^3$VL*being more reactive in its protonated form. Smith et al. (2016) also observed a pH dependence for the direct photodegradation of VL (0.005 mM) (rate constants at pH ≤ 3 are ~two times lower than at pH ≥ 5) which they attributed to the

sensitivity of the excimer of VL (i.e., the charge-transfer complex formed between an excited state VL molecule and a separate ground state VL molecule; Birks, 1973, Turro et al., 2010) to acid-base chemistry. The opposite trend observed in this study for 0.1 mM VL may be due to the reactivities of the protonated and neutral forms of the $^3$VL* being dependent on the VL concentration (Smith et al., 2016). It has been reported that the quantum yield for direct VL photodegradation is higher at pH 5 than at pH 2 for 0.005 mM VL, but they are not statistically different for 0.03 mM VL (Smith et al., 2016). We have added a statement to include the sensitivity of the excimer of VL to acid-base chemistry as another possibility for the observed pH dependence. In addition, we have removed the discussion of N(III) photolysis as an alternative explanation for the effects of pH on VL decay kinetics as there is no significant difference between the decay rate constants of VL* and VL+AN. Changes in the text are as follows:

Lines 331–352: The decay rate constants for both VL* and VL+AN increased as pH decreased (VL* and VL+AN at pH 2.5: 1.6 and 1.4 times faster than at pH 4, respectively) (Fig. S3b). These differences in decay rate constants are small but statistically significant ($p < 0.05$). The p$K_a$ for the VL triplet has been reported to be 4.0 (Smith et al., 2016). As there are a greater fraction of VL triplets that are protonated at pH 2.5 (0.96) than at pH 4 (0.5), it is possible that the pH dependence of the decay rate constants observed in this study is due to $^3$VL* being more reactive in its protonated form. Smith et al. (2016) also observed a pH dependence for the direct photodegradation of VL (0.005 mM) (rate constants at pH ≤ 3 are ~two times lower than at pH ≥ 5) which they attributed to the sensitivity of the excimer of VL (i.e., the charge-transfer complex formed between an excited state VL molecule and a separate ground state VL molecule; Birks, 1973, Turro et al., 2010) to acid-base chemistry. The opposite trend observed in this study for 0.1 mM VL may be due to the reactivities of the protonated and neutral forms of the $^3$VL* being dependent on the VL concentration (Smith et al., 2016).  It has been reported that the quantum yield for direct VL photodegradation is higher at pH 5 than at pH 2 for 0.005 mM VL, but they are not statistically different for 0.03 mM VL (Smith et al., 2016). ~~Also, increases in hydrogen ion concentration can enhance the formation of HO₂• and H₂O₂ and in turn, •OH formation (Du et al., 2011). In addition to these pH influences on VL*, the dependence of N(III) (NO₂⁻ + HONO) speciation on solution acidity (Pang et al., 2019a) also contributed to the observed pH effects for VL+AN. At pH 3.3, half of N(III) exists as HONO (Fischer and Warneck, 1996; Pang et al., 2019a), which has a higher quantum yield for •OH formation than that of NO₂⁻ in the near-UV region (Arakaki et al., 1999; Kim et al., 2014). The increased •OH formation rates as pH decreases can lead to faster VL decay (Pang et al., 2019a). Also, NO₂⁻/HONO can generate •NO₂ via oxidation by •OH (Reactions 4 and 15; Table 1) (Pang et al., 2019a).HONO being the dominant N(III) species can lead to faster VL photo-oxidation~~ may have led to faster VL photo-oxidation.

Line 356: further indicating that $^3$VL*  may be more reactive in their protonated form.

Line 372: Essentially, the higher reactivity of $^3$VL* and predominance of HONO over nitrite at lower pH may have resulted in increased formation of products mainly composed of oligomers and functionalized monomers.

**Specific comments:**

1. Line 25: The authors conclude that photosensitized reactions of VL were "more efficient" relative to nitrate-mediated photo-oxidation. However, as pointed out by the authors, VL is much more light-absorbing that nitrate. Can the authors make a comparative statement after taking this difference into account? Which is more efficient on a per-photon-absorbed basis? This would be a more appropriate comparison of reaction efficiency.

Response: Thank you for this suggestion. The apparent quantum efficiency of GUA photodegradation ($\phi_{GUA}$) in the presence of either VL or nitrate during simulated sunlight illumination can be defined as (Anastasio et al., 1996; Smith et al., 2014, 2016):

$$\Phi_{GUA} = \frac{\text{mol GUA destroyed}}{\text{mol photons absorbed}} \qquad \text{(Eq. S9)}$$

$\Phi_{GUA}$ was calculated using the measured rate of GUA decay and rate of light absorption by either VL or nitrate through the following equation:

$$\Phi_{GUA} = \frac{\text{rate of GUA decay}}{\text{rate of light absorption by VL or nitrate}} = \frac{k\prime_{GUA} \times [GUA]}{\Sigma[\left(1-10^{-\varepsilon_\lambda [C]l}\right) \times I'_\lambda)} \qquad \text{(Eq. S10)}$$

where $k'_{GUA}$ is the pseudo-first-order rate constant for GUA decay, [GUA] is the concentration of GUA (M), $\varepsilon_\lambda$ is the base-10 molar absorptivity (M$^{-1}$ cm$^{-1}$) of VL or nitrate at wavelength $\lambda$, [C] is the concentration of VL or nitrate (M), $l$ is the pathlength of the illumination cell (cm), and $I'_\lambda$ is the volume-averaged photon flux (mol-photons L$^{-1}$ s$^{-1}$ nm$^{-1}$) determined from 2NB actinometry:

$$j(2NB) = 2.303 \times \Phi_{2NB} \times l \times \Sigma_{300\ nm}^{350\ nm} (\varepsilon_{2NB,\lambda} \times I'_\lambda \times \Delta\lambda) \qquad \text{(Eq. S11)}$$

The $\phi_{GUA}$ in the presence of nitrate ($1.3 \times 10^{-2} \pm 2.9 \times 10^{-3}$) is ~14 times larger than that in the presence of VL ($9.0 \times 10^{-4} \pm 4.0 \times 10^{-4}$), suggesting that nitrate-mediated photo-oxidation of GUA is more efficient than that by photosensitized reactions of VL. We have revised this in the text as follows and added the information shown above in the supporting information: **Text S7. Estimation of the apparent quantum efficiency of guaiacol photodegradation.**

Line 29: Furthermore, comparisons of the apparent quantum efficiency of guaiacol photodegradation indicate that in this study, guaiacol oxidation by photosensitized reactions of VL  is less efficient relative to nitrate-mediated photo-oxidation.

Other relevant revisions in the text are as follows:

Line 514: As mentioned earlier,  $^3$VL* chemistry appears to be more important than that of nitrate photolysis even at 1:10 molar ratio of VL/nitrate on account of the much higher molar absorptivity of VL compared to that of nitrate (Fig. S1) and the high VL concentration (0.1 mM) used in this study. However, the apparent quantum efficiency of GUA photodegradation ($\phi_{GUA}$) in the presence of nitrate ($1.3 \times 10^{-2} \pm 2.9 \times 10^{-3}$) is ~14 times larger than that in the presence of VL ($9.0 \times 10^{-4} \pm 4.0 \times 10^{-4}$), suggesting that nitrate-mediated photo-oxidation of GUA is more efficient than that by photosensitized reactions of VL (see Text S7 for the more details).

Line 619: The oxidation of guaiacol, a non-carbonyl phenol, by photosensitized reactions of vanillin was also shown to be  less efficient than that by nitrate photolysis products based on its lower apparent quantum efficiency.

2. Line 226: The authors at several points claim that VL triplet states and nitrate photolysis products have a "synergistic effect," but evidence in support of this claim is lacking, or at best the evidence supporting it is not adequately explained. The inadequately supported claim is repeated in line 497.

Response: Thank you for this point. The trends in line 297 (formerly 226) pertain to nitrate enhancing the increased normalized abundance of products and formation of more oxidized aqSOA from VL photo-oxidation in the presence of $O_2$ (VL* and VL+AN under air-saturated conditions) at pH 4, suggesting a potential enhancement of VL nitration in the presence of $O_2$. This has been revised as follows:

Lines 295–300: In brief, the presence of  $^3$ $O_2$ increased the normalized abundance of products and promoted the formation of more oxidized aqSOA.  Compared to $N_2$-saturated condition, the higher normalized abundance of nitrogen-containing products under air-saturated condition for VL+AN (at pH 4) suggests a potential enhancement of VL nitration in the presence of $O_2$.

Line 606–611:  In contrast, $^3$VL*-initiated reactions proceeded rapidly under air-saturated conditions ($O_2$ is present) as indicated by higher VL decay rate constant and increased normalized abundance of products. For pH 4 experiments, the presence of both $O_2$ and nitrate resulted in the highest normalized abundance of products (including N-containing compounds) and $<OS_c>$, which may be due to the presence of $O_2$ promoting VL nitration.

3. Line 258: This explanation of opposite pH trends at 0.1 and 0.005 mM VL is extremely speculative.

Response: Smith et al. (2016) reported that the relative reactivities of the protonated and neutral forms of VL triplets depend on the VL concentration. Specifically, they found that the quantum yield for direct VL photodegradation is higher at pH 5 than at pH 2 for 0.005 mM VL, but they are not statistically different for 0.03 mM VL (Smith et al., 2016). The opposite trend observed in our study may then be due to this concentration dependence of the reactivities of the protonated and neutral forms of $^3$VL*. Moreover, as mentioned in our response to major comment #6 by Reviewer 1, the comparable pH dependence of the aqSOA formed from VL* at pH 4 and 2.5 over a range of pH conditions from 1.5 to 10.5 suggests that the pH dependence observed is likely due to the acid-base chemistry of the reactions which may involve $^3$VL* or the excimer of VL (Smith et al., 2016). Please see our response to your major comment #4 for relevant changes made in the text.

4. Line 272: For greater clarity, it would be helpful if the manuscript would always match product formulas mentioned in the text to the structures shown in Table S3. Is this product structure #21 in Table S3?

Response: We thank the reviewer for this suggestion. We have added the product structures (if applicable) to the formulas mentioned in the text. No, product structure #21, now #12, refers to a GUA tetramer that was observed only in the GUA+VL experiment (Line 544). Unfortunately, we do not have a product structure for the tetramer mentioned in Line 358 (formerly 272), although we have now added the proposed formula for this tetramer as follows:

Line 358: Furthermore, a tetramer ($C_{31}H_{24}O_{11}$) was observed only in VL* at pH 2.5.

5. Line 297: is this dimer product structure #5 in Table S3?

Response: Sentences related to •OH scavengers have been deleted from the original manuscript.

6. Line 334: The solvent cage effect explanation seems questionable. Why would two negatively charged ions share a solvent cage, given their electrostatic repulsion? Furthermore, in line 339 the authors state that "NaBC did not cause any substantial change in the decay of VL," thus making this whole solvent cage discussion irrelevant to the data at hand.

Response: Sentences related to •OH scavengers have been deleted from the original manuscript.

7. Line 341 – 346: the authors state that "no tetramers were observed in VL*+NaBC" and "VL+AN+IPA had more oligomers," and then go on to suggest that the formation of oligomers can be promoted by inorganic ions, likely via the generation of radicals such as .CO3. No evidence has been provided, as far as I can tell, that NaBC promotes oligomer formation, so I was confused by the authors' claim here that bicarbonate does in fact promote oligomer formation via .CO3 radicals.

Response: Sentences related to •OH scavengers have been deleted from the original manuscript.

8. Line 363: ESI-MS is routinely used to detect macromolecules in biochemistry. This suggestion that the method cannot detect molecules with more than 25 carbons is an erroneous conclusion to draw from Lin et al. (2018).

Response: Lin et al. (2018) also studied BrC from BBOA, similar to our experiments. As they reported that majority of ESI-detected compounds in their study are smaller molecules with fewer than 25 carbon atoms, we raise the possibility that this may also be the case for our study.

9. Line 379: The logic needs to be better spelled out here. Why is the formation of more oxidized products suggested by a larger fraction of small-mass products observed for 1:1 VL/nitrate mixtures compared to 1:100? Do small product masses imply fragmentation, or is there a competition with oligomerization?

Response: We apologize for the confusion. The larger fraction of small product masses (< 200 m/z) observed for 1:1 compared to 1:100 VL/nitrate suggests the prevalence of functionalization for the former. In addition, the higher $<OS_c>$ for 1:1 VL/nitrate indicates the formation of more oxidized products compared to 1:100 VL/nitrate. The small product masses (< 200 m/z) imply functionalization, while the contribution of > 300 m/z suggests oligomerization. Fragmentation was indicated by the observed small organic acids (analyzed using IC), but not the mass spectrometric analyses as the small organic acids are not detectable in the positive ion mode. Fragmentation, which leads to the decomposition of previously formed oligomers and generation of small, oxygenated products such as organic acids (Huang et al., 2018), may also occur for the low [VL] experiments. However, its importance would likely be observed at longer irradiation times, similar to the high [VL] experiments. Oligomerization probably occurred for both 1:1 and 1:100 VL/nitrate as suggested by the observed contribution of > 300 m/z, although to a lesser extent than functionalization based on the higher contribution of < 200 m/z. Lines 466–485 have been revised as follows:

Lines 466–485: For both VL* and VL+AN, the contribution of < 200 m/z to the normalized abundance of products was higher at low [VL] than at high [VL], while the opposite was observed for > 300 m/z (Fig. S13d). This indicates that functionalization was favored at low [VL], as supported by the higher $<OS_c>$, while oligomerization was the dominant pathway at high [VL], consistent with more oligomers or polymeric products reported from high phenols concentration (e.g., 0.1 to 3 mM) (Li et al., 2014; Slikboer et al., 2015; Ye et al., 2019). This is probably due to an increased concentration of phenoxy radicals (in resonance with a carbon-centered cyclohexadienyl radical) at high [VL], promoting radical-radical polymerization (Sun et al., 2010; Li et al., 2014).  At low [VL], the contribution of < 200 m/z to the normalized abundance of products was higher for 1:1 than 1:100 VL/nitrate molar ratio,  suggesting the prevalence of functionalization for the former .  In addition, 1:1 VL/nitrate (A11; Table S2) had higher $<OS_c>$ than 1:100 VL/nitrate (A12; Table S2), indicating the formation of more oxidized products, but had fewer N-containing compounds compared to the latter. A possible explanation is that at 1:1 VL/nitrate, VL may  compete with $NO_2^-$ for $^\bullet OH$ (from nitrate or nitrite photolysis, Reaction 4; Table 1) and indirectly reduce $^\bullet NO_2$. Similarly, hydroxylation has been suggested to be  more

important pathway for 1:1 VL/nitrite than in 1:10 VL/nitrite (Pang et al., 2019a).  Fragmentation, which leads to the decomposition of previously formed oligomers and generation of small, oxygenated products such as organic acids (Huang et al., 2018), may also occur for the low [VL] experiments. However, its importance would likely be observed at longer irradiation times, similar to the high [VL] experiments.

10. Line 389: $C_8H_9NO_3$ should be identified as product structure #2 (an amine) on Table S3.

Response: Thank you for this suggestion. We have now added the structure number to $C_8H_9NO_3$ in line 494 (formerly 389) as follows:

Line 494: … $C_8H_9NO_3$ (No. 2, Table S2) was also…

11. Line 408: The nitrate photolysis explanations may not be needed, given that the observed enhancement of nitrate on guaiacol decay rates was only a factor of 1.2. Is this a statistically significant change?

Response: Thank you for pointing this out. The reviewer is correct that the nitrate photolysis explanation is not needed in this case, given that the observed enhancement for GUA+AN is not statistically significant. Changes in the text are as follows:

Line 510: The enhancement of GUA decay rate constant in the presence of VL is statistically significant ($p < 0.05$), while that in the presence of AN is not ($p > 0.05$).

12. Line 418: The word "Similarly" is being used to relate two seemingly dissimilar observations, causing needless confusion. In the previous sentence, VL shows much higher absorbance enhancement than nitrate, but in this sentence nitrate is being compared to an experiment without nitrate.

Response: The reviewer is correct. The word 'similarly' has been removed in Line 525 (formerly 418) to avoid confusion.

Line 525:  Yang et al. (2021) also observed greater light absorption during nitrate-mediated photo-oxidation relative to direct GUA photodegradation.

13. Line 471: This sentence is confusing. Doesn't this work address (among other things) the effects of nitration on triplet-generating aromatics?

Response: Thank you for catching this. We have clarified this sentence as follows:

Line 574: Although nitrate did not substantially affect the VL decay rate constants likely due to much higher molar absorptivity of VL than nitrate and high VL concentration used in this work, the presence of nitrate promoted functionalization and nitration, indicating the significance of nitrate photolysis in this aqSOA formation pathway.  This work demonstrates that nitration, which is  an important process for producing light-absorbing organics or BrC (Jacobson, 1999; Kahnt et al., 2013; Mohr et al., 2013; Laskin et al., 2015; Teich et al., 2017; Li et al., 2020),  can also affect the aqueous-phase processing of triplet-generating aromatics .

14. Line 481: Why would VL photodegrade 10 times slower in ALW relative to dilute cloudwater? This effect is important for applying this work to the atmosphere. Could the authors provide some theory or explanation here?

Response: There is no generalization yet for these increased or decreased photodegradation of methoxyphenols and by far, only VL has been observed to exhibit decreased photodegradation in ALW. The study for VL in ALW (Loisel et al., 2021) stated that the nature of inorganic ions would affect the photodegradation of organic compounds. Further work on the effects of inorganic ions on photodegradation of VL in ALW is warranted.

15. (a) On Table S2, experiments without nitrate are listed as "—" in the column of normalized abundances of N-containing compounds. Is this because no N-containing compounds were detected in the top 50, or because these samples were not analyzed for N-containing compounds? (b) It would be helpful to map the reactant molecule onto the Figure S12 graph.

Response: (a) The reviewer is right. The samples for experiments without nitrate were not analyzed for N-containing compounds. This information has been added to the footnote of Table S2 as follows:

...[d]The normalized abundance of products was calculated using Eq. 2. The samples for experiments without nitrate (marked with N/A '–') were not analyzed for N-containing compounds.

(b) VL has now been added to Fig. S12, now S11.

**Technical Corrections:**

1. Line 349: "increased" should be "increase"

Response: Sentences related to •OH scavengers have been deleted from the original manuscript.

2. Line 377: "an important" should be "a more important"

Response: Thank you for pointing this out. Line 478 (formerly 377) has been revised accordingly:

Line 478: Similarly, hydroxylation has been suggested to be  more important pathway for 1:1 VL/nitrite than in 1:10 VL/nitrite (Pang et al., 2019a).

3. Line 459: "decompose" should be "decomposes"

Response: Sentences related to •OH scavengers have been deleted from the original manuscript.

4. Sodium nitrate in my opinion would be better abbreviated "NaN" to be more consistent with other abbreviations such as "NaBC."

Response: Sentences related to •OH scavengers have been deleted from the original manuscript.

5. Table S3:  Compound number 4, the most abundant product in some studies, is missing an oxygen atom.  It should be clarified that structure #1 is the reactant molecule vanillin rather than a product.

Response: Thank you for the correction and suggestion. We have corrected these on Table S2, formerly S3.

[revised manuscript text omitted]

---

## Author Comment (AC3)

Author Response for "Aqueous SOA formation from the photo-oxidation of vanillin: Direct photosensitized reactions and nitrate-mediated reactions" by Mabato et al.

We thank the reviewer for the thorough review and many constructive comments that helped improve the manuscript. Our point-by-point responses are below (changes to the original manuscript text and supporting information are in red, moved content in double-line strikethrough, and removed content in strikethrough). Please note that the line numbers in the responses refer to our revised manuscript with tracked changes. Also, please note that because we restructured the manuscript, the numbering of some figures and tables in the revised manuscript is different from those in the original manuscript.

**Reviewer 3**

This study investigated the aqueous photo-oxidation of vanillin (VL) via both direct photosensitized reaction and nitrate-mediated photo-oxidation and discussed the influence of secondary oxidants from triplet excited states ($^3$VL*), solution pH, VOCs, and inorganic anions, etc. in detail. The experiments and data analysis are well done, and the mechanisms that are proposed are plausible. This study provides valuable information about the chemical composition, optical properties, and possible reaction mechanisms for SOA formed from the VL photo-oxidation under different conditions. However, there are a few major and minor comments I would like the authors to address before it is considered for publication in ACP.

**Major comments**

(1) With the experiment design, it is difficult to directly compare $^3$VL* pathway and nitrated-mediated pathway, as also mentioned by the authors that the VL concentration was very high, and $^3$VL* chemistry dominated in all the VL + ammonium nitrate (AN) experiments. Maybe more precisely, what was compared was photo-oxidation of VL via $^3$VL* chemistry with and without nitrate. However, both the title and some places in the manuscript are misleading.

Response: We agree with the reviewer that $^3$VL* chemistry dominated in VL+AN experiments, as mentioned throughout the manuscript. However, we did not intend to compare $^3$VL* and nitrated-mediated pathways either. It is our opinion that the current title does not have such connotation. Despite the high VL concentration, the VL+AN experiments showed some differences in the aqSOA formed. Hence, we would like to keep the current title. With the revised abstract, we hope it will not confuse readers into expecting a comparison.

(2) I suggest the authors restructure the manuscript: on the one hand, to move part of the figures and tables from the SI to the manuscript, e.g. Table S2 and Figure S12, to make it easier to follow. On the other hand, to simplify the article by cutting some "maybe interesting" but not that important/related findings/discussions to make the main storyline clearer.

Response: Thank you for the suggestion. Speculative discussions have been removed from the revised text. First, Section 3.1.1 has been amended to emphasize the importance of VL triplets:

Line 187:  VL photo-oxidation under $N_2$ and air-saturated conditions

Lines 188–240:  The photo-oxidation of VL  under both $N_2$- and air-saturated conditions (Fig. S3a) were carried out at pH 4, which is representative of moderately acidic aerosol and cloud pH values (Pye et al., 2020). No significant VL loss was observed for dark experiments. The oxidation of ground-state VL by $^3VL^*$ via H-atom abstraction or electron transfer can form phenoxy (which is in resonance with a carbon-centered cyclohexadienyl radical that has a longer lifetime) and ketyl radicals (Neumann et al., 1986a, 1986b; Anastasio et al., 1996). The coupling of phenoxy radicals or phenoxy and cyclohexadienyl radicals can form oligomers as observed for both $N_2$- and air-saturated experiments (see discussions later). However, the little decay of VL under $N_2$-saturated condition indicates that these radicals probably predominantly decayed via back-hydrogen transfer to regenerate VL (Lathioor et al., 1999). A possible explanation for this is the involvement of $O_2$ in the secondary steps of VL decay. For instance, a major fate of the ketyl radical is reaction with $O_2$ (Anastasio et al., 1996). In the absence of $O_2$, radical formation occurs, but the forward reaction of ketyl radical and $O_2$ is blocked, leading to the regeneration of VL as suggested by the minimal VL decay. Aside from potential inhibition of secondary oxidants generation (Chen et al., 2020), $N_2$ purging may have also hindered the secondary steps for VL decay.

 Contrastingly, the VL* decay rate constant under air-saturated conditions was 4 times higher. As mentioned earlier, secondary oxidants ($^1O_2$, $O_2^{\bullet-}$/$^{\bullet}HO_2$, $^{\bullet}OH$) can be generated from $^3VL^*$ when $O_2$ is present (e.g., under air-saturated conditions). However, the photo-oxidation of VL in this study is likely mainly governed by $^3VL^*$ and that these secondary oxidants have only minor participation.  $^1O_2$ is also a potential oxidant for phenols (Herrmann et al., 2010; Minella et al., 2011; Smith et al., 2014), but $^1O_2$ reacts much faster (by ~60 times) with phenolate ions compared to neutral phenols (Tratnyek and Hoigne, 1991; Canonica et al., 1995; McNally et al., 2005). Under the pH values (pH 2.5 to 4) considered in this study, the amount of phenolate ion is negligible, so the reaction between VL and $^1O_2$ should be slow. Interestingly, however, $^1O_2$ has been shown to be important in the photo-oxidation of 4-ethylguaiacol ($pK_a$ = 10.3) by $^3C^*$ of 3,4-dimethoxybenzaldehyde (solution with pH of ~3) (Chen et al., 2020). Furthermore, while the irradiation of other phenolic compounds can produce $H_2O_2$, a precursor for $^{\bullet}OH$ (Anastasio et al., 1996), the amount of $H_2O_2$ is small. Based on this, only trace amounts of $H_2O_2$ were likely generated from VL* (Li et al., 2014) under-air saturated conditions, suggesting that contribution from $^{\bullet}OH$ was minor. Overall, these suggest that VL photo-oxidation

in this study is driven by $^3$VL*. Further study on the impact of $O_2$ on the reactive intermediates involved is required to understand the exact mechanisms occurring under air-saturated conditions. Nonetheless, the VL* decay trends clearly indicate that $O_2$ is important for efficient VL photo-oxidation The decay rate constant for VL+AN under air-saturated conditions was also higher  (6.6 times) than $N_2$-saturated conditions which may  be due to  reactions facilitated by nitrate photolysis products  that may have been enhanced in the presence of $O_2$  (Vione et al., 2005; Kim et al., 2014; Pang et al., 2019a). As shown later, more nitrogen-containing species were observed under air-saturated conditions. An example is enhanced VL nitration likely from increased $^•NO_2$ formation such as from the reaction of $^•OH$ and $O_2^{•-}$ with $NO_2^-$ (Reactions 4 and 5, respectively; Table 1) or the autoxidation of $^•NO$ from $NO_2^-$ photolysis (Reactions 6–9; Table 1) in aqueous solutions (Pang et al., 2019a). ~~Reactions involving $^•HO_2/O_2^{•-}$ which may originate from the photolysis of nitrate alone, likely from the production and subsequent photolysis of peroxynitrous acid (HOONO) (Reaction 10; Table 1) (Jung et al., 2017; Wang et al., 2021), or the reactions of $^3$VL* in the presence of $O_2$, may have contributed as well. For instance, Wang et al. (2021) recently demonstrated that nitrate photolysis generates $^•HO_2/O_2^{•-}$ (aq) and HONO(g) in the presence of dissolved aliphatic organic matter (e.g., nonanoic acid, ethanol), with the enhanced HONO(g) production caused by secondary photochemistry between $^•HO_2/O_2^{•-}$ (aq) and photoproduced NO$_{x(aq)}$ (Reactions 11 and 12; Table 1), in agreement with Scharko et al. (2014). The significance of this increased HONO production is enhanced $^•OH$ formation (Reaction 13; Table 1). In addition, $^•HO_2$ can react with $^•NO$ (Reaction 10; Table 1) from $NO_2^-$ photolysis (Reaction 6; Table 1) to form HOONO, and eventually $^•NO_2$ and $^•OH$ (Reaction 14; Table 1) (Pang et al., 2019a).s~~ constants for VL* and VL+AN imply that $^3$VL* chemistry still dominates even at 1:10 molar ratio of VL/nitrate, probably due to the much higher molar absorptivity of VL compared to that of nitrate (Fig. S1) and the high VL concentration (0.1 mM) used in this study. Although we have no quantification of the oxidants in our reaction systems as it is outside the scope of this study, these observations clearly substantiate that photosensitized oxidation of VL and nitrate-mediated VL photo-oxidation are more efficient in the presence of $O_2$.

Moreover, Section 3.1.3 (Effect of VOCs and inorganic anions) and related sentences have also been deleted based on the likely minor contribution of $^•OH$ to VL photo-oxidation in this study as pointed out by Reviewer 1 and suggested by other published literature (Anastasio et al., 1996; Li et al., 2014). Also, Section 3.5 has been deleted and Fig. 4, now 2, has been shown for the first time when potential aqSOA formation pathways were discussed, then referred to throughout the text. Table 1, which was reduced to half, was maintained in the revised version, while Table S2 (now 2) was moved to the main text, as suggested by the reviewer.

(3) (a) It is very interesting to see the changes in optical properties, and their relation to the changes in chemical composition. However, I only see very general discussions about it (e.g. line 234-238 and line 282-289). It will be nice to discuss the specific compounds, possible chromophores, and to explain the changes in the optical properties.

(b) To explain the pH-dependency, the authors cited Pang et al. 2019a, which reported the pH-dependent light absorbance of nitrophenols. However, the dominating products in this study were those without N, different from those in Pang et al. 2019a. In addition, the chemical composition of SOA with pH 4 and pH < 4 are quite different, which could also lead to different functional group/chromophores, and changes in optical properties.

Response:

(a) Thank you for this suggestion. We agree that identifying the BrC chromophores would enrich the discussion for the changes in the optical properties. However, it is possible that the products detected using UHPLC-qToF-MS in positive ESI mode might not have contributed significantly to all products formed and hence may not be the primary contributors to the absorbance enhancement. In other words, the absorbance enhancement may not necessarily correlate directly with the products detected. Detailed characterization of specific chromophores is indeed interesting, but it is outside of the scope of this study. Instead, the changes in the optical properties for this study are based on the integrated area of the absorption spectra from 350 to 550 nm in order to consider the contributions of all potentially light-absorbing products.

(b) The reviewer is correct that the dominant products in this study do not contain nitrogen. We have revised this explanation based on the comparable pH dependence of the aqSOA formed from VL* at pH 2.5 and 4 over a range of pH conditions from 1.5 to 10.5. This suggests that the observed pH dependence is due to the acid-base chemistry of the reactions, which may involve $^3$VL* or the excimer of VL (Smith et al., 2016). Changes in the text are as follows:

Lines 375–386: Higher absorbance enhancement for both VL* and VL+AN (Fig. 3b) was observed as pH increased . To determine whether the pH dependence is due to the acid-base chemistry of the products or of the reactions, we measured the pH dependence of the aqSOA formed from VL* at pH 4 and 2.5 over a range of pH conditions from 1.5 to 10.5 (Fig. S10). For both cases, the intensity of absorption at longer wavelengths significantly increased as the pH of the solutions was raised. Moreover, the comparable pH dependence of the two solutions suggest that the observed pH dependence may be attributed to the acid-base chemistry of the reactions, which may involve $^3$VL* or the excimer of VL (Smith et al., 2016), as discussed earlier. ~~When a phenolic molecule deprotonates at higher pH, an ortho- or para- electron withdrawing group, such as a nitro or aldehyde group, can attract a portion of the negative charge towards its oxygen atoms through induced and conjugated effects, leading to the extension of chromophore from the electron-donating group (e.g., -O⁻) to the electron-withdrawing group via the aromatic ring (Carey, 2000; Williams and Fleming, 2008; Pang et al., 2019a). Hence, the delocalization of the negative charge in phenolates leads to significant redshifts (Mohr et al., 2013).~~

For reference, changes in the revised text that are relevant to lines 375–386 are shown below:

Lines 331–352: The decay rates constants for both VL* and VL+AN increased as pH decreased (VL* and VL+AN at pH 2.5: 1.65 and 1.43 times faster than at pH 4, respectively) (Fig. S3b). These differences in decay rate constants are small but statistically significant ($p < 0.05$). The p$K_a$ for the VL triplet has been reported to be 4.0 (Smith et al., 2016). As there are a greater fraction of VL triplets that are protonated at pH 2.5 (0.96) than at pH 4 (0.5), it is possible that the pH dependence of the decay rate constants observed in this study is due to $^3$VL* being more reactive in its protonated form. Smith et al. (2016) also observed a pH dependence for the direct photodegradation of VL (0.005 mM) (rate constants at pH ≤ 3 are ~two times lower than at pH ≥ 5) which they attributed to the sensitivity of the excimer of VL (i.e., the charge-transfer complex formed between an excited state VL molecule and a separate ground state VL molecule; Birks, 1973, Turro et al., 2010) to acid-base chemistry. The opposite trend observed in this study for 0.1 mM VL may be due to the reactivities of the protonated and neutral forms of the $^3$VL* being dependent on the VL concentration (Smith et al., 2016). For VL*, this pH trend indicates that $^3$VL* are more reactive in their protonated form, which is opposite to that reported for 0.005 mM VL (Smith et al., 2016), likely due to the concentration dependence of the relative reactivities of protonated and neutral forms of $^3$VL*. It has been reported that the quantum yield for direct VL photodegradation is higher at pH 5 than at pH 2 for 0.005 mM VL, but they are not statistically different for 0.03 mM VL (Smith et al., 2016). Also, increases in hydrogen ion concentration can enhance the formation of $HO_2^{\bullet}$ and $H_2O_2$ and in turn, $^{\bullet}$OH formation (Du et al., 2011). In addition to these pH influences on VL*, the dependence of N(III) ($NO_2^- + HONO$) speciation on solution acidity (Pang et al., 2019a) also contributed to the observed pH effects for VL+AN. At pH 3.3, half of N(III) exists as HONO (Fischer and Warneck, 1996; Pang et al., 2019a), which has a higher quantum yield for $^{\bullet}$OH formation than that of $NO_2^-$ in the near UV region (Arakaki et al., 1999; Kim et al., 2014). The increased $^{\bullet}$OH formation rates as pH decreases can lead to faster VL decay (Pang et al., 2019a). Also, $NO_2^-$/HONO can generate $^{\bullet}NO_2$ via oxidation by $^{\bullet}$OH (Reactions 4 and 15; Table 1) (Pang et al., 2019a). As pH decreases, the higher reactivity of $^3$VL* and sensitivity of the excimer of VL to acid-base chemistry HONO being the dominant N(III) species can lead to faster VL photo-oxidation may have led to faster VL photo-oxidation.

(4) Adding the experiments of guaiacol (GUA) is a little bit confusing, as the title is the photo-oxidation of VL. I understand it is a good addition to the manuscript, and these experiments nicely compared the photo-oxidation of GUV via the two pathways. However, the conclusion (line 25-26) "guaiacol oxidation by photosensitized reactions of VL was observed to be more efficient relative to nitrate-mediated photo-oxidation" is still problematic, as the concentration of VL in GUA + VL experiment was still 10 times higher than the observed value in the cloud and fog but the concentration of AN in GUA + AN experiments was similar to the observed concentration.

Response: Thank you for this suggestion. The apparent quantum efficiency of GUA photodegradation ($\phi_{GUA}$) in the presence of either VL or nitrate during simulated sunlight illumination can be defined as (Anastasio et al., 1996; Smith et al., 2014, 2016):

$$\Phi_{GUA} = \frac{\text{mol GUA destroyed}}{\text{mol photons absorbed}} \qquad \text{(Eq. S9)}$$

$\Phi_{GUA}$ was calculated using the measured rate of GUA decay and rate of light absorption by either VL or nitrate through the following equation:

$$\Phi_{GUA} = \frac{\text{rate of GUA decay}}{\text{rate of light absorption by VL or nitrate}} = \frac{k'_{GUA} \times [GUA]}{\sum[\left(1 - 10^{-\varepsilon_\lambda[C]l}\right) \times I'_\lambda]} \qquad \text{(Eq. S10)}$$

where $k'_{GUA}$ is the pseudo-first-order rate constant for GUA decay, [GUA] is the concentration of GUA (M), $\varepsilon_\lambda$ is the base-10 molar absorptivity (M$^{-1}$ cm$^{-1}$) of VL or nitrate at wavelength $\lambda$, [C] is the concentration of VL or nitrate (M), $l$ is the pathlength of the illumination cell (cm), and $I'_\lambda$ is the volume-averaged photon flux (mol-photons L$^{-1}$ s$^{-1}$ nm$^{-1}$) determined from 2NB actinometry:

$$j(2NB) = 2.303 \times \Phi_{2NB} \times l \times \sum_{300\,nm}^{350\,nm} (\varepsilon_{2NB,\lambda} \times I'_\lambda \times \Delta\lambda) \qquad \text{(Eq. S11)}$$

The $\phi_{GUA}$ in the presence of nitrate ($1.3 \times 10^{-2} \pm 2.9 \times 10^{-3}$) is ~14 times larger than that in the presence of VL ($9.0 \times 10^{-4} \pm 4.0 \times 10^{-4}$), suggesting that nitrate-mediated photo-oxidation of GUA is more efficient than that by photosensitized reactions of VL. We have revised this in the text as follows and added the information shown above in the supporting information: **Text S7**. Estimation of the apparent quantum efficiency of guaiacol photodegradation.

Line 29: Furthermore, comparisons of the apparent quantum efficiency of guaiacol photodegradation indicate that in this study, guaiacol oxidation by photosensitized reactions of VL  is less efficient relative to nitrate-mediated photo-oxidation.

Other relevant revisions in the text are as follows:

Line 514: As mentioned earlier,  $^3$VL* chemistry appears to be more important than that of nitrate photolysis even at 1:10 molar ratio of VL/nitrate on account of the much higher molar absorptivity of VL compared to that of nitrate (Fig. S1) and the high VL concentration (0.1 mM) used in this study. However, the apparent quantum efficiency of GUA photodegradation ($\phi_{GUA}$) in the presence of nitrate ($1.3 \times 10^{-2} \pm 2.9 \times 10^{-3}$) is ~14 times larger than that in the presence of VL ($9.0 \times 10^{-4} \pm 4.0 \times 10^{-4}$), suggesting that nitrate-mediated photo-oxidation of GUA is more efficient than that by photosensitized reactions of VL (see Text S7 for the more details).

Line 619: The oxidation of guaiacol, a non-carbonyl phenol, by photosensitized reactions of vanillin was also shown to be  less efficient than that by nitrate photolysis products based on its lower apparent quantum efficiency.

**Minor comments**

1. Line 27-28 In the abstract, the sentence "which nitrate photolysis products can further enhance" sounds not clear to me.

Response: We apologize for the confusion. This sentence was supposed to convey two things: a) the direct photosensitized oxidation of VL may be an important aqSOA source and b) the addition of nitrate photolysis products to (a) can initiate further reactions that can enhance aqSOA formation. We have revised this sentence as follows:

Line 32: This study indicates that the direct photosensitized oxidation of VL, and nitrate-mediated photo-oxidation of VL  may be  important aqSOA sources in areas influenced by biomass burning emissions.

2. Line 121 Did you average these replicates for mass spectra and/or decay rates? Please clarify it.

Response: Yes, the reported mass spectra are based on the average of results from duplicate experiments, while the decay rate constants and absorbance enhancement are average of results from triplicate experiments. We have added the following sentence in the methods to clarify this:

Line 132: Each experiment was repeated independently at least three times and measurements were done in triplicate. The reported decay rate constants and absorbance enhancement are the average of results from triplicate experiments, and the corresponding errors represent one standard deviation.

The footnote of Table S2, now 2, has been revised as follows:

Table 2: [b]The data fitting was performed in the initial linear region. Each value is the average of results from triplicate experiments. Errors represent one standard deviation.

3. Line 168-169 It would be nice to explain it together with the chemical composition shown in Figure 1.

Response: This paragraph discusses the VL decay trends. Line 249 (formerly 196) already includes an example for this ($C_{16}H_{10}N_2O_9$; No. 3, Table S2). Changes in the text are as follows:

Lines 220: The decay rate constant for VL+AN under air-saturated conditions was also higher  (6.6 times) than $N_2$-saturated conditions, which may  be due to  reactions facilitated by nitrate photolysis products  that may have been enhanced in the presence of $O_2$  (Vione et al., 2005; Kim et al., 2014; Pang et al., 2019a).

Lines 247–254: Oligomers,  functionalized monomers (e.g., demethylated VL; Fig. S4), and nitrogen-containing compounds (e.g., $C_{16}H_{10}N_2O_9$; No. 3,Table S2) (for VL+AN) had higher normalized abundance  under air-saturated conditions  (Figs. 1c-d), likely due to efficient  $^3$VL*-initiated oxidation and enhanced VL nitration in the presence of  $O_2$  For both VL* and VL+AN under air-saturated conditions, the most abundant product was $C_{10}H_{10}O_5$ (No. 4, Table S2), a substituted VL.

4. Line 181 "VL*" should be "$^3$VL*"?

Response: Thank you for catching this error. This has been corrected as follows:

Line 235: Nevertheless, the comparable decay rate constants for VL* and VL+AN imply that $^3$VL* chemistry

5. Line 187 In both VL* and VL + AN under N2-saturated conditions (Fig. 1(a) and (b)), trimer signals are very high. Any explanations?

Response: $N_2$-saturated experiments would inhibit the formation of secondary oxidants, which can lead to $^3$VL*-driven reactions (Chen et al., 2020) (line 107). Compared to •OH-mediated oxidation which yields more functionalized/oxygenated products, triplet-driven oxidation has been suggested to produce higher molecular weight products, probably with less fragmentation (Yu et al., 2014; Chen et al., 2020), as mentioned in lines 241–243. This likely explains the prevalence of dimers and trimers for the $N_2$-saturated experiments.

6. Line 212 Could you give some numbers to show "majority"?

Response: It is 58% of the 50 most abundant products for experiments A5 to A8. This has been added to the text as follows:

Lines 277–279: For experiments A5 to A8,  H:C ratios were mostly around 1.0 and double bond equivalent (DBE) values were typically (58% of the 50 most abundant products) > 7, indicating that the products (Table S2) were mainly oxidized aromatic  compounds (Xie et al., 2020).

7. Line 255 Should it be pH 4?

Response: Thank you for catching this error. The text has now been corrected as follows:

Line 331: the range of 2.5 to 4

**References**

[revised manuscript text omitted]

---

## Referee Report (RR1)

Mobato et al., studied the aqueous-phase photochemistry of vanillin (VN) in cloud/for relevant conditions leading to aqSOA. They conducted two types of photochemistry, photolysis of VN and nitrate-mediated VN (irradiation of VN + Ammonium Nitrate mixture) and analyzed VN decay, product mass spectra, O2/pH/ammonium effects, and UV spectra for light absorbance of products. They indeed conducted full analyses of aqSOA chemistry. However, novel findings with compelling experimental evidence in this manuscript are not well written. Instead, their arguments are mostly speculative:

1. The purpose of conducting two types of photochemistry is not clear. Didn't you compare the photolysis and OH-radical reactions of VN? Then, it is not clear how much of OH radical was generated in their condition. According to Fig. S3 and also the main text, photochemistry by OH radical is negligible. Then, why not set the concentration of OH radical relevant to cloud/fog condition adjusting nitrate concentration, which can be obtained by a kinetic model based on decay rate of VN and table 1? If the purpose is to study the nitration, then why was ammonium nitrate (AN) used? Ammonium clearly complicates the system. Since authors used sodium nitrate (SN), why did not authors conduct explicit comparison between AN and SN that could lead to chemical insights in the aqueous phase?

2. The explicit mechanisms that could support their arguments are lacking. Fig. 2 is oversimplified. Their arguments about explaining chemistry were heavily based on chemical mechanisms, but the arguments were not convincing because authors did not propose explicit mechanisms—radical-based full mechanisms containing steps in details.

3. Most of the discussions sound speculative (words like "may, probably" were used often). Experimental data and analyses do not seem to support their arguments. Although they attempt to show their results to be consistent with others previous studies, it is difficult to find the novelty of this manuscript.

4. Therefore, substantial revisions and restructure are recommended for publication.

Line 55: There needs to be a reason why NO3- in the aqueous phase is the source of nitration in this work. In most of cases, nitration (organonitrate formation) occurs in the gas phase. Why your aqueous phase study is suitable instead of multiphase experiments.

Line 100-101: I don't understand this statement. Shouldn't you want nitrates to affect the kinetics of VN photolysis by generating sufficient OH radicals? The purpose of adding ammonium nitrate should be clearly stated.

Line 126-127: I disagree the equal ionization efficiency for different compounds. This cannot be true.

Line 133-138: The term, "normalized abundance of products" is scientifically meaningless, unless ionization efficiencies for each product were taken into account.

Line 149-156: A schematic containing radical-based full mechanisms is required. This should be Fig. 2, which is currently oversimplified.

Line 179-181: Building a kinetic mode based on Table 1 is recommended. You can verify it by simulating a kinetic model.

Line 192-194: I do not understand the purpose of conducting N2 experiments in the first place. Clearly, O2 is the oxidant that required for the oxidation. Therefore, you expect better oxidation with O2. There always exists O2 in cloud/fog droplets. This is not a novel finding.

Line 225-227: Negative-mode analysis sounds more suitable for oxidized products. The reason for conducting positive-mode analysis that would bring a benefit and unique results in this work needs to be addressed.

Line 234-237: This statement is weak. You need to make a strong statement based on your compelling evidence.

Line 244-246: It is difficult to conclude that products are mainly conjugated pi system based on your ESI analysis (even MS-MS analysis).

Line 266: The range of pH 2.5 and pH 4 seems too narrow to study a pH effect.

Line 270-271: This is speculative, again. You need to make a strong statement based on your novel analysis.

Line 296-300 & 305-306: Speculative

Line 315: The term, "the 50 most abundant products" provides no scientific meaning. You cannot tell the abundance of products by ESI signal. Besides, why did you choose 50?

Line 330-332: Speculative. Propose an explicit mechanism to support your argument.

Line 350-352: How can you prove this based on your experiments?

Line 352-354: What are the anticipated product based on NH4+ chemistry? What are the corresponding products you have found? How does that related to BrC absorbance or conjugate bond?

Line 366: speculative

---

## Author Response (AR2)

Author Response for "Aqueous SOA formation from the direct photosensitized oxidation of vanillin in the absence and presence of ammonium nitrate" by Mabato et al.

We thank the reviewers for the thorough review and many constructive comments that helped improve the manuscript. The major issues identified during the review are as follows: (1) focus of the study, (2) uncertainties for the normalized abundance of products, and (3) speculative discussions due to the lack of detailed reaction mechanisms and kinetic model analysis. Below are our overall responses to these, followed by our point-by-point responses to the reviewers' comments (changes to the second version of the manuscript and supporting information are in red, moved content in double-line strikethrough, and removed content in strikethrough). Please note that the line numbers in the responses refer to our revised manuscript with tracked changes.

(1) First, we have revised the title to 'Aqueous SOA formation from the direct photosensitized oxidation of vanillin in the absence and presence of ammonium nitrate' to reflect the focus of this study. We have also removed the guaiacol + nitrate experiments to avoid distracting the readers from the main objectives of the paper. The novelty of the paper includes the (a) evaluation of aqSOA formation from the direct photosensitized oxidation of a triplet precursor (vanillin, VL, a phenolic aromatic carbonyl) alone (VL*) and in the presence of ammonium nitrate (VL+AN) based on VL decay kinetics, detected products, and light absorbance changes, (b) investigation of the effects of different reaction conditions (air- vs. $N_2$-saturated experiments, solution pH, and reactants concentration and molar ratios) on aqSOA formation, and (c) assessment of aqSOA formation from the oxidation of guaiacol, a non-carbonyl phenol, via photosensitized reactions of VL.

(2) Moreover, the rationale and purpose of using the normalized abundance of products were already stated in the previous version of the manuscript. The comparisons of peak abundance in mass spectrometry have been used in many recent studies (e.g., Lee et al., 2014; Romonosky et al., 2017; Wang et al., 2017; Fleming et al., 2018; Song et al., 2018; Klodt et al., 2019; Ning et al., 2019) to show the relative importance of different types of compounds (Wang et al., 2021). Also, the assumption of equal ionization efficiency for calculating the normalized abundance of products is commonly used to estimate O:C ratios of SOA (e.g., Bateman et al., 2012; Lin et al., 2012; Laskin et al., 2014; De Haan et al., 2019). In the revised version, we emphasize the uncertainties associated with the normalized abundance of products due to the variability in ionization efficiencies for various compounds. Despite the intrinsic uncertainties, the normalized abundance of products enables a semi-quantitative analysis that gives an overview of how the signal intensities (extracted ion chromatogram peak areas from UHPLC-qToF-MS) changed under different experimental conditions. In addition, the use of the more common 'relative abundance' (product peaks are normalized to the highest peak) (e.g., Lee et al., 2014; Romonosky et al., 2017; Fleming et al., 2018; Klodt et al., 2019) would yield the same major products determined using the normalized abundance of products and maintain the conclusions of this work. This demonstrates the usefulness of the normalized abundance of products for comparisons among different experimental conditions.

(3) Lastly, there are not enough experimental results on the reactive intermediates in this study to provide detailed mechanisms without being speculative and to build a kinetic model. Further work is needed to exclude all other possibilities, and hence we are on the conservative side. Regardless, the discussions in this study were supported by our results and compared with previous studies.

**References**

[revised manuscript text omitted]

**Reviewer 3**

The authors have addressed some of the major comments and questions and the paper has improved upon revision. However, I think this manuscript still fails to address the following important issues. Though the topic is very interesting, this manuscript doesn't tell a very clear and nice story that meets the requirements of ACP.

1. The novelty of this study is that it characterized the product compositions and light absorbance changes. However, it failed to show a link between the observed changes in the chemical composition among different experiments (e.g. more oligomers, functionalized monomers, nitration) and the changes in light absorbance. The authors replied in the final response "However, it is possible that the products detected using UHPLC-qToF-MS in positive ESI mode might not have contributed significantly to all products formed and hence may not be the primary contributors to the absorbance enhancement.", but meanwhile they got the conclusion that "The majority of the most abundant products from both VL photo-oxidation pathways were potential Brown carbon (BrC) chromophores.". In Fig. S11, they showed the major compounds from pH 4 experiments under air-saturated conditions are potential BrC compounds, including both monomers and oligomers, compounds with and without N. However, this doesn't help to explain the differences in light absorption under different experimental conditions (Fig. 3).

Response: Thank you for your comment. The main novelty of this study is the investigation of aqSOA formation from the direct photosensitized oxidation of VL in the absence and presence of ammonium nitrate. The reactions were characterized based on VL decay kinetics, detected products, and light absorbance changes. As mentioned in lines 189-190, 'the major products detected in this study are probably those with high concentration or high ionization efficiency in the positive ESI mode.' Based on this, the products detected using UHPLC-qToF-MS in positive ESI mode might not have contributed significantly to the total products formed and hence may not be the primary contributors to the absorbance enhancement. In other words, the absorbance enhancement may not necessarily correlate directly with the products detected.

Correlating speciated chromophores with absorbance changes may be useful in demonstrating how aqSOA influence the Earth's radiative balance and identifying chemical reactions that can affect the overall light absorption by aqSOA. This can be accomplished by using liquid chromatography coupled with photodiode array (PDA) detector and high-resolution mass spectrometry (LC/PDA/HRMS platform) (e.g., Lin et al., 2017; Jiang et al., 2021; Misovich et al., 2021). In our experiments, VL (and GUA) concentration measurements, product characterization, and absorbance measurements were performed using UHPLC-PDA, UHPLC-qToF-MS, and UV-Vis spectrophotometry, respectively. A similar approach is then possible using the current methods in this work by matching the retention time (RT) of the products detected using UHPLC-qToF-MS with that in the PDA. However, the concentration of the chromophores in this study is below the detection limit of the PDA based on the lack of distinct PDA signals from the products. Nonetheless, the overall light absorbance changes were characterized by integrating the area of absorbance from 350 to 550 nm to incorporate all light-absorbing compounds. Fig. S11 was used to show that majority of the detected products (50 most abundant products) from pH 4 experiments were potential BrC chromophores. The differences in the light absorbance changes under different experimental conditions (Fig. 3) were discussed in lines 321-360 for VL photo-oxidation under $N_2$ and air-saturated conditions, lines 411-418 for VL photo-oxidation under varying pH conditions, and lines 540-543 for oxidation of guaiacol by photosensitized reactions of VL, next to the discussions for detected products and their normalized abundance. Overall, we agree with the reviewer that it would be interesting and valuable to examine and identify the specific BrC chromophores responsible for the changes in the optical properties, unfortunately, this is an experimental limitation for the current work. Changes in the text and clarifications regarding these are as follows:

Lines 325-344: For both VL* and VL+AN, evident absorbance enhancement was observed under air-saturated conditions, while the absorbance changes under $N_2$-saturated conditions were minimal, consistent with the VL decay trends. Dimers and functionalized products have been shown to contribute to chromophore formation for the aqueous photo-oxidation of guaiacyl acetone (another aromatic phenolic carbonyl) by [3]DMB*(Jiang et al., 2021). Based on this, the higher normalized abundance of  oligomers, which have large, conjugated π-electron systems (Chang and Thompson, 2010), and hydroxylated products (Li et al., 2014; Zhao et al., 2015) observed under air-saturated conditions have contributed to the absorbance enhancement. However, it is worth noting that the products detected may not have contributed significantly to the total products formed and hence may not be the primary contributors to the absorbance enhancement. As mentioned earlier, the major products detected in this study are probably those with high concentration or high ionization efficiency in the positive ESI mode. In other words, the absorbance enhancement may not necessarily correlate directly with the products detected.

Correlating speciated chromophores with absorbance changes may be useful in demonstrating how aqSOA influence the Earth's radiative balance and identifying chemical reactions that can affect the overall light absorption by aqSOA. This can be accomplished by using liquid chromatography coupled with photodiode array (PDA) detector and high-resolution mass spectrometry (LC/PDA/HRMS platform) (e.g., Lin et al., 2017; Jiang et al., 2021; Misovich et al., 2021). In our experiments, VL (and GUA) concentration measurements, product characterization, and absorbance measurements were performed using UHPLC-PDA, UHPLC-qToF-MS, and UV-Vis spectrophotometry, respectively. A similar approach is then possible using the current methods in this work by matching the retention time (RT) of the products detected using UHPLC-ToF-MS with that in the PDA. However, the concentration of the chromophores in this study is below the detection limit of the PDA based on the lack of distinct PDA signals from the products.

2. The addition of guaiacol (as another non-carbonyl phenol) oxidation experiments doesn't help a lot with the topic of this manuscript. The major conclusion from this part (as written in the abstract) is "guaiacol oxidation by photosensitized reaction of VL is less efficient relative to nitrate-mediated photo-oxidation." As claimed by the authors in the final response:

"However, we did not intend to compare 3VL* and nitrated-mediated pathways either", why did they compare the efficiency of photosensitized reaction of VL with nitrated-mediated photo-oxidation of guaiacol?

Response: Thank you for pointing this out. We apologize for the confusion. As stated in lines 495-498: 'The oxidation of phenols by $^3C*$ has been mainly studied using non-phenolic aromatic carbonyls (Anastasio et al., 1997; Smith et al., 2014, 2015; Yu et al., 2014; Chen et al., 2020) and aromatic ketones (Canonica et al., 2000) as triplet precursors. Recently, $^3VL*$ has also been shown to oxidize syringol (Smith et al., 2016), a non-carbonyl phenol, although the reaction products remain unknown.' The guaiacol experiments were added to examine the formation of aqSOA from guaiacol oxidation via VL photosensitized reactions. Indeed, we did not intend to compare $^3VL*$ and nitrated-mediated pathways in this study. The primary objective of this study is to investigate aqSOA formation from the direct photosensitized oxidation of VL in the absence and presence of ammonium nitrate. To reflect the focus of the paper, we have revised the title to 'Aqueous SOA formation from the direct photosensitized oxidation of vanillin in the absence and presence of ammonium nitrate'. Also, to avoid distracting the readers from the main objectives of the paper, guaiacol + nitrate experiments have been removed from the manuscript. Changes in the text are as follows:

Abstract: Vanillin (VL), a phenolic aromatic carbonyl abundant in biomass burning emissions, forms triplet excited states ($^3VL*$) under simulated sunlight leading to aqueous secondary organic aerosol (aqSOA) formation. Nitrate and ammonium are among the main components of biomass burning aerosols and cloud/fog water. uUnder atmospherically relevant cloud and fog conditions, solutions composed of either VL only or VL with ammonium nitrate were subjected to simulated sunlight irradiation to compare aqSOA formation via the direct photosensitized oxidation of VL in the absence and presence of ammonium nitrate. The reactions were characterized This direct photosensitized oxidation of VL was compared with nitrate-mediated VL photo-oxidation under atmospherically relevant cloud and fog conditions through by examining the VL decay kinetics, product compositions, and light absorbance changes. The majority of the most abundant products from both VL photo-oxidation pathways were potential Brown carbon (BrC) chromophores. In addition, Bboth pathways conditions generated oligomers, functionalized monomers, and oxygenated ring-opening products, and ammonium but nitrate promoted functionalization and nitration, which can be ascribed likely due to its photolysis products ($^•OH$, $^•NO_2$, and N(III), $NO_2^-$ or HONO). Moreover, a potential imidazole derivative observed from nitrate-mediated VL photo-oxidation in the presence of ammonium nitrate suggested that ammonium may be involvedparticipated in the reactions. Tthe majority of the most abundant products from both conditions were potential Brown carbon (BrC) chromophores. The effects of oxygen ($O_2$), pH, and reactants concentration and molar ratios on VL photo-oxidation the reactions were also explored. Our findings show that $O_2$ plays an essential role in VL photo-oxidation the reactions and oligomer formation was enhanced at pH < 4. Also, functionalization was dominant at low VL concentration, whereas oligomerization was favored at high VL concentration. Furthermore, oligomers and hydroxylated products were detected from the oxidation of comparisons of the apparent quantum efficiency of guaiacol (a non-carbonyl phenol) via VL photosensitized reactions . Lastly, potential aqSOA formation pathways via the direct photosensitized oxidation of VL in the absence and presence of ammonium nitrate  were proposed. This study indicates that the direct photosensitized oxidation of VL  may be an important aqSOA source in areas influenced by biomass burning  and underscores the importance of nitrate in the aqueous-phase processing of aromatic carbonyls.

Lines 89-122: Nitrate and ammonium are also among the main biomass burning aerosol components (Xiao et al., 2020; Zielinski et al., 2020). As BB aerosols are typically internally mixed with other aerosol components (Zielinski et al., 2020), VL may coexist with ammonium nitrate in BB aerosols. The  direct photosensitized oxidation of VL in the absence and presence of ammonium nitrate may then reveal insights into the atmospheric processing of BB aerosols. Moreover, the $^3$C* of non-phenolic aromatic carbonyls (e.g., 3-4-dimethoxybenzaldehyde, DMB; a non-phenolic aromatic carbonyl) (Smith et al., 2014; Yu et al., 2014; Jiang et al., 2021) and phenolic aromatic carbonyls (e.g., acetosyringone, vanillin) (Smith et al., 2016) have been shown to oxidize phenols, but the reaction products from the latter are unknown.

Previous works aqSOA formation via triplet-mediated oxidation are mostly based on reactions between phenols and a non-phenolic aromatic carbonyl as triplet precursor (e.g., Smith et al., 2014; Yu et al., 2014; Jiang et al., 2021). Also, studies examining the effects of inorganic nitrate on aqSOA formation and properties remain limited. The present study aimed to evaluate aqSOA formation via the direct photosensitized oxidation of a triplet precursor (VL) alone. Furthermore, aqSOA formation via the direct photosensitized oxidation of VL in in the presence of ammonium nitrate was also examined. Accordingly, the main goals of this study are (1) to compare aqSOA formation in cloud/fog water via the direct photosensitized oxidation of VL in the absence and presence of ammonium nitrate, (2) to evaluate the influences of $O_2$, solution pH, and reactants concentration and molar ratios on the reactions, (3) to investigate the participation of ammonium in the direct photosensitized oxidation of VL in the presence of ammonium nitrate, and (4) to examine aqSOA formation from the oxidation of guaiacol, a non-carbonyl phenol, via photosensitized reactions of VL. To achieve these goals, solutions composed of either VL only or VL in the presence of ammonium nitrate were subjected to simulated sunlight irradiation under atmospherically relevant cloud and fog conditions. Solutions composed of VL in the presence of sodium nitrate were also examined for comparison with the presence of ammonium nitrate.  The reactions were characterized based on VL decay kinetics, detected products, and light absorbance changes.  these two ~~photo-oxidation pathways were also assessed. The ³C* of non-phenolic aromatic carbonyls (e.g., 3-4-dimethoxybenzaldehyde, DMB; a non-phenolic aromatic carbonyl) (Smith et al., 2014; Yu et al., 2014; Jiang et al., 2021) and phenolic aromatic carbonyls (e.g., acetosyringone, vanillin) (Smith et al., 2016) have been shown to oxidize phenols, but the reaction products from the latter are unknown. We then examined the photo-oxidation of guaiacol, another~~

 . Finally, we proposed aqSOA formation pathways via the direct photosensitized oxidation of VL in the absence and presence of ammonium nitrate . This work presents a comprehensive comparison of aqSOA formation from the direct photosensitized oxidation of VL  in the absence and presence of ammonium nitrate .

Lines 145-146: Moreover,  the photo-oxidation of guaiacol (GUA) (0.1 mM), a non-carbonyl phenol, in the presence of VL (0.1 mM) was studied .

Lines 211-213: In this work, the direct photosensitized oxidation of VL in the absence (VL only experiments) and  presence of ammonium nitrate are referred to as VL* and VL+AN, respectively.

Lines 554-556: In this study,  the direct photosensitized  oxidation of VL in the absence and  presence of ammonium nitrate under atmospherically relevant cloud and fog conditions have been shown to  generate aqSOA composed of oligomers, functionalized monomers, oxygenated ring-opening products, and nitrated compounds (from VL+AN).

**Minor comments:**

1. The updated explanation of pH-dependency of light absorbance is not clear to me. The pH range tested in this study was 2.5 – 4. In Fig. S10, the UV-Vis absorption spectra of VL*-derived aqSOA didn't change between the pH range of 1.5 – 4, though the intensity of absorption at longer wavelengths significantly increased when the pH was higher than 5.5

Response: In Fig. S10, we showed the changes in the UV-Vis absorption spectra of the aqSOA formed from VL* at pH 4 and 2.5 measured over a range of pH conditions from 1.5 to 10.5 to determine whether the observed pH dependence is due to the acid-base chemistry of the products or of the reactions. If the changes in the UV-Vis absorption spectra for the two solutions of varying pH are different, the pH dependence is due to different products formed from VL* at pH 4 compared to pH 2.5. However, the comparable changes in the UV-Vis absorption spectra for the two solutions indicate that the observed pH dependence (Fig. 3b) may be attributed to the acid-base chemistry of the reactions, which may involve $^3$VL* or the excimer of VL (Smith et al., 2016). Changes in the text are as follows:

Lines 411-418: Higher absorbance enhancement for both VL* and VL+AN (Fig. 3b) was observed as pH increased. To determine whether the pH dependence is due to the acid-base chemistry of the products or of the reactions,  the changes in the UV-Vis absorption spectra  of the aqSOA formed from VL* at pH 4 and 2.5 were measured over a range of pH conditions from 1.5 to 10.5 (Fig. S10). For both cases, the intensity of absorption at longer wavelengths significantly increased as the pH of the solutions was raised. Moreover, the changes in the UV-Vis absorption spectra for the two solutions of varying pH are comparable, suggestings that the observed pH dependence  is rooted in acid-base chemistry of the reactions involving $^3$VL* or the excimer of VL (Smith et al., 2016), as discussed earlier.

$<OS_c> = 2 \times <O{:}C> - <H{:}C>$                                                              (Eq. S3)

Based on the typical MS/MS fragmentation behavior for individual functional groups (Table S1)

and DBE values, examples of structures for products identified from VL (and GUA)

photo-oxidation experiments were proposed (Table S2).

**Text S6.** Photon flux measurements.

In this work, 2-nitrobenzaldehyde (2NB), a chemical actinometer, was used to determine the photon flux in the aqueous photoreactor. Briefly, the photolysis of 50 μM 2NB in the reactor was monitored by determining its concentration every 5 min for a total of 35 min, during which 2NB was almost completely decayed. The concentration of 2NB was measured using UHPLC-PDA, and the settings (e.g., column, mobile phase, gradient, oven temperature) were the same as those for VL decay analysis (Text S3). The channel with UV absorption at 254 nm was used for the quantification of 2NB. The concentration of 2NB in the reactor followed exponential decay, and its decay rate constant was determined using the following equation:

$$ln\left(\frac{[2\text{NB}]_t}{[2\text{NB}]_0}\right) = -j(2\text{NB}) \times t \qquad \text{(Eq. S4)}$$

where $[2\text{NB}]_t$ and $[2\text{NB}]_0$ are the 2NB concentrations at time $t$ and 0, respectively. The calculated 2NB decay rate constant, $j(2\text{NB})$, was 0.0026 s$^{-1}$. The following equation can also be used to calculate $j(2\text{NB})$:

$$j(2\text{NB}) = 2.303 \times (10^3 \text{ cm}^3 \text{ L}^{-1} \times 1 \text{ mol}/N_A \text{ mlc}) \times \Sigma\left(I'_\lambda \times \Delta\lambda \times \varepsilon_{2\text{NB},\lambda} \times \Phi_{2\text{NB}}\right) \quad \text{(Eq. S5)}$$

where $N_A$ is Avogadro's number, $I'_\lambda$ is the actinic flux (photons cm$^{-2}$ s$^{-1}$ nm$^{-1}$), $\Delta\lambda$ is the wavelength interval between actinic flux data points (nm), and $\varepsilon_{2\text{NB},\lambda}$ and $\Phi_{2\text{NB},\lambda}$ are the base-10 molar absorptivity (M$^{-1}$ cm$^{-1}$) and quantum yield (molecule photon$^{-1}$) for 2NB, respectively. Values of $\varepsilon_{2NB,\lambda}$ (in water) at each wavelength under 298 K and a wavelength-independent $\Phi_{2\text{NB}}$ value of 0.41 were adapted from Galbavy et al. (2010). Similar to Smith et al. (2014, 2016), we measured the spectral shape of the photon output of our illumination system (i.e., the relative flux at each wavelength) using a high-sensitivity spectrophotometer (Brolight Technology Co. Ltd,

Hangzhou, China). Using a scaling factor (SF), this measured relative photon output, $I_\lambda^{\text{relative}}$, is related to $I_\lambda'$ as follows (Hullar et al., 2020):

$I_\lambda' = I_\lambda^{\text{relative}} \times \text{SF}$                                                        (Eq. S6)

Substitution of Eq. S6 into Eq. S5 and rearrangement yields:

$\text{SF} = \dfrac{j(2\text{NB})}{2.303 \times (10^3 \text{ cm}^3 \text{ L}^{-1} \times 1 \text{ mol}/N_A \text{ mlc}) \times \sum(I_\lambda^{\text{relative}} \times \Delta\lambda \times \varepsilon_{2\text{NB},\lambda} \times \Phi_{2\text{NB}})}$     (Eq. S7)

and substitution of Eq. S6 into Eq. S7 yields:

$I_\lambda' = I_\lambda^{\text{relative}} \dfrac{j(2\text{NB})}{2.303 \times (10^3 \text{ cm}^3 \text{ L}^{-1} \times 1 \text{ mol}/N_A \text{ mlc}) \times \sum(I_\lambda^{\text{relative}} \times \Delta\lambda \times \varepsilon_{2\text{NB},\lambda} \times \Phi_{2\text{NB}})}$     (Eq. S8)

Finally, $I_\lambda'$ was estimated through Eq. S8. The estimated photon flux in the aqueous reactor is shown in Fig.ure S1. The actinic flux during a haze event over Beijing (40º N, 116º E) on January 12,

2013, at 12:00 pm (GMT+8) (Che et al., 2014) estimated using the National Center for

Atmospheric Research —Tropospheric Ultraviolet-Visible (TUV) Radiation Model (http://cprm.acom.ucar.edu/Models/TUV/Interactive_TUV/) is also shown in Figure S2. The parameters used for the Quick TUV calculator were: Overhead Ozone Column: 300 du; Surface

Albedo: 0.1; Ground Elevation: 0 km asl; Measured Altitude: 0 km asl; Clouds optical depth: 0, base: 4, top: 5; Aerosols optical depth: 2.5, single scattering albedo: 0.9, Angstrom exponent: 1;

Sunlight direct beam, diffuse down, diffuse up: 1; 4 streams transfer model. For clear days, the actinic flux was estimated over Beijing (at the same date and time) using the default parameters.

**Text S7.** Estimation of the apparent quantum efficiency of guaiacol photodegradation.

The apparent quantum efficiency of GUA photodegradation ($\varphi_{GUA}$) in the presence of either VL or nitrate during simulated sunlight illumination can be defined as (Anastasio et al., 1996;

Smith et al., 2014, 2016):

$\Phi_{GUA} = \dfrac{\text{mol GUA destroyed}}{\text{mol photons absorbed}}$ (Eq. S9)

$\Phi_{GUA}$ was calculated using the measured rate of GUA decay and rate of light absorption by either

VL or nitrate through the following equation:

$\Phi_{GUA} = \dfrac{\text{rate of GUA decay}}{\text{rate of light absorption by VL or nitrate}} = \dfrac{k'_{GUA} \times [\text{GUA}]}{\sum[(1-10^{-\varepsilon_\lambda[C]l}) \times I^t_\lambda]}$ (Eq. S10)

where $k'_{GUA}$ is the pseudo-first-order rate constant for GUA decay, [GUA] is the concentration of

GUA (M), $\varepsilon_\lambda$ is the base-10 molar absorptivity ($M^{-1} cm^{-1}$) of VL or nitrate at wavelength $\lambda$, [C] is the concentration of VL or nitrate (M), $l$ is the pathlength of the illumination cell (cm), and $I'_\lambda$ is the volume-averaged photon flux (mol photons $L^{-1} s^{-1} nm^{-1}$) determined from 2NB actinometry:

───────────

$j(2NB) = 2.303 \times \Phi_{2NB} \times l \times \sum_{300\,nm}^{350\,nm}(\varepsilon_{2NB,\lambda} \times I'_\lambda \times \Delta\lambda)$ (Eq. S11)

**Table S1.** Typical fragmentation behavior observed in MS/MS spectra for individual functional groups from Holčapek et al. (2010).

| Functional group | Fragment ions | MS/MS loss |
|---|---|---|
| Nitro ($RNO_2$) | $[M+H-OH]^{+\bullet}$ | -OH |
| | $[M+H-H_2O]^+$ | $-H_2O$ |
| | $[M+H-NO]^{+\bullet}$ | -NO |
| | $[M+H-NO_2]^{+\bullet}$ | $-NO_2$ |
| Nitroso (RNO) | $[M+H-NO]^+$ | -NO |
| Carboxylic acid (ROOH) | $[M+H-H_2O]^+$ | $-H_2O$ |
| | $[M+H-CO_2]^+$ | $-CO_2$ |
| | $[M+H-H_2O-CO]^+$ | $-H_2O-CO$ |
| Phenol (ROH) | $[M+H-H_2O]^+$ | $-H_2O$ |
| | $[M+H-CO]^+$ | -CO |
| Methoxy ($ROCH_3$) | $[M+H-CH_3]^{+\bullet}$ | $-CH_3$ |
| | $[M+H-CH_3O]^{+\bullet}$ | $-CH_3O$ |
| | $[M+H-CH_3OH]^+$ | $-CH_3OH$ |
| | $[M+H-HCOH]^+$ | -HCOH |
| Ester ($R^1COOR^2$) | $[M+H-R^2OH]^+$ | $-R^2OH$ |
| | $[M+H-R^2OH-CO]^+$ | $-R^2OH-CO$ |
| Amine | $[M+H-NH_3]^+$ | $-NH_3$ |
| Aldehyde (RCHO) | $[M+H-CO]^+$ | -CO |

**Table S2.** Examples of proposed  structures for products detected  from
vanillin (and guaiacol)-photo-oxidation experiments in this study.

| No. | Formula | DBE | Proposed structure | MS/MS fragment ions | | |
|-----|---------|-----|--------------------|---------------------|---|---|
| 1 | $C_8H_8O_3$ (VL; triplet and aqSOA precursor) | 5 |  | -CO-CH$_3$OH | -CO | -CO-CH$_3$OH-CO |
| 2 | $C_7H_8O_2$ (GUA; aqSOA precursor) | 4 |  | | | |
| 3 | $C_8H_9NO_3$ | 5 |  | -CO-CH$_3$ | -NH$_3$ | |
| 4 | $C_{16}H_{10}N_2O_9$ | 13 |  | -NO$_2$ | | |
| 5 | $C_{10}H_{10}O_5$ | 6 |  | -CH$_3$OH | -CH$_3$OH-CO | |
| 6 | $C_5H_5N_3O_2$ | 5 |  | -NH | | |
| 7 | $C_{16}H_{14}O_6$ | 10 |  | -CO-CH$_3$OH-CO | -CO-CH$_3$OH-CO-CH$_3$OH | -CO-CH$_3$OH-CO-CO |

| | | | | | | |
|---|---|---|---|---|---|---|
| 8 | C$_7$H$_6$O$_3$ | 5 |  | | | |
| 9 | C$_8$H$_8$O$_4$ | 5 |  | -CO-CH$_3$OH | -CO | -H$_2$O |
| 10 | C$_{15}$H$_{12}$O$_8$ | 10 |  | -COOH | | |
| 11 | C$_7$H$_4$N$_2$O$_7$ | 7 |  | | | |
| 12 | C$_8$H$_6$O$_4$ | 6 |  | -CO | -CO-CO | |
|  |  |  |  |  | | |
|  |  |  |  | | | |
| 13 | C$_{14}$H$_{14}$O$_4$ | 8 |  | | | |

| | | | | | | |
|---|---|---|---|---|---|---|
| 11 | $C_{21}H_{20}O_6$ | 12 | | | | |
| 15 | $C_{28}H_{24}O_8$ | 16 | | | | |
|  |  |  | |  |  | |
|  |  |  | |  |  |  |
| 16 | $C_7H_4O_4$ | 6 | | -CO | -CO-CO | |

[Figure]

**Figure S1.** The base-10 molar absorptivities ($\varepsilon$, $M^{-1}$ $cm^{-1}$) of vanillin (VL, green solid line), 2-
nitrobenzaldehyde (2NB, green dotted line), guaiacol (GUA, green dashed line), $NO_2^-$ (blue
dashed line), $NO_3^-$ (blue solid line), and photon flux in the aqueous reactor (red line) during typical
haze days (black line) or clear days (grey line) in Beijing, China. The $\varepsilon$ values for 2NB and $NO_2^-$
were adapted from Galbavy et al. (2010) and Chu and Anastasio (2007), respectively.

[Figure]

**Figure S2.** Calibration curves for (a) VL and (b) GUA standard solutions (10–130 µM). Error
bars represent one standard deviation.

[Figure]

**Figure S3.** (a–c) The decay of VL under different experimental conditions for direct
photosensitized oxidation of VL in the absence (VL*) and presence of ammonium nitrate mediated
VL photo-oxidation (VL+AN): (a) VL* and VL+AN at pH 4 under N₂- (A6, A8) and air-saturated
(A5, A7) conditions. No statistically significant difference ($p > 0.05$) was noted between VL+AN
(A7) and VL+SN (A9; not shown here). (b) Effect of pH on VL* and VL+AN at pH 2.5 (A1, A2),
3 (A3, A4), and 4 (A5, A7) under air-saturated conditions. (c) The decay of VL (and GUA) during
direct GUA photodegradation (A13) and photo-oxidation of GUA in the presencevia
photosensitized reactions of VL (GUA+VL; A14) or nitrate (GUA+AN; A15) at pH 4 under air-
saturated conditions after 6 h of simulated sunlight irradiation. Error bars represent one standard
deviation; most error bars are smaller than the markers.

[revised manuscript text omitted]

---

## Author Response (AR3)

Author Response for "Aqueous SOA formation from the direct photosensitized oxidation of vanillin in the absence and presence of ammonium nitrate" by Mabato et al.

Comments to the author: Dear authors, Thank you for addressing and clarifying the remaining questions of the reviewers. The revised manuscript has been largely improved and clearer. I have one more question, was O3 involved in the reactions or it had been excluded by the use of ozone free xenon lamp.

Response: We thank the editor for the positive comments. As the lamp was situated outside the reactor in our experiments, ozone generating or ozone free lamp will not affect the reactions. We have clarified that ozone is not involved in the reactions as follows (additional text in red):

[revised manuscript text omitted]